# Caveolin-1 dolines form a distinct and rapid caveolae-independent mechanoadaptation system

Fidel-Nicolás Lolo [1] ✉, Nikhil Walani [2,18], Eric Seemann [3,18], Dobryna Zalvidea[4,5], Dácil María Pavón[1,17], Gheorghe Cojoc[6], Moreno Zamai [7], Christine Viaris de Lesegno[8], Fernando Martínez de Benito [9,10], Miguel Sánchez-Álvarez[1], Juan José Uriarte[11], Asier Echarri [1], Daniel Jiménez-Carretero[9], Joan-Carles Escolano[6,12], Susana A. Sánchez[13], Valeria R. Caiolfa[7,14], Daniel Navajas[4,11], Xavier Trepat [4,11,15,16], Jochen Guck [6,12], Christophe Lamaze [8], Pere Roca-Cusachs [4,11], Michael M. Kessels [3], Britta Qualmann [3], Marino Arroyo [2,4] & Miguel A. del Pozo [1]✉

In response to different types and intensities of mechanical force, cells modulate their physical properties and adapt their plasma membrane (PM). Caveolae are PM nano-invaginations that contribute to mechanoadaptation, buffering tension changes. However, whether core caveolar proteins contribute to PM tension accommodation independently from the caveolar assembly is unknown. Here we provide experimental and computational evidence supporting that caveolin-1 confers deformability and mechanoprotection independently from caveolae, through modulation of PM curvature. Freeze-fracture electron microscopy reveals that caveolin-1 stabilizes non-caveolar invaginations—dolines—capable of responding to low-medium mechanical forces, impacting downstream mechanotransduction and conferring mechanoprotection to cells devoid of caveolae. Upon cavin-1/PTRF binding, doline size is restricted and membrane buffering is limited to relatively high forces, capable of flattening caveolae. Thus, caveolae and dolines constitute two distinct albeit complementary components of a buffering system that allows cells to adapt efficiently to a broad range of mechanical stimuli.

The interplay between cells and mechanical cues determines organismal development, cancer behaviour or cardiovascular physiology and disease[1]. Changes in plasma membrane (PM) tension are sensed, transduced and accommodated through as yet poorly characterized molecular mechanisms[2]. Eisosomes couple changes in PM tension to nutrient transport[3]. Dynamin-independent pathway CLIC/GEEC-regulated endocytosis can also modulate PM tension[4]. Caveolae[5] are small, flask-like PM invaginations with distinct lipid (enriched for cholesterol and saturated phospholipids) and scaffolding protein composition[6,7]. Caveolin-1 (Cav1) and cavin-1/polymerase I and transcript release factor (PTRF), strictly required for caveolae formation in virtually all tissues[8–11], are tightly co-regulated, and depletion of one scaffold leads to robust downregulation of the other[9,12]. Beyond signalling module organization and membrane internalization regulation[6,13], caveolae are key elements for sensing and transducing mechanical forces[5,6,14,15]. Tissues subject to wide variations of PM tension, such as

endothelium, muscle, fibroblasts or adipocytes, exhibit a high density of caveolae and require them for mechanical homeostasis[16–18]. Robust mechanical stress induces caveolae flattening, Cav1 scaffolds disassemble and PTRF is released into the cytoplasm[14,19]. However, these mechanisms fail to explain how cells sense and transduce low-range forces at short timescales. This is a critical shortcoming because a large share of biological processes involve mechanical forces below those required experimentally to observe caveolae flattening[20–24]. Furthermore, cell types such as lymphocytes or neurons[25,26] are devoid of caveolae but do express Cav1, which can organize discrete PM domains of different sizes, termed 'scaffolds', in the absence of PTRF in mammalian cells;[27,28] similar structures are observed in invertebrates[29]. However, whether core components such as Cav1, independently of caveolae, contribute to PM physicochemical organization and tension accommodation is unknown.

In this Article, we developed genetically engineered PTRFKO mouse embryonic fibroblast (MEF) lines to express endogenous levels of Cav1, while unable to stabilize caveolae. Cav1 re-expression protected caveolae-null cells from hypo-osmotic shock to an extent comparable to wild-type (WT) cells. Orthogonal biophysics and cell biology approaches showed that Cav1 increases cellular deformability, allows cells to mechanically adapt to forces exerted on the PM and transduce this mechanical information, and buffers changes in PM tension, in the absence of caveolae. Cav1 scaffolds PM invaginations, which we name dolines. PTRF expression restricts their size and limits caveolar mechanosensing and mechanoprotection to high forces. Endogenous Cav1 expression in neurons (devoid of caveolar structures) is required for mechanoprotection. Our results support a continuum model for Cav1-dolines and caveolae as a buffering system with different degrees of complexity, ranging from Cav1 clusters—capable of membrane bending in response to a wide range of forces—to fully assembled caveolae, which flatten upon exposure to higher forces beyond a certain threshold.

## Results

### Cav1 protects against hypo-osmotic shock

Knockout of either Cav1 or PTRF leads to substantial downregulation of the other[9,11,12]. Cav1 plays caveolae-independent roles[30–32], consistent with the presence of Cav1 pools not co-localizing with PTRF[33]. To understand the roles of Cav1 independently from PTRF and caveolae, we generated isogenic cell lines from PTRFKO MEFs, either re-expressing PTRF (and hence, Cav1; PTRFKO + PTRF MEFs, referred as control) or selectively re-expressing Cav1 to endogenous levels, while lacking PTRF expression (PTRFKO + Cav1 MEFs). (Fig. 1a–c). Endogenous expression levels in rescued cell lines were confirmed (Extended Data Fig. 1a,c). PTRF knockdown increases a ubiquitylated pool of Cav1 (ref. 34), we thus assessed Cav1 ubiquitylation across genotypes. We identified Cav1-specific, ubiquitin-positive bands in cells expressing Cav1 but not in Cav1-depleted cells (Extended Data Fig. 1e)[34,35]. There was more ubiquitinated Cav1 in PTRFKO + Cav1 cells, despite having similar Cav1 protein levels as compared with control cells (Extended Data Fig. 1f).

We further characterized the subcellular distribution of re-expressed proteins. A pool of Cav1 localizes to the PM (Extended Data Fig. 1b,d). The presence or absence of caveolae was then assessed by electron microscopy (EM) across genotypes. Caveolae were observed in control cells, but not in PTRFKO nor in PTRFKO + Cav1 MEFs (Fig. 1d–f, black arrows). Thus, our system bypasses biological effects derived from the structural contribution of caveolae, and is suitable for the characterization of Cav1-intrinsic, caveolae-independent mechanoadaptation.

We studied whether Cav1 alone confers mechanoprotection when cells are subjected to hypo-osmotic shock. Cells swell upon acute decrease of extracellular osmolarity, leading to increased PM tension and rupture[36]. As expected, PTRFKO MEFs exhibited significantly

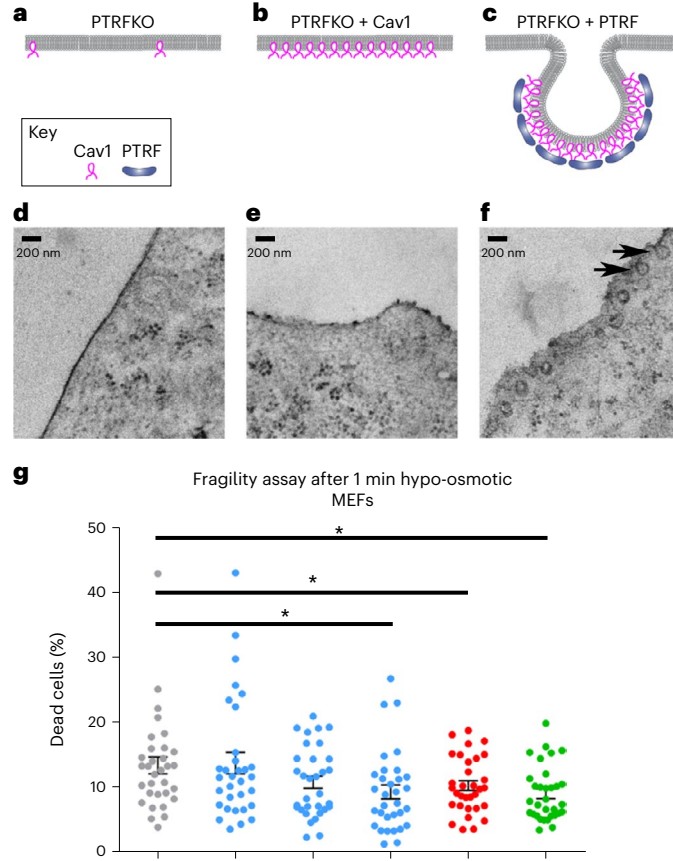

**Fig. 1 | Cav1 confers protection to hypo-osmotic treatment. a–c**, Schematic representations of the different caveolae-related phenotypes. **d–f**, EM images of PM regions from MEFs, showing the presence of caveolae exclusively in PTRFKO MEFs reconstituted with PTRF (**f**, black arrows). **g**, Percentage of dead cells after hypo-osmotic shock (fragility assay in suspension; for details, see Methods) across indicated genotypes. Data are presented as mean ± s.e.m. $n = 32$ independent fragility assays. Statistical comparisons were by two-tailed Student's $t$-test (comparing PTRFKO with either PTRFKO + Cav1 high expression, $P = 0.0195$; PTRFKO + PTRF, $P = 0.0434$; or MEFs WT, $P = 0.0048$), with significance assigned at *$P < 0.05$. Source numerical data are available in source data.

lower survival rates as compared with either control cells or WT MEFs, which were indistinguishable, validating PTRFKO + PTRF as reference cell line (Fig. 1g). Strikingly, PTRFKO + Cav1 MEFs were resistant to hypo-osmotic shock-induced rupture to an extent comparable to control cells (Fig. 1g). This effect was dose-dependent when comparing PTRFKO + Cav1 subpopulations sorted by their Cav1 re-expression levels (Fig. 1g and Extended Data Fig. 1g–n). These results are consistent with an intrinsic, caveolae-independent role for Cav1 in mechanoprotection against PM rupture[29].

### Cav1 and caveolae regulate different cell mechanical properties

To better understand the biophysics of the mechanoprotective role of Cav1, we first studied deformation dynamics across all genotypes at different timescales, using (1) real-time deformability cytometry (RT-DC; Methods), a high-throughput technique capturing response times at millisecond scale;[37] and (2) optical stretching (OS; Fig. 2a, Extended Data Fig. 2e and Supplementary Video 1), measuring cell mechanics at second scale[38]. No significant differences were observed across genotypes with RT-DC (Extended Data Fig. 2a–d), which might indicate that longer deformation times are required to reveal any differences in cellular elasticity. We observed by OS that control and

PTRFKO + Cav1 MEFs exhibited higher deformability as compared with PTFKO MEFs (Fig. 2b and Extended Data Figs. 2f,g). These results suggest that Cav1, independently from its organization into caveolae, contributes to cell mechanics. We characterized stiffness in adhered cells across genotypes by atomic force microscopy (AFM; Fig. 2c)[39,40] at regions distant from the nucleus. PTRFKO + Cav1 MEFs were more compliant than PTRFKO MEFs, and exhibited cellular stiffness similar to that displayed by control cells (Fig. 2d).

To better evaluate PM mechanics, we used magnetic tweezers (Fig. 2e–g). Magnetic beads attached to the cell surface are pulled and oscillated by applying a pulsatory magnetic force (1 Hz) of 1 nN (ref. 41) (Supplementary Video 2), and local stiffness of the bead–cell interface is inferred from the relationship between the applied force and the resulting bead movement. Cells can respond through a phenomenon known as reinforcement, by which they gradually strengthen cell–bead adhesion and increase its stiffness. We discriminated forces transmitted directly through the PM from those channelled through integrins and focal adhesions, by coating beads with either concanavalin A (ConA) or fibronectin (FN) (Fig. 2e). Analysis with ConA-coated beads revealed that PTRFKO MEFs develop higher reinforcement as compared with control cells, which showed buffering abilities as expected (Fig. 2i). PTRFKO + Cav1 MEFs displayed reinforcements similar to control cells (Fig. 2i). We observed no differences in cell adhesion to ConA-coated plates (Fig. 2k), indicating that the observed reinforcement differences are not due to differential cell adhesion, nor net surface glycoprotein density. No differences were observed in experiments performed with FN-coated beads (Fig. 2h), suggesting that integrin-driven mechanosensing is similar across genotypes; neither did we detect differences in cell adhesion to FN-coated plates (Fig. 2j). Thus, caveolar and non-caveolar Cav1 PM structures have intrinsic responsiveness to mechanical cues.

## Cav1 alone buffers PM tension in response to osmotic swelling

To specifically measure PM tension buffering, we applied optical tweezers (OTs, Fig. 3a)[14]. PTRFKO MEFs exhibited increased PM tension after hypo-osmotic shock (Fig. 3b), as shown before[14]. Control cells displayed a behaviour indistinguishable from WT cells (Fig. 3b and Extended Data Fig. 3a), showing significant relative buffering as reported[14]. PTRFKO + Cav1 MEFs phenocopied control cells and did not show significant increases in PM tension, supporting that Cav1 provides a buffering system in the absence of caveolae (Fig. 3b). We further measured the response to PM tension changes across discrete subpopulations of PTRFKO + Cav1 cells, sorted by Cav1 re-expression levels. We observed a direct positive correlation between tension buffering and Cav1 expression levels (Fig. 3c). These results suggest that Cav1 constitutes a novel, caveolae-independent PM mechanoadaptation system.

## Cav1 forms heterogeneously sized clusters in the absence of PTRF

Cav1 is predicted to induce membrane curvature and cholesterol clustering at the PM[42], in agreement with our structural model

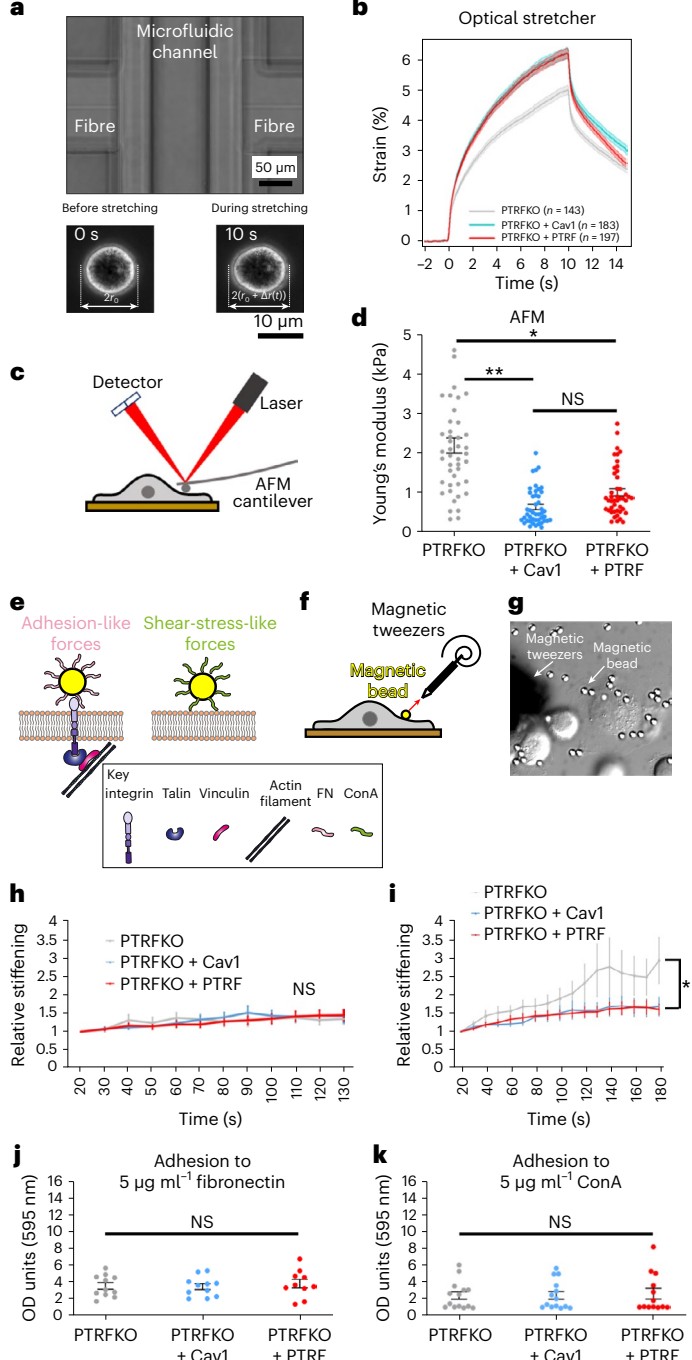

**Fig. 2 | Cav1 increases cellular deformability. a**, Phase contrast micrograph of the trapping region of the Optical Stretcher device. The position of the opposing optical fibres facing a hollow glass capillary where cells are trapped is indicated. Underneath, a cell deformation example, showing phase contrast images before and during stretching. **b**, Deformation curves (strain in percentage versus time in seconds) across genotypes. The number of cells analysed per genotype is indicated. **c**, AFM experiment scheme, indicating the fibroblast, the laser beam (red), the detector (blue) and the AFM cantilever. **d**, Young's modulus measured with AFM of the different MEF genotypes, at mid-distance between nucleus and cell edge. $n = 15$ cells each from three independent AFM measurements. Statistical comparisons were by one-way ANOVA (Holm–Sidak pairwise multiple comparison analysis, $*P = 0.0216$ and $**P = 0.0066$). (NS, non-significant, $P = 0.367$). **e**, Representation of the two magnetic beads coatings used: FN and ConA. **f**, Reinforcement experiment scheme, indicating the fibroblast, the magnetic bead and the magnetic tweezers apparatus. The red arrow represents the magnetic force exerted on the bead by the magnet. **g**, DIC image showing an MEF, the tip of the magnetic tweezers and magnetic beads (white arrows). **h,i**, Mean stiffness (reinforcement) of beads as a function of time normalized by initial stiffness of the different MEF genotypes for FN-coated beads (**h**) or ConA-coated beads (**i**). A value of 1 indicates no change in stiffness with respect to initial value; greater values show stiffening. $n = 20$ beads per genotype pooled from four independent experiments. Two-tailed Student's *t*-test ($P = 0.0459$; NS, non-significant). **j,k**, Relative adhesion of the indicated genotypes (expressed as optical density, OD, at 595 nm) to plates coated with 5 µg ml⁻¹ FN (**j**) or 5 µg ml⁻¹ ConA (**k**). $n = 11$ adhesion independent experiments for PTRFKO and PTRFKO + Cav1, $n = 9$ adhesion independent experiments for PTRFKO + PTRF. Statistical analyses were by one-way ANOVA. NS, non-significant. For **b**, **d** and **h–k**, data are presented as mean ± s.e.m. Source numerical data are available in source data.

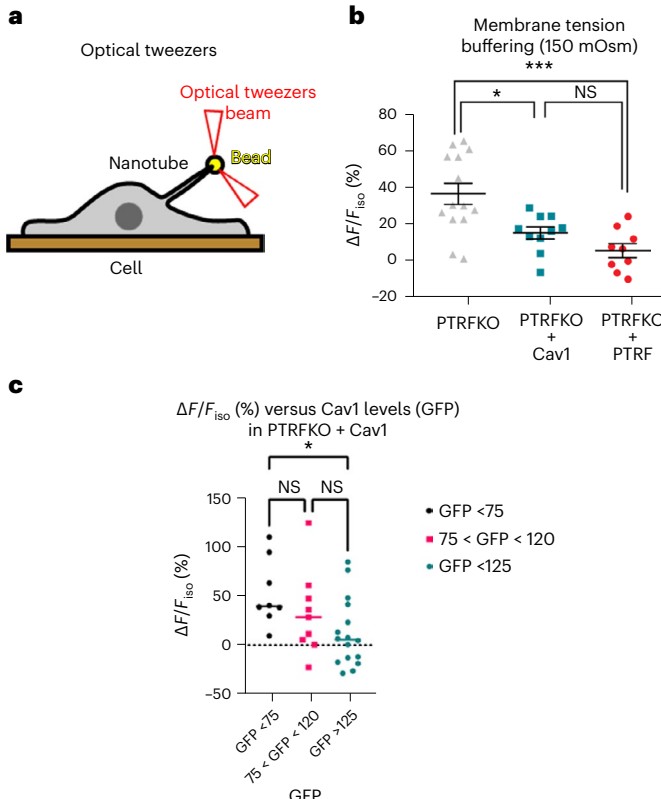

**Fig. 3 | Cav1 buffers tension in the absence of caveolae. a**, OT experiment scheme, indicating the cell, the OT beam, the cellular nanotube and the bead attached to the cell surface. **b**, Relative change of the mean tether force after hypo-osmotic shock (150 mOsm) for PTRFKO MEFs reconstituted with PTRF ($n = 9$), Cav1 ($n = 10$) or empty vector ($n = 14$). $n$ indicates number of cells pooled from eight independent experiments. **c**, Relative change of the mean tether force after hypo-osmotic shock (60 mOsm) as a function of GFP intensity (which correlates with Cav1 levels, Extended Data Fig. 1; a. u., arbitrary units) of PTRFKO reconstituted with Cav1 ($n = 33$). $n$ indicates number of cells pooled from six independent experiments. For **b** and **c**, individual values are plotted (data are presented as mean ± s.e.m.), statistical analysis strategy used was one-way ANOVA with Tukey's multiple comparisons test, with significance assigned at $*P < 0.01$ and $***P < 0.0003$, NS, non-significant. Source numerical data are available in source data.

(see below and Extended Data Fig. 4r–u). Such interplay may indicate a molecular self-assembly, concentration-dependent mechanism[43]. To study the organization of Cav1 in the absence of PTRF and better understand the mechanisms by which Cav1 regulates PM tension, we first assessed 2D Cav1 distributions by dSTORM across genotypes. PTRFKO MEFs could be hardly imaged, showing very few labelled Cav1 molecules under our experimental conditions (Extended Data Fig. 3b). Conversely, Cav1 appeared as sparse and organized clusters of multiple sizes in PTRFKO + Cav1 MEFs, control cells and WT MEFs as determined by Feret diameter[44] (Fig. 4a–c and insets; frequency plots in Fig. 4e). Clusters >60 nm diameter were observed in all cell lines, comprising ~50% of the total (frequency plot, Fig. 4e and representative examples; insets Fig. 4a–c). No significant differences were observed in cluster density (density plot in Fig. 4d), nor size distribution (frequency plot in Fig. 4e). Nevertheless, size heterogeneity and density variability were more evident in PTRFKO + Cav1 MEFs cells than in control and WT cells expressing endogenous Cav1 (frequency plot, Fig. 4e), as inferred from larger statistical deviations, presumably owing to the absence of PTRF. These observations suggest that size heterogeneity might correlate with Cav1 expression levels in PTRFKO + Cav1 cells representing different buffering capacities, as observed across PTRFKO + Cav1 subpopulations with OT.

## Cav1 increases cholesterol stabilization in the absence of PTRF

We analysed how Cav1 impacts PM cholesterol distribution and organization, which can in turn affect PM mechanics, by measuring fluorescence lifetime of 25 NBD-cholesterol (which recapitulates uptake rates, subcellular distribution displayed by native cholesterol and reflects differences in membrane order)[45–47] using the phasor-fluorescence-lifetime imaging microscopy (FLIM) method for lifetime data analysis[48]. Control cells exhibited homogeneous distributions of cholesterol as compared with other genotypes, with lifetime values across the cell body within a narrow range (medium lifetime, Extended Data Fig. 4c,c',d,e,f–q). PTRFKO MEFs showed a wider distribution clearly shifted towards short lifetimes (Extended Data Fig. 4a,a',d,e), indicative of fewer Cav1 organized domains (in accordance with less membrane order, see below and freeze-fractured images; Fig. 5h). PTRFKO + Cav1 MEFs showed reduced cholesterol organization heterogeneity, shifting back lifetime measurements to ranges compatible with increased membrane order (Extended Data Fig. 4b,b',d,e). We thus analysed changes in membrane order as inferred by Laurdan generalized polarization (GP) imaging[49,50], at different timepoints after sustained stretching (Extended Data Figs. 5a–d and 6c–e and Methods). Control cells exhibited a reduction in membrane order after 10 min of stretching, indicating that membrane phases become more homogeneous (Extended Data Fig. 5e,f). This effect was completely abrogated in PTRFKO MEFs, but still detectable in PTRFKO + Cav1 cells, suggesting that this phenotype is at least partially Cav1 dependent (Extended Data Fig. 5e). Membrane order was progressively recovered over time in control cells (Extended Data Fig. 6a,b), suggesting that ordered domains, including Cav1 clusters, reform under constant membrane tension. Thus, Cav1 expression affects cholesterol condensation (that is, stabilization) and membrane order in the absence of PTRF. To get further insight into this interpretation, we developed an in silico structural model for Cav1 (ref. 51) (Extended Data Fig. 4r–u, Supplementary Videos 9 and 10 and Methods), which supported that dimer spacing allows for increased cholesterol condensation (Extended Data Fig. 4t,u), providing local membrane buffering capability.

## Cav1 forms large invaginations—dolines—in the absence of PTRF

Dispersion of flat cholesterol-rich Cav1 clusters, leading to phase homogenization and membrane buffering[52] would afford for a very small buffering capacity (Supplementary Note 1). We hypothesized that Cav1 could bend the PM, as suggested by our own structural model (Extended Data Fig. 4r–u), molecular simulations[42] and observations in invertebrates[29]. Three-dimensional super-resolution imaging and mathematical reconstruction analysis have recently revealed that Cav1 'scaffolds' are formed in the absence of PTRF[27,28]. However, such structures have not been characterized at ultrastructural level. We applied a freeze-fracturing procedure (which circumvents previously described fixation-related artefacts[53]), platinum shadowing and anti-Cav1 immunogold labelling to visualize Cav1 clusters (Fig. 5a–d; for anti-Cav1 immunogold labelling densities in the different cell lines, see Extended Data Fig. 7a–d). This allows for evaluating the 3D topology of large cell membrane areas[18,54]. The method also allows for analysing membrane deformations as tomograms to obtain further in-depth 3D information[18]. In line with previous observations[11], anti-Cav1 immunolabelling density at the PM of PTRFKO MEFs was reduced to about 20 per μm⁻², ~40% of the value determined at control PTRFKO + PTRF MEFs and WT MEFs membranes, respectively (Fig. 5e, absolute values are shown). However, PTRFKO + Cav1 MEFs showed Cav1 densities not statistically different from those of the control or the WT cells (Fig. 5e), further supporting the validity of our cell model to restore Cav1 pools in the absence of caveolae.

We observed a reduction in invaginated structures with classical caveolae-like appearance (70 nm in diameter, uniformly round, deeply invaginated) and in shallow caveolae-like appearance (positive for anti-Cav1 immunostaining) in both PTRFKO MEFs and PTRFKO + Cav1

MEFs, as compared with control or WT cells (Fig. 5a–d,f,g). We classified Cav1-positive signals according to their arrangement in 'clusters', as opposed to 'disperse' positive signals (for details, see Methods). While PRTFKO + Cav1 MEFs showed Cav1 levels at the PM comparable to those in control cells (Fig. 5e), the density of clustered anti-Cav1 labels dropped by ~50% (Fig. 5h); conversely, the density of disperse anti-Cav1 labelling increased in PTRFKO + Cav1 cells to 20 µm$^{-2}$, almost doubling that observed in PTRFKO + PTRF control cells, and more than five-fold higher as compared with that observed in WT cells (Fig. 5i). Thus, PTRF promotes the clustering of Cav1; however, even in the absence of PTRF, Cav1 retained at least some ability to form membrane-associated clusters.

We noticed in PTRFKO-Cav1 cells an increased occurrence of unusual Cav1-immunopositive membrane topologies (Fig. 5j,j'). These membrane structures did not resemble caveolar structures at all (compare Fig. 5j and Fig. 5a): they were unusually large, had irregular morphologies as opposed to more spherical caveolae and often appeared almost flat in top views onto the freeze-fractured membranes, as they often lacked substantial shadowing. We also observed smaller versions of these structures in WT cells (Fig. 5k), ruling out that they derived from non-endogenous Cav1 re-expression in PTRFKO + Cav1 MEFs (Methods and Extended Data Fig. 7e,f).

We conducted 3D EM tomography[53] on these preparations (Supplementary Videos 3 and 4). Cav1-decorated membrane profiles in PTRFKO + Cav1 cells were invaginations, often deeper than suggested by shadowing in perpendicular view, resembling a pan or a wok, that is, with low initial membrane curvature at the rim of the invagination, resulting in wide openings (Fig. 5l,m) and perhaps explaining the modest depth suggested by shadowing techniques (Fig. 5j,j'). Depth frequently reached far beyond ≤100 nm—typical 'classic' caveolae depth range (see Fig. 5n,o and for comparison, and Fig. 5m). Because of their resembling karstic processes forming big depressions in the ground upon collapse of the surface layer—such as the 'Gran Dolina', a key element at the archaeological site of Atapuerca, Burgos, northern Spain[55]—we propose the term dolines for these invaginations. Dolines exhibited high variability in their diameters (Fig. 5p) and lower abundance as compared with 'classical' caveolae (0.2 µm$^{-2}$ versus 1.6 µm$^{-2}$) (Fig. 5q,f). Lack of 3D-topology resolution and perpendicular views on large membrane areas—required for identification and reliable analyses of their occurrence—might explain the absence of previous descriptions of these structures. Both 'classical' caveolae and shallow caveolae have defined, smaller average diameters (~70 nm and ~90 nm, respectively) and much lower diameter variability, clearly distinguishing them from these novel structures (Fig. 5a,p,q). Diameter distribution analyses confirmed that only a small fraction of the observed non-caveolar structures had diameters that could at least theoretically still represent incorrectly classified flat caveolae (<100 nm) (Fig. 5p). The vast majority of Cav1-positive non-caveolar structures in WT, PTRFKO + Cav1 and PTRFKO MEFs (87–90% in these three types of MEF) were larger than 100 nm in diameter (Fig. 5p). Strikingly, the size of anti-Cav1 immunopositive dolines clearly depended on PTRF: absence of PTRF allowed for the assembly of extremely large dolines (Fig. 5r), ranging from 300 nm to giant structures of up to almost 700 nm in diameter, that is, ten-fold that of classical caveolae. These giant dolines were undetectable in MEFs expressing PTRF (Fig. 5p,r and Extended Data Fig. 7g). In line with the hypothesis that PTRF seems to be important for restricting the growth of such non-caveolar structures, re-expression of PTRF in PTRFKO cells (PTRFKO + PTRF) led to few dolines (Fig. 5q) and the Cav1-positive, non-caveolar structures that were still observable also were much smaller than those in WT cells (Fig. 5p, red shadowing; Fig. 5s).

## Mathematical modelling supports dolines and caveolae behaviour

Caveolae buffer tension by releasing membrane area as they flatten out[14,56], but how Cav1-mediated buffering in cells devoid of caveolar

structures works is unclear. We developed an axisymmetric computational continuum model of these structures (for more details, see ref. 57 and Supplementary Note 1). We chose parameters so that assembly of protein-rich domains is mediated by curvature, not by strongly favourable protein–protein interactions (Supplementary Note 1), consistent with the fact that Cav1 molecules disperse upon tension-mediated disassembly of caveolae[14] and of Cav1 structures in our PTRFKO + Cav1 cells (Figs. 4 and 5). Thus, predicted protein-rich domains were curved. A key parameter in our model is the spontaneous curvature of the protein coat. Consistent with previous models[58], we hypothesized that spontaneous curvature of Cav1 coats ('Cav1 model') was smaller—about 1/200 nm$^{-1}$—than that of full Cav1-PTRF coats ('PTRF model')−1/50 nm$^{-1}$. We found that, at low tension, the high-spontaneous curvature PTRF model resulted in the formation of budded protein-rich domains with narrow necks (Fig. 6a,a1 and Supplementary Video 5), whereas the low-spontaneous curvature Cav1 model resulted in shallower curved domains with wide necks, (Fig. 6a,b1 and Supplementary Video 6). Upon increased tension within physiological limits, both of these protein-rich domains flattened; while the domains in the PTRF model flattened through a sharp snapping—a consequence of a bi-stable switch between a flat homogeneous state and a protein-rich curved domain—domains in the Cav1 model unfolded continuously as tension increased (Fig. 6a and Supplementary Videos 7 and 8). Furthermore, the PTRF model exhibited hysteresis upon unloading. To understand if and how this qualitative difference had an effect on the buffering behaviour of large membrane areas containing many domains, we developed an extended thermodynamic model whereby the shape of each domain was simplified[56] but the number of domains could change (Fig. 6b and Supplementary Note 1). Given tension and average protein coverage, the model predicts the number density, protein enrichment and shape of domains for either the Cav1-only or the PTRF model by minimizing free energy. At low tension and for a given average protein coverage, the Cav1 model organizes into very large domains with a contact angle smaller than 90°, whereas the PTRF model develops a distribution of fully budded spherical domains of smaller size (Fig. 6c1,c2). The models exhibit radically different responses to increasing tension. While the Cav1 model adapts by splitting sparse, large and deep domains into several smaller and shallower ones, the PTRF model progressively reduces the number of domains without changing their shape. Thus, our model suggests two distinct mechanisms by which the membrane responds to tension by either splitting/coalescing shallow domains of variable shape and size, or by snapping/assembly of domains of very precise shape and size.

We then looked at the buffering capacity quantified by the projected areal strain of the membrane as a function of tension. We found that the Cav1 model was able to release more area at lower tension, but then its buffering capacity saturated for relatively low tensions at low areal strains (Fig. 6c3). In contrast, the PTRF model was stiffer at low tension, where it exhibited a coverage-independent mechanical response, but was able to release more area at high tension in a coverage-dependent fashion. By assuming that the absence of PTRF reduces the spontaneous curvature of the protein coat, our theoretical modelling suggests a mechanism of tension buffering by large non-caveolar Cav1-rich domains with open necks. In agreement with our observations, this model predicts that the size of such domains is highly variable and can reach several hundreds of nanometres at low tension. Furthermore, according to our theoretical results, the mechanoprotection provided by these non-caveolar domains should depend on the expression levels, in agreement with our observations (Figs. 1g and 3c), and should be stronger at low tensions and weaker at high tensions.

## Dolines and caveolae respond differently to membrane tension

To experimentally test the differential buffering abilities of Cav1 dolines versus caveolae, as predicted by the mathematical model, we firstly subjected cells from each genotype to an extended range of osmotic

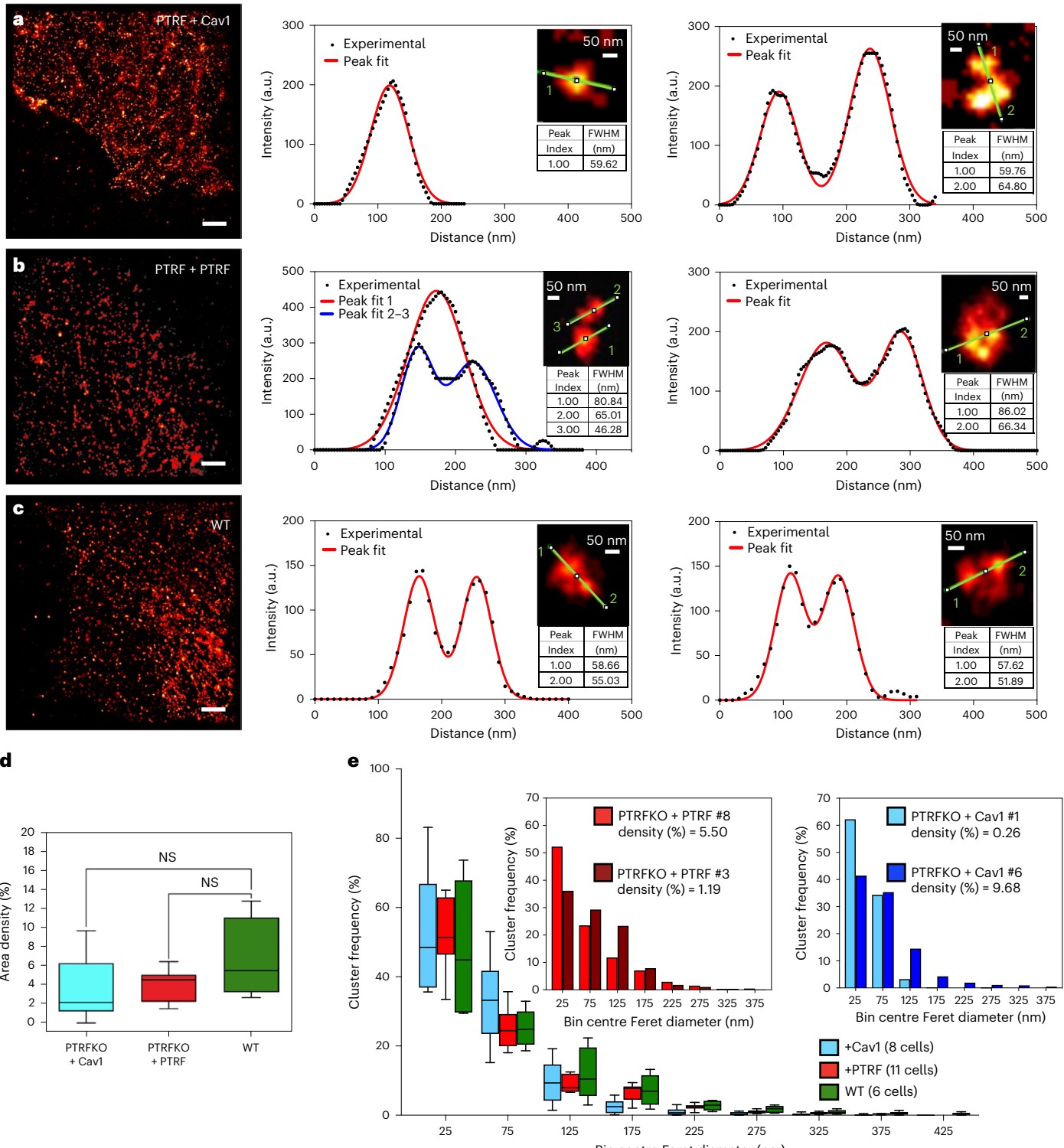

**Fig. 4 | Cav1 forms clusters of different sizes in the absence of PTRF. a–c,** dSTORM representative images of MEFs of the indicated genotypes (left) and analyses of clusters showing the size and separation of nanostructures (right). These plots were taken from the corresponding dSTORM images and show the cutting curves (green) for profile analysis and full width at half maximum (FWHM) for each fitted peak in representative clusters. Scale bar, 2 µm; a.u., arbitrary units. Results are representative of 6 WT, 10 PTRFKO + PTRF and 11 PTRFKO + Cav1 cells from two independent replicates. **d,** Cluster density expressed as percentage area for the different MEF genotypes. **e,** Normalized cluster frequency segmented according to the Feret diameter (longest distance between any two points along the selection boundary[44]; Results) of PTRFKO + Cav1 MEFs (cyan), PTRFKO + PTRF MEFs (red) and Cav1WT MEFs (green). Insets: normalized cluster frequency of two cells from PTRFKO + Cav1 and PTRFKO + PTRF lines chosen at the minimum (cyan and brown bars) and maximum (blue and red bars) values of cluster density in plot (**d**). For **d** and **e**, boxes show the first quartile (Q1) to the third quartile (Q3) and the central line corresponds to the median. The whiskers go from Q1/Q3 quartile to the lowest/greatest observed data point that falls at a distance of 1.5 times the IQR below/above the corresponding quartile. WT (6 cells pooled from two replicate experiments), PTRFKO + Cav1 (11 cells pooled from two replicate experiments) and PTRFKO + PTRF (10 cells pooled from two replicate experiments). Source numerical data are available in source data.

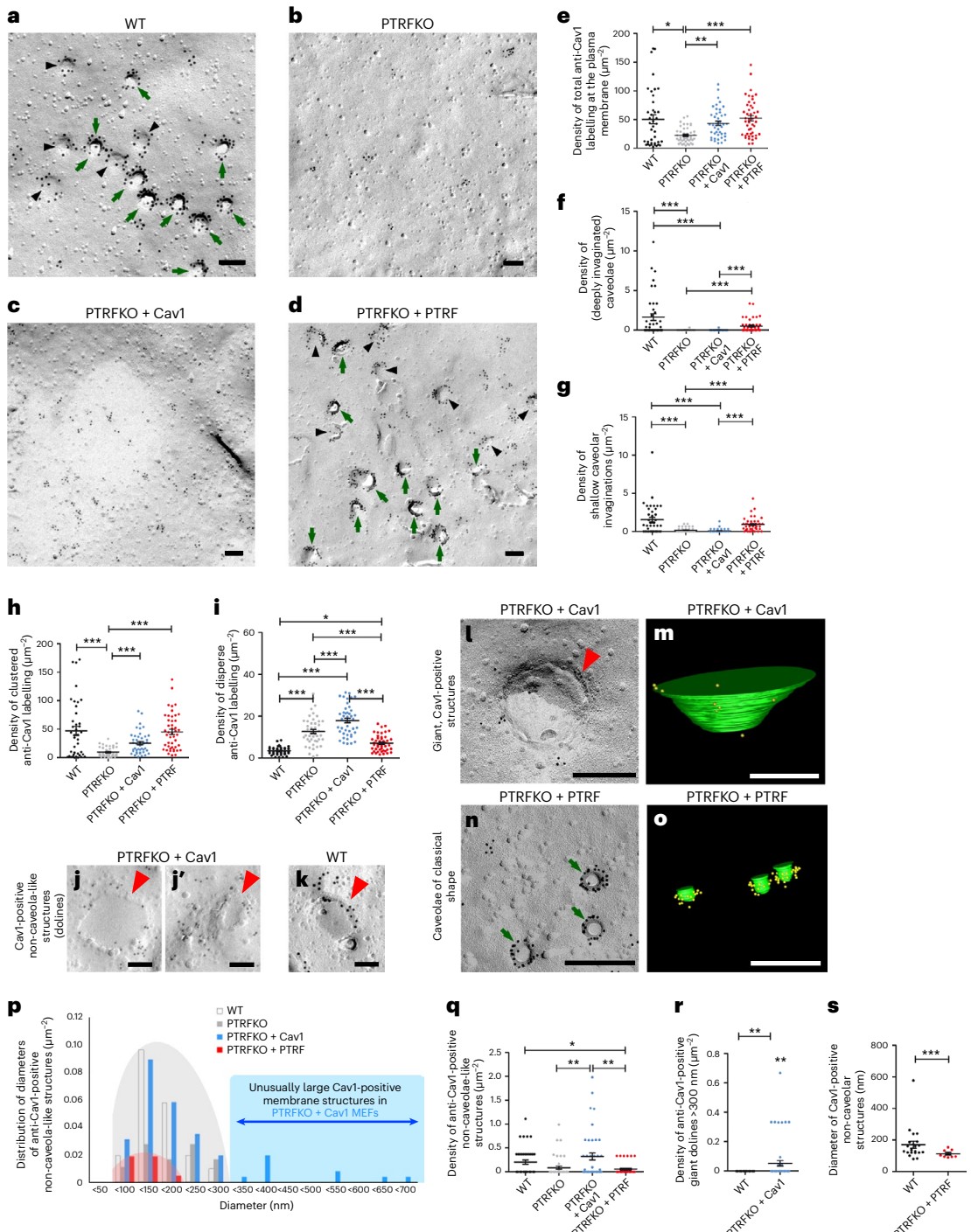

**Fig. 5 | PTRF genetic ablation leads to loss of caveolae and accumulation of non-caveolar Cav1 immunopositive structures. a–d**, Anti-Cav1 immunogold-labelled PMs of WT (**a**), PTRFKO (**b**), PTRFKO + Cav1 (**c**) and PTRFKO + PTRF MEFs (**d**). Caveolae (**a**, arrows) and anti-Cav1-positive shallow invaginations (**a**, arrowheads) were absent in **b** and **c**. Scale bars, 100 nm. **e**, Anti-Cav1 immunolabelling density across all four MEFs genotypes. **f,g**, Caveolae densities (**f**, deep; **g**, shallow). **h,i**, Densities of 'clustered' (**h**) and 'disperse' (**i**) anti-Cav1 immunogold labels. **j,k**, Cav1-immunopositive non-caveolar structures (red arrowheads) in PTRFKO + Cav1 MEFs (**j** and **j'**), morphologically distinct from caveolae, are also found (usually smaller) in WT cells (**k**). Scale bars, 100 nm. **l–o**, Electron tomograms (**l** and **n**) and 3D reconstructions (**m** and **o**) of a non-caveolar structure at the PM of PTRFKO + Cav1 cells (**l** and **m**), and 'classical' caveolae in PTRFKO + PTRF cells (**n** and **o**). Marks as in **a**, **j'** and **k**, respectively. In **m** and **o**, gold particles: yellow; PM: green. Scale bars, 200 nm. **p**, Density distribution of non-caveolar anti-Cav1-positive structures in PRTFKO + PTRF and WT cells (grey shadowed area; diameters up to 299 nm);

and in PTRFKO + Cav1 cells (blue shadowed area; diameters up to 300–699 nm). Red shadowing: PTRFKO + PTRF cells. **q–s**, Density analyses of all (**q**) and large (**r**, diameter ≥300 nm) non-caveolar Cav1-immunopositive structures; diameters in **s**. Plots: mean ± s.e.m.; WT control, $n = 40$ images per total ROI area 104.25 μm²; PTRFKO, $n = 46$ per 214.75 μm²; PTRFKO + Cav1, $n = 44$ per 258.4 μm²; PTRFKO + PTRF, $n = 46$ per 182.01 μm² from three independent experiments (**e–i**, **q** and **r**). For non-caveolar structure characterization: WT, $n = 22$; PTRFKO, $n = 18$; PTRFKO + Cav1, $n = 70$ and PTRFKO + PTRF, $n = 9$ (**p** and **s**). The technique does not allow to determine how many cells are analysed, as only patches of cell membrane are observed. Statistical analyses: Kruskal–Wallis test with Dunn's post-test (**e**, $P$ values: *0.048; **0.001; ***<0.0001; **f**, $P$ values: all ***<0.0001; **g**, $P$ values: all ***<0.0001; **h**, $P$ values: WT versus PTRFKO < 0.0001; PTRFKO versus PTRFKO + Cav1 0.008; PTRFKO versus PTRFKO + PTRF <0.0001), one-way ANOVA with Tukey post-test (**i**, $P$ values: all ***<0.0001; *0.0136), and two-tailed Student's $t$-test (**r**, $P = 0.0044$, **s**, $P = 0.0002$), respectively. Source numerical data are available in source data.

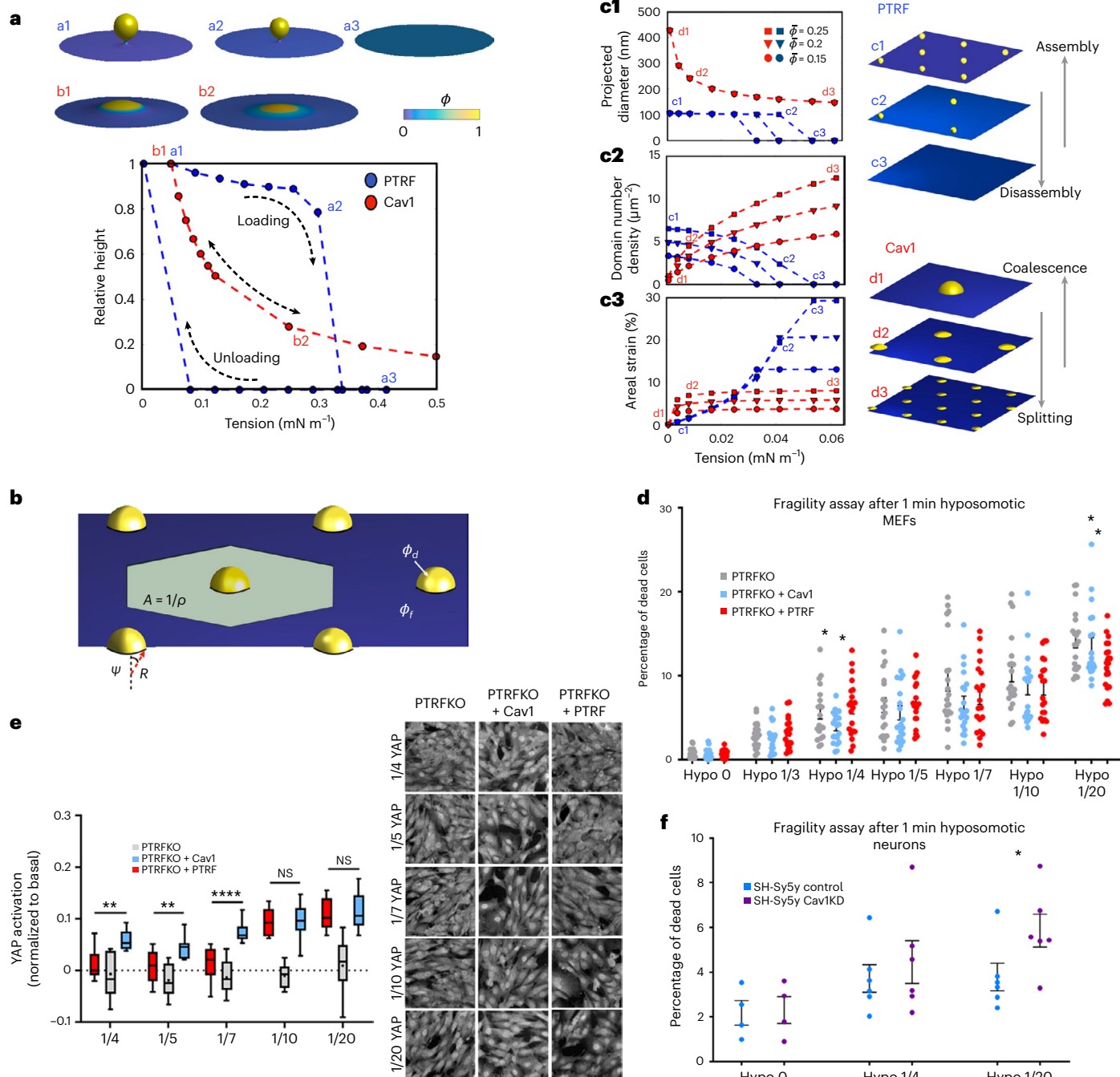

**Fig. 6 | Dolines and caveolae respond differently to membrane tension.**
**a**, Formation and tension-induced disassembly of protein-rich curved domains according to theoretical model; protein distribution: colour map depicting protein area fraction. Spontaneous curvature is smaller in Cav1 as compared with PTRF model, leading to shallow caps and full buds respectively when tension is low (a1 and b1). The Cav1 model exhibits gradual and reversible disassembly whereas for the PTRF model it is sudden and hysteretic. **b**, Theoretical model to understand the behaviour of an ensemble of such domains (Methods). **c**, Predictions of this model for the projected diameter of each domain, density of domains and projected areal strain of the membrane as a function of applied tension for three different average protein area fractions. Rightmost panels show representative states as tension varies for each model. **d**, Percentage of dead cells upon increasing medium dilution and hypo-osmotic shock (1 min fragility assay in adhesion) across indicated genotypes. Plots: mean ± s.e.m. $n = 20$ independent assays. Statistics: one-way ANOVA with Bonferroni post-test ($P = 0.0484$ at 1/4 dilution and $P = 0.0324$ at 1/20 dilution). **e**, YAP activation as

inferred from nuclear-to-perinuclear intensity ratio normalized to that averaged by untreated cells. Response to treatments is represented as deviation from 1 for each indicated treatment across genotypes, using nine independent wells per condition. Representative images for YAP immunostaining are shown on the right for indicated treatments and genotypes (for more details, see 'Statistics and reproducibility'). Boxes span Q1 to Q3 quartiles, with whiskers indicating lowest/greatest observed data point within 1.5× IQR below/above Q1/Q3. Middle line represents median, asterisks denote average value. **f**, Percentage of dead SH-Sy5y differentiated neurons under normal medium (hypo-osmotic 0), 1/4 or 1/20 dilution hypo-osmotic shock (1 min fragility assay in adhesion; for details, see Methods) comparing control with Cav1KD cells. Plots: mean ± s.e.m. $n = 6$ independent assays for 1/4 and 1/20 hypo-osmotic dilution, and $n = 4$ independent assays for hypo-osmotic 0. Statistics: two-tailed Student's $t$-test ($P = 0.0496$). Source numerical data are available in source data. Dataset from YAP experiments and script for YAP analysis are available at Zenodo (DOI: 10.5281/zenodo.7061911 and DOI: 10.5281/zenodo.7061924, respectively).

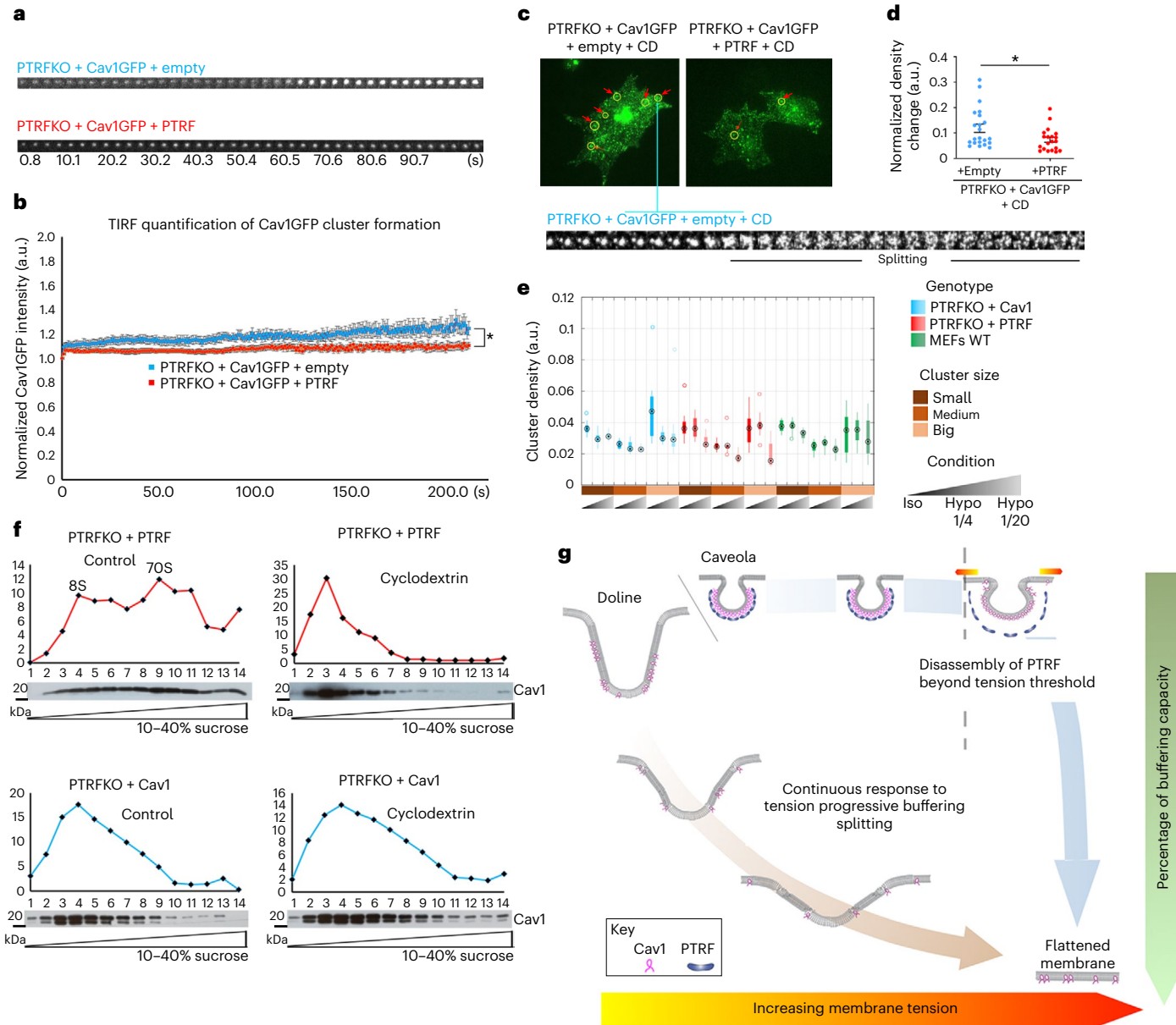

**Fig. 7 | Characterization of dolines and working model description.**
**a,b**, TIRF microscopy snapshots of Cav1GFP cluster formation (**a**), comparing
PTRFKO MEFs co-electroporated with either Cav1GFP and empty vector (Cav1
alone cells) or Cav1GFP and PTRF vectors (control cells) (**b**). Mean ± s.e.m.
*n* = 13 independent fields pooling together three cells (Cav1 alone); and *n* = 15
independent fields pooling together four cells (control). Two-tailed Student's
*t*-test (*P* = 0.0317). **c,d**, Maximal projection of TIRF microscopy image series
(from Supplementary Videos 13 and 14) of Cav1GFP clusters from Cav1 alone
(left) or control (right) PTRFKO MEFs (see **a** and **b**), after treatment with
methyl-β-cyclodextrin. Time lapse of a splitting cluster shown below.
In **d**, mean ± s.e.m. *n* = 22 independent experiments pooling together 22 cells
(Cav1 alone); and *n* = 20 independent experiments pooling together 20 cells
(control), after treatment with methyl-β-cyclodextrin. Two-tailed Student's *t*-test
(*P* = 0.0221). **e**, Cluster densities across genotypes and osmotic conditions from
dSTORM imaging, in three size groups. Data points outside the whiskers are
considered outliers and plotted as empty circles. Boxes span Q1 to Q3 quartiles,

with whiskers indicating lowest/greatest observed data point
within 1.5× IQR below/above Q1/Q3. Middle line represents median; asterisks
denote average value. **f**, Representative quantification of biochemical
fractionation of Cav1 complexes from PTRFKO + PTRF (control cells, top) and
PTRFKO + Cav1 MEFs (bottom) on 10–40% continuous sucrose gradients as
indicated in ref. 65 (blots against Cav1 below the corresponding quantification
graphs). Rightmost panels, cells treated with methyl-β-cyclodextrin. Results
are representative of three independent experiments. **g**, A working model for
Cav1-based PM tension buffering: Cav1 dolines provide gradual buffering to a
wide range of mechanical perturbations; in contrast, caveolae, as a result of PTRF
size restriction, constitute a mechanical switch that provides acute buffering to
higher forces, only flattening beyond a certain tension threshold
(high-range force sensor). Source numerical data and unprocessed blots
are available in source data. STORM images set are available at Zenodo
(DOI: 10.5281/zenodo.7062213).

forces by a series of culture medium dilutions (for details, see Meth-
ods), and quantified the percentage of dead cells after 1 min treat-
ment to infer early mechanoadaptation. Strikingly, under mild tension
increases (1:4 medium dilution) PTRFKO + Cav1 exhibit higher viability
than PTRFKO or control MEFs (Fig. 6d). This observation suggests that

membrane tensions within this range are less likely to induce cave-
olae flattening (that is not reaching the energy barrier required for
opening caveolae), whereas dolines could already provide mechani-
cal buffering (Fig. 6d). Conversely, higher membrane tension (1:20
medium dilution) led to decreased viability of PTRFKO + Cav1 MEFs as

compared with control MEFs; our model predicts a reduced net buffering capability of PTRF + Cav1 cells, exhausted earlier than that of control cells (Fig. 6d).

As a complementary readout, we studied the nuclear translocation of the mechanotransducer Yes-associated protein 1 (YAP; Methods and Extended Data Fig. 7j)[59]. Initially described as the main effector of the Hippo pathway, YAP undergoes nuclear translocation to regulate specific gene expression programmes upon mechanical stimuli[60]. Control cells assembling caveolae exhibited increased YAP nuclear translocation only when exposed to highest dilutions (1/10 and 1/20), capable of inducing robust PM tension changes (Fig. 6e, red boxes). In accordance with previous reports of YAP mechanoregulation by caveolae, PTRFKO cells exhibited a deficient response to PM tension changes (Fig. 6e, grey boxes). In contrast, PTRFKO + Cav1 cells exhibited significant increase in YAP nuclear translocation even at low osmotic forces, supporting our interpretation that dolines respond to force ranges below the threshold required for caveolae flattening (Fig. 6e, blue boxes).

To explore the role of Cav1 in the absence of caveolae in an unrelated context, we studied in vitro differentiated SH-Sy5y neuroblast cells[61], because neurons are a cell type that are physiologically devoid of caveolae[6,62–64]. Differentiated neurons were transduced with lentiviral vectors expressing either a non-targeting or Cav1-targeting short hairpin RNA, subjected to 1 min hypo-osmotic shock, and assessed for cell death rate. Cav1-deficient differentiated neurons showed reduced viability as compared with control cells, suggesting that Cav1 may play a mechanical role in neurons despite their virtual lack of caveolae (Fig. 6f).

To experimentally test the differential buffering behaviours predicted by the mathematical model (doline splitting versus caveola snapping), we first analysed de novo formation of Cav1 clusters by total internal reflection fluorescence (TIRF) microscopy by co-electroporating PTRFKO MEFs with either Cav1-EGFP and empty vector (Cav1 alone cells), or Cav1-EGFP and PTRF vectors (control cells), following a previously published protocol[65]. Interestingly, while control cells increased domain Cav1-EGFP intensity until a certain plateau was reached, owing to PTRF domain size restriction[65], domain signal intensity kept growing in cells expressing Cav1 alone (Fig. 7a,b and Supplementary Videos 11 and 12). This might indicate that, in the absence of PTRF, Cav1 domains can grow larger, forming the giant structures (dolines) we found by FRIL (Fig. 5m). We then treated cells with methyl-β-cyclodextrin to study the role of cholesterol in Cav1-EGFP cluster stabilization. Cav1-EGFP clusters started fragmenting onto smaller clusters (Fig. 7c and Supplementary Video 13) upon cholesterol removal (which is known to increase PM tension[66]). This was observed more frequently in cells expressing Cav1 alone, as compared with control cells (Fig. 7d and Supplementary Videos 13 and 14). These fragments might be constituted by 8S-like complexes, as suggested by the biochemical purification of Cav1 fractions on continuous sucrose gradients (Fig. 7f). These 'fragmentation' events are reminiscent of the splitting behaviour predicted by the mathematical model for Cav1-only domains (Fig. 6c3), and suggest that Cav1 clusters are sensitive to cholesterol levels.

As an independent complementary approach, we studied PTRFKO + Cav1, PTRFKO + PTRF and WT MEFs by dSTORM before and after subjecting cells to either mild (1/4 hypo-osmotic dilution) or high (1/20 hypo-osmotic dilution) forces. Whereas PTRFKO + Cav1 MEFs already showed reduction in cluster density after 1/4 dilution, control/WT MEFs only showed the same reduction after 1/20 dilution (Fig. 7e). Thus, Cav1 clusters formed in PTRFKO + Cav1 MEFs break into smaller ones (where Cav1 is less condensated) under mild tension increases, whereas Cav1 clusters in control or WT cells change only under high forces.

Our data thus support the existence of a broad continuum of Cav1-based buffering modalities (Fig. 7g), which constitutes a versatile mechanoadaptative system with potential consequences for YAP signalling and neuronal mechanoadaptation.

## Discussion

The role of caveolae in cell mechanics has been extensively studied[6,14,67,68]. Interestingly, Cav1—an essential caveolar protein component—can form scaffolds of different sizes in the absence of caveolae[27] and becomes sparsely distributed upon hypo-osmotic-induced disassembly of caveolae[14,69]. Cav1-dependent invaginations also preserve tissue integrity during ascidian *Ciona savignyi* embryogenesis[29]. Still, several questions remained: is the full caveolar structure required for mechanosensing and mechanoadaption? Do independent caveolar components, such as PM non-caveolar Cav1 clusters, have intrinsic buffering abilities in vertebrates? What is the role of PTRF in caveolae mechanoadaptation? Combining cell systems engineered to isolate the contribution of Cav1 from that of the caveolar structure and biophysical approaches, we observed that Cav1 confers protection against mechanical PM rupture, cell deformability and reduced stiffness in the absence of caveolae, by virtue of an intrinsic dose-dependent mechanical buffering ability for Cav1 independent from cytoskeletal dynamics (Fig. 3c). Context- or experimental-setting-dependent parameters known to affect mechanical properties, such as temperature[70] (which varied as required across different techniques; Methods) had no significant impact on them. Nonetheless, these contextual parameters should always be observed in future studies[70]. PTRFKO + Cav1 MEFs exhibit higher Cav1 cluster size heterogeneity, as compared with control cells (Fig. 4). Our ultrastructural studies revealed the ability of Cav1 to form invaginations of varying diameters in the absence of PTRF, which we have named dolines. Our comparison across genotypes shed light onto previous observations of invaginated structures in membrane fractions which do not contain PTRF, considered artefacts[71]. Thus, PTRF would instruct the assembly of Cav1-derived clusters and, consequently, the size of both dolines and caveolae, behaving as a switch between both structures.

Cav1 is predicted to bend the PM[42]. However, curved scaffolds containing ~15 Cav1 molecules are the smallest ones described in vitro[27]. Whether Cav1 curving ability depends on oligomerization is unknown. To study this possibility, we developed an in silico structural model for Cav1 (ref. [51]) (Extended Data Fig. 4r–u). Interestingly, the docking of a Cav1 dimer imposes a curvature angle of ~5 degrees, also apparent from the monomer structure alone (Extended Data Fig. 4r,s and Supplementary Videos 9 and 10). This arrangement includes the whole N-terminal domain of Cav1, missing in previous models[58,72], and further supports that the conformation of membrane-bound Cav1 monomers is sufficient to induce membrane curvature[73,74]. Dimer spacing allows for increased cholesterol condensation (Extended Data Fig. 4t,u), which potentially occurs upon caveolae deformation[75]. Cholesterol distribution and content affect membrane organization, regulating membrane–cytoskeleton adhesion[76] and PM mechanical properties;[77] also enriching PM cholesterol decreases membrane stiffness in response to pulling forces[77,78]. Cav1 regulates many aspects of PM cholesterol organization[79,80]. According to our FLIM studies, Cav1 increases cholesterol condensation in PTRFKO + Cav1 MEFs, which could contribute to membrane tension buffering. However, cholesterol decondensation in Cav1-derived invaginations may have a limited potential for PM tension buffering, (Supplementary Note 1)[52,81]. Cav1 organization into curved scaffolds may constitute the main buffering system, whereas cholesterol might be playing a role in domain stabilization and sensitivity (Fig. 7a–d). Cav1 clusters could thus establish different buffering mechanisms depending on whether they participate in caveolar assemblies or not. Our novel mathematical models predict that Cav1 cluster size is determined by both Cav1 average density and local changes in membrane tension, so that an inverse relationship between cluster size and membrane tension is established (that is, larger clusters form in regions of low membrane tension, and smaller clusters form in regions with high membrane tension). Interestingly, recent studies support that PM tension is not homogeneous, and PM subdomains subject to different local tension values can be observed[82]. These differences in local tension could account for the size heterogeneity

observed in Cav1 dolines of PTRFKO + Cav1 cells (Figs. 4 and 5). Cav1 clusters reform if tension stands constant below a maximum limit; indeed, membrane order was progressively recovered over time in control cells after sustained mechanical stretching (Extended Data Fig. 6a,b). These results suggest that the size of the Cav1 dolines is tightly coupled to membrane tension.

Our model also predicted that tension required for flattening caveolae is much higher than that required for Cav1 dolines. This prediction is consistent with our fragility assay testing a wide range of hypo-osmotic conditions (that is, membrane tension values): Cav1 dolines readily buffer for mild tension increases (Fig. 6d). Caveolae would then behave as mechanical switches that do not release area below a threshold tension and abruptly release it above this threshold, whereas dolines continuously release their area as tension increases, much like springs would do. Interestingly, these differential dynamics clarify previous controversies regarding the buffering ability of caveolae at very short timescales and low tension changes[4,83]: dolines could provide such mechanoadaption. Another important prediction of the model is Cav1 domain splitting in response to membrane tension, supported by our dSTORM analysis of Cav1 clusters under hypo-osmotic medium and TIRF analysis after cyclodextrin treatment leading to cluster disassembly (Fig. 7a–e), as cholesterol removal increases PM tension[66]. Biochemistry studies suggest that dolines might assemble from lighter (8S) Cav1 particles distinct from the heavier 70S aggregates typically observed in cells competent for caveolae formation (Fig. 7f). Interestingly, dolines—considered as 8S complexes clusters—are not further affected by CD treatment, suggesting they remain as biochemically resistant smaller 8S domains. These small clusters could well represent the newly discovered Cav1 discs[72]. It is tempting to speculate that splitting could represent a coding mechanism to finely adjust physical quantum into biochemical quantum. Accordingly, our observations indicate this has an impact on downstream mechanotransduction. Cells with dolines exhibit YAP nuclear translocation when exposed to a wide range of PM tension changes, whereas cells efficiently assembling caveolae exhibit increased YAP activation only beyond a certain force threshold (Fig. 6e): a novel layer for YAP signalling regulation, whereby the relative proportion of dolines and caveolae would fine-tune the sensitivity of this mechanoadaptive network. Importantly, a substantial share of physiological processes and environments entail forces below caveolae flattening threshold[20–24]. Ligand-independent integrin activation is known to occur in response to changes in PM tension;[84,85] thus, modulating the proportion of dolines (sensitive to low forces) versus 'classic' caveolae (sensing high forces) could be a means to fine-tune mechanically driven integrin signalling, explaining the association between invasive phenotypes and PTRF depletion displayed by prostate cancer cells[86,87]. This conceptual framework also invites to re-evaluate the 'stiffness-independent growth' concept[88], and the 'stiffness-sensing' loss[89] displayed by many tumour cell types, which, rather than lacking rigidity sensing, might develop higher sensitivity to low forces. This could be especially relevant for those tumour cell types that have been already shown to have less surface caveolae[90], as they could potentially present more dolines. Finally, Cav1 dolines might be particularly relevant to understand mechanosensing and mechanoprotection in cells and tissues that physiologically express very low levels of Cav1 and are virtually devoid of caveolae (that is, hepatocytes, lymphocytes and neurons[25,26,91]; Fig. 6f). The existence of these PM structures at cell regions devoid of caveolae also provides a specific potential mechanism by which mechanosensing can be organized at subcellular scales to respond to different force ranges[92]. In this model, PTRF would constitute a gate onto which regulatory inputs would converge to modulate the relative density of each Cav1-dependent invagination type, fine-tuning force sensing dynamics in the cell. Thus, PTRF would behave as a switch between the types of membrane tension buffering mechanism provided by dolines versus caveolae.

Our data support a model whereby Cav1 forms a variety of domains with different sizes and different buffering abilities constituting, together with caveolae, a versatile mechanoadaptation system, specifically contributing responsiveness to mild mechanical forces (Fig. 7g). Once caveolae and Cav1 scaffolds are completely flattened, cells will depend on long-range mechanoadaptation systems such as cortical actin remodelling[93,94], if membrane tension is further increased. Our work will hopefully lead to future studies to unravel the different levels of complexity of cellular mechanoadaptation.

## Online content

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

[1]Mechanoadaptation and Caveolae Biology lab, Cell and Developmental Biology Area, Centro Nacional de Investigaciones Cardiovasculares (CNIC), Madrid, Spain. [2]Cell and Tissue Mechanobiology Research Programme; LaCàn, UPC-BarcelonaTech, Barcelona, Spain. [3]Institute of Biochemistry I, Jena University Hospital - Friedrich Schiller University Jena, Jena, Germany. [4]Institute for Bioengineering of Catalonia (IBEC), the Barcelona Institute of Technology (BIST), Barcelona, Spain. [5]PreciPoint GmbH, Freising, Germany. [6]Biotechnologisches Zentrum (Biotec), Technische Universität Dresden, Dresden, Germany. [7]Microscopy and Dynamic Imaging Unit, Centro Nacional de Investigaciones Cardiovasculares (CNIC), Madrid, Spain. [8]Institut Curie, PSL Research University, INSERM U1143, CNRS UMR 3666, Membrane Mechanics and Dynamics of Intracellular Signaling Laboratory, Paris, France. [9]Bioinformatics Unit, Centro Nacional de Investigaciones Cardiovasculares (CNIC), Madrid, Spain. [10]Centro de Investigación Biomédica en Red de Enfermedades Cardiovasculares (CIBERCV), Madrid, Spain. [11]Unitat de Biofísica i Bioenginyeria, Facultad de Medicina, Universitat de Barcelona, Barcelona, Spain. [12]Max Planck Institute for the Science of light and Max-Planck-Zentrum für Physik und Medizin, Erlangen, Germany. [13]Departamento de Polímeros; Facultad de Ciencias Químicas, Universidad de Concepción, Concepción, Chile. [14]Center of Experimental Imaging, Ospedale San Raffaele, Milan, Italy. [15]Institució Catalana de Recerca i Estudis Avançats (ICREA), Barcelona, Spain. [16]Centro de Investigación Biomédica en Red en Bioingeniería, Biomateriales y Nanomedicina (CIBER–BBN), Barcelona, Spain. [17]Present address: Allergy Therapeutics S.L. (Parque Tecnológico Tecnoalcala), Alcalá de Henares, Spain. [18]These authors contributed equally: Nikhil Walani, Eric Seemann. ✉e-mail: flolo@cnic.es; madelpozo@cnic.es

## Methods

### Cells culturing, cloning, retroviral transduction and reagents

PTRFKO MEFs were a kind gift from Prof. Rob Parton (Institute for Molecular Biosciences, Queensland, Australia). A complementary DNA encoding a Cav1-FLAG fusion was excised from pCDNA3.1 Cav1 wt vector with BamH1/EcoR1, blunted with Klenow fragment, and ligated into the retroviral vector MIGR1 (Addgene ref. #27490, which also expresses EGFP from an IRES) cut with EcoRI and Klenow fragment blunted[95]. A cDNA encoding for PTRF was excised from pIRES2-cavin1 EGFP (a kind gift from Prof. Rob Parton, Institute for Molecular Biosciences, Queensland, Australia) with BglII/BamH1 and ligated into BglII-digested MIGR1. Retroviral particles were produced in 293T-Phoenix cells and used for MEF transduction according to standard protocols. SH-Sy5y cells were a kind gift from Dr Sergio Casas Tintó (Cajal Institute, Madrid, Spain) and differentiated into neurons as previously indicated[96], and then transduced with lentiviral vectors expressing either Cav1-targeting short hairpin RNA or a scrambled sequence[68]. Transduced cells were purified to required marker intensity and homogeneity as batch cell cultures (Cellomics Unit, CNIC). All cells were cultured at 37 °C and 5% $CO_2$ in Dulbecco's modified Eagle medium (DMEM; Thermo Fisher Scientific) supplemented with 10% foetal bovine serum and 1% penicillin and streptomycin, and routinely checked for mycoplasma contamination.

The following primary antibodies were used: rat monoclonal anti-mouse total beta 1 integrin (clone MB1.2, MAB1997 Millipore, 1:1,000); rabbit monoclonal anti-mouse Cav1 (Cell Signaling, 1:1,000 for western blot and 1:100 for immunofluorescence); rabbit polyclonal anti-mouse PTRF (ab48824, Abcam, 1:1,000 for western blot and 1:100 for immunofluorescence); mouse monoclonal anti-alpha tubulin (ab7291, Abcam, 1:10,000); anti-Cav1 N-20 antibody (Santa Cruz Sc-894); Cav1 SIGMA SAB4200216 (mouse monoclonal); ubiquitin (Enzo, ENZ-ABS840-0100); mouse monoclonal anti-GAPDH (sc-47724, Santa Cruz, 1:1,000); and mouse anti-GFP (catalogue number 118114460001, Roche, 1:1,000). The following secondary antibodies were used: Alexa Fluor–488 goat anti-mouse (Thermo Scientific, 1:100); Alexa Fluor-488 goat anti-rabbit (Thermo Scientific, 1:100); Alexa Fluor-647 goat anti-rabbit (Thermo Scientific, 1:100).

### EM of sections of chemically fixed cells

MEFs were processed for EM using standard procedures. Briefly, cells were fixed for 1 h with 2.5% glutaraldehyde in 100 mM cacodylate buffer, pH 7.4, and then post-fixed for 3 h with 1% osmium tetroxide in 100 mM cacodylate buffer, pH 7.4. The samples were dehydrated with acetone, embedded in Epon, sectioned and stained. Ruthenium red (1 mg ml$^{-1}$) was added during fixing and post-fixing to decorate PM.

### Hypo-osmotic treatment

**In suspension.** For studying mechanoprotection from hypo-osmotic swelling, $5 \times 10^5$ MEFs of the indicated genotypes were seeded on p6 plates for 24 h. Then, they were washed twice with PBS 1×, trypsinized and resuspended in diluted DMEM (1:10) with MilliQ water. After 1 min, cells were centrifuged, stained with Trypan Blue (Sigma) and counted in a Neubauer chamber.

**In adhesion.** A total of $5 \times 10^5$ MEFs of the indicated genotypes were seeded on p6 plates for 24 h, and then washed twice with PBS 1×, and treated with different dilutions of DMEM (0, 1:3, 1:4, 1:5, 1:7, 1:10 and 1:20 in MilliQ water). After 1 min, hypo-osmotic medium was removed, and cells were trypsinized, centrifuged, stained with Trypan Blue (Sigma) and counted in a Neubauer chamber.

**YAP image analysis.** Assays for YAP subcellular distribution were conducted on an Opera HCS II automated spinning confocal station as follows. Cells were plated on 384-well Cell Carrier optical plates at 5,000 cells per well on 30 μl of complete growth medium. Twenty-four

hours later, cells were subjected to the indicated hypo-osmotic shock treatments for 1 min, and immediately fixed by directly adding an equal volume of 8% paraformaldehyde. Cells were processed for immunostaining using standard procedures. Images were acquired for nuclear DNA content (Hoechst 33342), retroviral GFP reporter signal, Cav1 immunostaining (rabbit monoclonal anti-mouse Cav1, D46G3 Cell Signaling, at 1:100 dilution; secondary antibody was Alexa Fluor-568 goat anti-rabbit, Thermo Scientific, at 1:100 dilution) and YAP immunostaining (mouse monoclonal anti-YAP, 63.7, sc-101199 Santa Cruz, at 1:400 dilution; secondary antibody was Alexa Fluor-647 goat anti-mouse, Thermo Scientific, at 1:100 dilution). Images were then analysed using the Acapella studio environment, with the following workflow (Extended Data Fig. 7j): nuclear detection, cytoplasm segmentation, delimitation of nuclear, perinuclear (four-pixel ring of cytoplasm grown radially from the segmented nuclear border) and membrane regions with boundaries defined as percentage distance from membrane to nucleus, and measurement of intensity for each channel and cell morphometric parameters. Mitotic and aberrant nuclei were filtered out, on the basis of Hoechst intensity and nuclear roundness and area. Further, to minimize effects from differential local confluency or cell spreading, cell subpopulations with analogous (mean ± standard deviation) values of cell area, width-to-length ratio and neighbour fraction (a proxy for cell confluency) to those displayed by PTRFKO + Cav1 cells were selected for all genotypes. The nuclear-to-perinuclear intensity ratio for the YAP immunostaining channel was normalized to that averaged by untreated cells. Response to treatments is represented as the deviation from 1 for each indicated treatment across genotypes.

### OS

**Principle and setup description.** A microfluidic version of the Optical Stretcher was used to investigate mechanical deformation of cells upon optical stress. Briefly, the Optical Stretcher is a dual beam laser trap capable of trapping and deforming cells through optically induced stress, acting on the cell surface. Two optical fibres placed co-axially, pointing at each other, are aligned perpendicular to a square glass capillary (Fig. 2a, top). Single cells in suspension are delivered into the trapping region through the glass capillary. The flow into the glass capillary is adjusted by the relative difference in heights of an inlet and outlet reservoir connected to the capillary. Detailed descriptions of the Optical Stretcher working principle and setup can be found elsewhere[38,97–99].

The device was mounted on an inverted microscope (Zeiss Axiovert 200 M) equipped with a LD Plan-NEOFLUAR Ph2 40×/0.60 numerical aperture (NA) objective. A camera (AVT MARLIN F-146B; Allied Vision) was attached to the microscope for image acquisition. The laser used was a single-mode, continuous-wave fibre laser at a wavelength of $\lambda = 1,064$ nm (YLM-5-1070-LP; IPG Photonics). For data acquisition and analysis, custom-built LABVIEW software (National Instruments) was used to track cell shape during stretching.

**Sample preparation.** For OS experiments, cells were detached from their flask and transferred in suspension to PBS. Cells were kept for about 30 min in PBS before launching the experiment, to allow stabilization. Experiments were performed at room temperature.

**Measurement.** Cells were exposed to optical stress for 10 s. Cell deformation along the major axis $r(t)$ was recorded for every timeframe, while $r_0$ was the measured length of the cell during the initial trapping period. The time-varying axial strain,

$$\text{Strain}(t) = \frac{r(t) - r_0}{r_0},$$

was then evaluated accordingly.

## AFM

**Cell culture.** MEFs were incubated under standard culture conditions (37 °C, 5% $CO_2$). DMEM growth medium supplemented with 10% foetal bovine serum, 2 mM L-glutamine (Gibco) and 1% penicillin and streptomycin were used and replaced every 3 days until confluence. For each group, $8 \times 10^4$ cells were seeded in 22.1 $cm^2$ surface area Petri dishes and maintained for 24 h in similar culture conditions.

**AFM measurements.** Cell mechanics was measured with a custom-built atomic force microscope coupled to an optical inverted microscope (TE2000, Nikon, Japan) by using previously described methods[39,100]. Cells were probed at room temperature using a microsphere (4.5 μm in diameter) attached to a V-shaped gold-coated silicon nitride cantilever of nominal spring constant $k = 0.03$ N $m^{-1}$ (Novascan Technologies). The actual spring constant of the cantilever was calibrated by means of the thermal fluctuations method. The cantilever was displaced in 3D with nanometric resolution with piezoactuators coupled to strain gauge sensors to measure cantilever displacement ($z$). The deflection of the cantilever ($d$) was measured with the optical lever method. The sensitivity of the optical lever was calibrated by recording a deflection–displacement ($d$–$z$) curve in a bare region of the glass slide. A linear calibration curve with a sharp contact point was taken as indicative of a clean and undamaged tip. The force applied by the cantilever was computed as $F = k \times d$. The indentation ($\delta$) of the sample was computed as $\delta = (z - z_c) - (d - d_o)$, with $z_c$ being the displacement of the cantilever at the tip-cell contact point and $d_o$ the cantilever deflection offset. Force–displacement curves were recorded at mid-distance between nucleus and cell edge in three culture samples of each cell type (15 cells measured per sample) with triangular displacement of the cantilever (3 μm amplitude, 1 Hz, maximum indentation ~1 μm). Force–indentation data were analysed with the spherical Hertz model[39,100]:

$$F = \frac{4ER^{1/2}}{3(1 - \mu^2)}\delta^{3/2}$$

where $E$ is the Young's modulus and $\mu$ is the Poisson's ratio (assumed to be 0.5). A non-linear least-squares fit was used to compute $E$ (MATLAB, The MathWorks). The stiffness of each culture sample was characterized by the average of the $E$ values obtained in the 15 cells probed in the sample.

**Statistical analysis.** Differences among groups were evaluated using one-way analysis of variance (ANOVA) and Holm–Sidak post-hoc pairwise multiple comparison test ($n = 3$). P values <0.05 were considered statistically significant.

## Magnetic tweezers and reinforcement measurements

**Bead coating.** Carboxylated magnetic beads (Invitrogen) were mixed in a solution containing 500 μl 0.01 M sodium acetate (pH 5), 0.75 mg Avidin (Invitrogen) and 4 mg EDAC (Sigma). Beads were incubated for 2 h at room temperature and then washed in PBS and further incubated for 30 min in 1 ml 50 mM ethanolamine (Polysciences). The beads were then washed three times in PBS and left in PBS on a cold room rotator.

**Force measurements.** Magnetic tweezers experiments were performed as described[101,102]. Briefly, carboxylated 3 μm magnetic beads (Invitrogen) were coated with biotinylated pentameric FN7-10 or ConA (Sigma-Aldrich) mixed 1:1 with biotinylated BSA. For measurements, cells were first plated on coverslips coated with 10 μg $ml^{-1}$ FN (Sigma) in Ringer's solution (150 mM NaCl, 5 mM KCl, 1 mM $CaCl_2$, 1 mM $MgCl_2$, 20 mM HEPES and 2 g $l^{-1}$ glucose, pH 7.4). FN-coated beads were then deposited on the coverslips and allowed to attach to the cells. The tip of the magnetic tweezers device was then used to apply a force of 1 nN for 2 or 3 min to beads attached to cell lamellipodia. The apparatus used to apply force to the magnetic beads was as previously described[103].

The system was then mounted on a motorized 37 °C stage on a Nikon fluorescence microscope. Differential interference contrast (DIC) images and videos were recorded with a 60× objective linked to a charge-coupled device (CCD) camera at a frequency of 250 frames $s^{-1}$.

## Adhesion assay

Cell adhesiveness was assessed by seeding MEFs on 96-well plates coated with FN or ConA (both at 5 μg $ml^{-1}$) and incubating at 37 °C for 30 min. Wells with no coating were included as negative controls. Cells were then fixed with methanol and stained with crystal violet (Sigma-Aldrich). Wells were washed thoroughly to remove excess dye and were finally eluted with a mixture of 50% ethanol and 50% 0.1 M sodium citrate (pH 4.2). Absorbance was read at 595 nm.

## Nanotube pulling experiments with OTs

**Force measurements.** PM tethers were extracted from cells by a ConA (Sigma-Aldrich) coated bead (3 μm in diameter, Polysciences) trapped in OTs. A custom-built OT setup coupled to an inverted Nikon C1 Plus confocal microscope (Tokyo, Japan) was used for pulling PM nanotubes, as described previously[104]. Briefly, a 1,064 nm continuous wave Ytterbium fibre laser (IPG Photonics) set to a 3 W input power was modulated to 400 mW (measured at the back aperture of the objective) using a polarizing beam splitter (Thorlabs), expanded through a telescope consisting of two plano-convex lenses with focal lengths of 100 mm and 150 mm (Thorlabs), and directed towards the back aperture of a Nikon CFI Plan Apochromat Lambda 100× 1.45 NA oil immersion objective (Tokyo, Japan). Displacements of a trapped bead from the fixed trap centre were recorded using an Allied Vision Marlin F-046B CCD camera at a frame rate of 20 frames $s^{-1}$, and later analysed by a custom ImageJ macro. As the optical trap itself was stationary, all relative movements were performed using a piezo-driven stage (Nano-LP100, MadCityLabs). Atop the stage, a temperature- and $CO_2$-controllable Tokai Hit STXG-WELSX stage-top incubator was attached, allowing cells to be maintained at 37 °C in a humidified, 5% $CO_2$ atmosphere during experimentation. The membrane tether was held at constant length to measure the static force. For measuring membrane tension changes due to hypo-osmotic shock, a second tether was pulled after 5 or 10 min after the medium was diluted until the osmolarity reached the indicated values. The position of the beads used to compute tether forces was detected from the images using a custom ImageJ macro.

**Statistical analysis.** Analyses were performed using GraphPad Prism version 7.0 for Mac OS X, GraphPad Software, La Jolla California USA, (www.graphpad.com).

## Total internal reflection fluorescent microscopy videos

Analysis of de novo formation of Cav1GFP clusters was performed as previously described[65]. TIRF microscopy was performed with a Leica AM TIRF MC microscope. TIRFm movies were acquired with a 100× 1.46 NA oil-immersion objective at 488 nm excitation and an evanescent field with a nominal penetration depth of 110 nm. Images were collected with an ANDOR iXon CCD at 840 ms per frame. Cav1GFP spots were analysed with TrackMATE plugin (ImageJ) to obtain mean fluorescence intensity over frames. Graphs represent normalized fluorescence intensity of Cav1 GFP signal over frames. For splitting analysis, particles were analysed by finding and counting local maxima using LoG 3D[105] plugin (ImageJ). C-terminal-tagged Cav1-GFP was a kind gift from Prof. Marie-Odile Parat[106] (The University of Queensland, Australia). FLAG-tagged PTRF was cloned into pCMV-myc and pCMV-myc empty was used as control (Addgene ref. #631604).

## dSTORM

Samples were immunostained with a rabbit monoclonal anti-mouse cav1 (1:100) antibody and Alexa Fluor®647-Fab1 fragment goat anti-rabbit (Jackson Immunoresearch; 1:10). dSTORM images were

acquired on a Leica SR GSD system (Leica Microsystems) equipped with an HC PL APO 160×/1.43 oil CORR GSD objective and an EMCCD back-illuminated camera (Andor iXON Ultra DU897). The field of view was 19.8 × 19.8 µm at high-power mode. A continuous wave fibre laser (MPBC, 642 nm, 500 mW) and a diode laser (405 nm, 30 mW) were used. Fluorescence emission was filtered through a quadruple filter (excitation: 400–410, 483–493, 527–537, 637–647; dichroic: 417, 496, 544, 655; emission: 421–477, 497–519, 547–621, 666–732 (all in nm)). The objective was linked to the sample with help of a suppressed-motion sample stage to minimize drifts. Samples were first illuminated at 642 nm and 100% power, and acquisition was started manually after observing blinking. The electron multiplying gain of the camera was set at 300. The laser power during the acquisition was 50–70% depending on sample, and it was chosen to ensure that the fraction of activated fluorophores at any given time would be sufficiently low to enable recognition of single blinking. Typically, we recorded 9,000–10,000 frames at rates of 9.194 ms per frame. Data were acquired and processed using LAS AF V 4.0.0. 11706 software (Leica Microsystems).

**Data processing.** Frame sequences were background subtracted using the rolling ball method (Sternberg SR (1983) Biomedical Image Processing. Computer 16: 22–34) before the standard localization routine by 'direct fit' fitting method. Positive intensity peaks with at least one pixel above a minimum threshold were fitted to a two-dimensional Gaussian to determine the $x$ and $y$ coordinates, amplitude, $1/e^2$ radius and offset of each point spread function. To reduce the number of localizations of the same fluorophore and improve localization precision, data were processed by averaging the coordinates of consecutive events within a radius of 20 nm around each localization. For cluster analysis by ImageJ-Fiji, a Gaussian filter of 0.5 radius was applied to the localization images. To determine cell density, regions of interest (ROIs) of typically 200–300 µm² were segmented on the basis of the epifluorescence image acquired before the blinking sequence. The number of clusters was determined by the standard ImajeJ-Fiji routine after automatic thresholding, and the Feret diameter, the measure of the longest distance between any two points along the selection boundary of each cluster above threshold, was used as 2D shape descriptor for the cluster size[107–111]. Clusters touching the ROI borders were excluded, and cluster density and frequency distributions were obtained by GraphPad. Further details about cluster analysis after different hypo-osmotic treatments can be found in supplementary information.

## dSTORM for cluster analysis
**Data processing.** Analysis of dSTORM data (sample processing and imaging acquisition were performed as indicated above) was performed working with coordinate maps of blinks directly, instead of using images reconstructed from that data, as other authors do in their analysis and applications[112,113]. To create the coordinate map from dSTORM videos, we used ThunderSTORM v1.3 (Release Version 1.3·zitmen/thunderstorm· GitHub) for ImageJ v1.53q software, performing drift correction and filtering of localizations using the following parameters: (intensity >300) & (intensity <5,000) & (uncertainty <35) & (sigma <300).

**Global density-based homogenization of STORM data.** To prevent the large variability of number of blinks, frames and blink densities from hindering proper analysis and comparison between samples, a density-based homogenization strategy was used to get coordinate maps with similar global density of blinks for all samples (Extended Data Fig. 7h). First, cellular area was delineated in each sample using ImageJ. Only blinks from the first $k$ frames of a dSTORM video were used, where $k$ was selected to better approximate a target global density of blinks in the corresponding segmented area. This global target density of blinks was fixed at $2 \times 10^{-4}$ blinks nm$^{-2}$, enough for not discarding samples with low number of blinks that failed to approach the target density.

**Clustering and measurements.** Previous homogenization allowed us to use density-based clustering algorithms to determine spatial groupings of blinks in a fair and unbiased manner. DBSCAN[114] algorithm was selected for that purpose, since it does not require to specify the number of desired clusters, does not make any assumption about cluster shapes and is robust to noise (an important component in this image modality). Therefore, DBSCAN was used to group neighbouring blinks (closer than $\varepsilon = 20$ nm) and clean noisy blinks by detecting outliers (minimum number of blinks in a group to be considered an actual cluster, minpts = 30). Resulting clusters are highly probable aggregations of Cav1 molecules that may represent caveolar (sub) structures. Owing to the initial homogenization of global densities in all samples, computation and comparison of blink densities in each cluster provide a valuable readout about the degree of aggregation of Cav1 molecules. The density of each cluster was measured as the number of blinks divided by the area enclosed by its boundary blinks. As densities may vary depending on the size of the clusters found, we stratified the structures in three groups depending on their enclosed area: small (area <25²·π nm²), medium (25²·π nm² ≤ area < 50²·π nm²), and large (area ≥50²·π nm²). Median density of blinks among all clusters in each group and sample was reported and compared between conditions (an example can be found in Extended Data Fig. 7i).

## Freeze-fracturing and immunolabelling
**Experimental conditions.** Freeze-fracturing of MEFs was essentially done as described before[18,54,115]. Briefly, MEFs were collected and then quick-frozen in between a copper head sandwich profile with a liquid-nitrogen-cooled ethane/propane mixture and a cooling rate of >4,000 K s$^{-1}$. The sandwiches were then subjected to freeze-fracturing in a BAF400T freeze-fracture unit (Balzer). Fractured membranes were carbon- and platinum-coated, extracted with 2.5% (w/v) SDS (in 30 mM sucrose and 10 mM Tris–HCl pH 8.4) overnight, washed, blocked with 1% (w/v) BSA, 0.5% (w/v) gelatine and 0.0005% (v/v) Tween20 in PBS and incubated with rabbit anti-caveolin-1 antibodies (sc-894; dilution 1:50, 4 °C, overnight) as well as with anti-rabbit 10 nm gold-conjugated secondary antibodies (2 h; room temperature).

Anti-Flag immunolabelling of PTRFKO + Cav1 MEFs and PTRFKO cells not expressing Flag-tagged Cav1 were performed using related replica cleaning, extraction and blocking procedures. Anti-Flag immunolabelling was performed overnight (M2; dilution 1:2,000, 4 °C) followed by incubations with anti-mouse 10 nm gold-conjugated secondary antibodies for 2 h at room temperature. Opposing membrane faces (E-face) and ice surfaces additionally served as intrinsic control surfaces for incubations with both primary and secondary antibodies (for quantitation of P-face versus E-face labellings, see Extended Data Fig. 7a-d).

**Antibodies.** Polyclonal rabbit anti-Cav1 (sc-894) was from Santa Cruz. Monoclonal anti-Flag antibodies (M2) were from Sigma. Gold-labelled goat anti-rabbit (10 nm) and gold-labelled goat anti-mouse (10 nm) secondary antibodies were from British Biocell International.

**EM.** Replica of freeze-fractured and immunolabelled MEFs were collected on uncoated copper grids (300 mesh) and analysed with a transmission electron microscope operated at 80 keV (EM902A, Zeiss). Imaging was done by systematic explorations of the grids. Images were recorded with a CCD camera (TVIPS; EM-Menu 4 and Tröndle Wide-angle Dual Speed 2K), processed with Adobe Photoshop and quantitatively evaluated with ImageJ.

**Electron tomography of freeze-fractured caveolae.** For electron tomography, replica specimens were placed in a tilt-rotate specimen holder (Model 626; Gatan). Tomographic datasets were recorded using a Philips CM120 operated at 120 kV. Images were captured with decreasing increment from −56° to 0° and with an increasing increment

from 0° to 52° range. A 2K CCD camera (TecCam F216, TVIPS) was used for image recordings.

The positions of the gold particles were used as fiducial points for alignments of tilted views. Electron tomograms were computed and segmented using the software IMOD package[116].

**Quantitative analyses of caveolar and non-caveolar invaginations.** Quantitative evaluations were performed using samples from several independent freeze-fracturing experiments. Caveolar invaginations were categorized as deeply invaginated (about 70 nm inner diameter; round; so deeply invaginated that bottom is not fully reached by platinum shadowing) and shallow caveolae (usually 70–90 nm in diameter, so shallow that the full bottom is reached by platinum shadowing), respectively. Efficient anti-Cav1 immunolabelling was used to confirm the caveolar nature of both deep and shallow caveolar invaginations.

In contrast, membrane structures with non-regular appearance and with partially much extended inner membrane surfaces completely devoid of any integral transmembrane proteins that were marked by anti-Cav1 immunogold labelling were scored as Cav1-positive, yet non-caveola-like invaginations. All three types of membrane topology analysed were determined as densities per full image (2.47 µm² and 3.03 µm² each, respectively) and analysable membrane ROI of an image, respectively. Anti-Cav1 labelling densities also were determined per image and analysable ROI, respectively. Immunolabels were only considered as localized to a caveolar invagination if localized <50 nm from the (inner) caveolar rim. In total, 759.5 µm² membrane were scored for quantitative analyses and 674 Cav1-positive invaginations of different types were evaluated and scored. Cluster analyses of anti-Cav1 immunogold signals were done according to a procedure established previously[18]. Circular ROIs of 150 nm diameter around caveolae were used. This cut-off reflects the 70 nm of inner caveolar diameter and additional zones that need to be considered (2 ×10 nm for the PM curvature zone around the caveolae and 2 × 30 nm for the maximally possible extension of primary/secondary antibody and gold particle). Four or more immunosignals per ROI were considered as Cav1 cluster.

**Statistical analyses.** No statistical methods were used to pre-determine sample size. All quantitative data shown represent mean and standard error of the mean (s.e.m.). No outlier suggestions were computed. No strongly scattering data points were excluded, but all quantitative evaluation data points were taken into account and averaged to fully represent biological and technical variabilities.

Statistical analyses were done using GraphPad Prism software. Statistical significances were marked by *P < 0.05, **P < 0.01 and ***P < 0.001 throughout.

## PM fractionation and western blot analysis

MEFs were processed for PM isolation as described[117]. Cells were first washed and pelleted by centrifugation at 14,000g for 5 min. Cells were then manually homogenized with 20 strokes of a PTFE head Tissue homogenizer (VWR) and centrifuged at 1,000g for 10 min. The post-nuclear supernatant was collected and layered on top of a 30% Percoll column. After centrifugation of the Percoll column at 84,000g for 30 min, the PM fraction was collected, separated by SDS–PAGE and analysed by western blot. Samples were immunoblotted with rabbit monoclonal anti-mouse cav1 and rat monoclonal anti-mouse total beta 1 integrin (clone MB1.2, MAB1997 Millipore) as a loading control. Total cell lysates were separated by SDS–PAGE and analysed by western blot with rabbit monoclonal anti-mouse Cav1 (Cell Signaling), rabbit polyclonal anti-mouse PTRF (Abcam), mouse anti-GFP (Roche), with mouse monoclonal anti-tubulin or mouse monoclonal anti-GAPDH used as the loading controls. Secondary antibodies were goat anti-mouse 800 and goat anti-rabbit 680. All membranes were scanned with the Odyssey imaging system (Li-COR).

## Identification of ubiquitinated Cav1

Cells were lysed in Ub lysis buffer (50 mM Tris–HCl pH 7.5, 150 mM NaCl, 1% Triton x-100, 0.1% SDS and 1% sodium deoxycholate) supplemented with phenylmethylsulfonyl fluoride (1 mM), sodium pervandate 1 mM final, sodium fluoride 30 mM, and leupeptin and aprotinin 10 µg ml⁻¹. Supernatants obtained after centrifugation at 16.000g were used to immunoprecipitate Cav1, after extensive washing and ubiquitinated forms of Cav1 were identified by western blotting against ubiquitin (Enzo, ENZ-ABS840-0100), as previously described[34].

**Image analysis.** To quantify the ubiquitinated signal, a wide region representative of the whole lane was quantified and the signal obtained in the Cav1 KO MEFs lane was subtracted as background. This signal was divided by the amount of Cav1, which was obtained by quantification of the region around Cav1 bands minus the signal obtained in the same region in the Cav1 KO MEF lane.

## Confocal microscopy

Confocal images were obtained with an LSM 700 inverted confocal microscope (Carl Zeiss) fitted with a 63× 1.4 NA objective and driven by Zen software (Carl Zeiss).

## RT-DC

In RT-DC, single suspended cells are flown at room temperature through a microfluidic chip that has a deformation channel that is 300 µm long and 30 µm × 30 µm in cross-section. As the channel is wider than the cells, they are deformed owing to hydrodynamic forces only. Cells are captured by a high-speed camera at the end of the deformation channel (1,000 cells s⁻¹ at around 1 kPa), where cells show a deformed-bullet-like shape. Images are analysed in real time to obtain the contour of each cell. Moreover, the contour is used to calculate cell size and deformation. To ensure cells travel in the middle of the deformation channel, a sheath flow is applied. Both the cell suspension and sheath fluid are driven by syringe pumps, which run at the flow rates 0.08 µl s⁻¹ and 0.24 µl s⁻¹, respectively. The measurement buffer of both, sample and sheath fluid, is based on PBS (without Mg²⁺ and Ca²⁺) that is supplemented with 0.5% (w/w) methyl cellulose (Sigma-Aldrich).

## Laurdan GP

The generalized polarization (GP) defined by E. Gratton and co-workers was measured in a custom made multiphoton microscope using the lipophilic fluorescence probe Laurdan[118]. Briefly, Laurdan is a fluorescent molecule that detects changes in membrane fluidity through its sensitivity to changes in the polarity of the environment in the lipid bilayer, such as penetration of water or cholesterol content. Polarity changes are transduced in shifts in the Laurdan emission spectrum. When Laurdan is immersed within a phospholipid bilayer in a disordered phase, its emission is centred at 490 nm, while is shifted to blue (around 440 nm) in a more packed phase. This can be quantified by calculating the GP defined as:

$$GP = \frac{I_{440} - GI_{490}}{I_{440} + GI_{490}}$$

where $I_{440}$ and $I_{490}$ are the emission intensities measured centred at 440 nm and 490 nm, and $G$ accounts for corrections for the wavelength dependence of the emission detection system (accomplished through the comparison of the GP value of a known solution (Laurdan in dimethyl sulfoxide[119]). We used three-photon excitation of the Laurdan molecule to achieve a small excitation volume and reduce out-of-focus photobleaching. Three-photon excitation was achieved using a fibre laser (FemtoPower, Fianium Ltd.) that delivers 180 fs pulses at 1,064 nm central wavelength, with 20 MHz repetition rate. The pulses, after passing a polarizer and a $\lambda/2$ waveplate to control the power (EKSMA Optics), were sent by a set of galvo-scanners (Thorlabs, GVSM002/M)

through a telescope, to fulfil the back aperture of the objective, to an upright microscope (Nikon Ni) equipped with an incubation chamber (LIS) for keeping cell culturing conditions (temperature and humidity control). For detection, one PMT (H9305-01, Hamamatsu) was used sequentially for the two emission channels of Laurdan with filters Brightline Fluorescence Filter 438/24 nm and Brightline Fluorescence Filter 483/32 nm (Semrock). The objective used was a 60× water immersion with NA 1.0 (Nikon). The setup is shown in Extended Data Fig. 6c. Z-stacks of individual cells were taken for each channel using homemade software (Labview), recording the voltage values of the PMT at each pixel. The factor G was calculated for each experiment from imaging a volume of Laurdan (10 μg ml⁻¹ in dimethyl sulfoxide) for the two channels. The third-order behaviour of the excitation within the power range used for our experiments is shown in Extended Data Fig. 6e.

**Stretching device and experiments.** Pure mechanical stretching was done using a stretching device[120], shown in Extended Data Fig. 6d, that comprises a stretchable polydimethylsiloxane (PDMS) membrane clamped between two Teflon rings and placed on top of a circular loading post. Application of vacuum to the outer elliptical region of the membrane causes uniaxial strain. Cells were plated on the PDMS membrane coated with collagen (10 μm ml⁻¹) 24 h before the experiments. Then, they were incubated in Laurdan (10 μM) for 1 h. Images were taken at two timepoints, before applying the stretching and during the application of the stretching. From the images the GP was calculated for each condition using homemade software developed in MATLAB.

**Fabrication of stretchable PDMS membranes.** PDMS was mixed 10:1 (base:crosslinker) and degassed for 1 h. Uncured PDMS was spin-coated on methacrylate plates to a thickness of 80–100 μm, and cured at 65 °C overnight. The resulting PDMS membranes were then peeled off the plates and clamped between the rings of the stretching device[120].

## Two-photon FLIM
Fluorescence lifetime was acquired by an inverted Nikon Eclipse Ti microscope, using a Plan Apo VC 60× A/1.20 WI water immersion objective (Nikon). Two-photon excitation was obtained using a tunable Spectra Physics femtosecond laser, model Mai Tai DeepSee, coupled to an acousto-optic pulse picking modulator and detected with an Alba imaging workstation (ISS). Cells treated with 25-NBD-cholesterol were excited at 900 nm with laser power of between 0.75 mW and 1.84 mW at the sample, and the emission was collected with an 530/43 nm bandpass filter. FLIM was performed using the digital frequency domain method and the FLIM box described in detail by Colyer et al.[121], which were developed at the Laboratory of Fluorescence Dynamics (LFD, University of California, Irvine, CA, USA) and implemented in an Alba imaging workstation. Data were acquired and processed by VistaVision_x64_V4.2_Build 364 software (ISS). The scan area (256 × 256 pixels), acquired with a pixel dwell time of 64 μs, was within the range of 50 × 50 μm² to 70 × 70 μm², with a pixel size between 195 nm and 273 nm, and a voxel size between 0.195 × 0.195 × 1.000 μm³ and 0.273 × 0.273 × 1.000 μm³ (X, Y, Z). Before sample measurements, a concentrated fluorescein solution at pH 9.5 was measured and used as fluorescence lifetime calibration. Fluorescein lifetime (4.04 ns) was determined separately in a fluorometer (PC1; ISS).

**FLIM analyses.** Fluorescence lifetime decays were analysed by the phasor-FLIM method[48,122], using phasor analysis module of VistaVision_x64_V4.2_Build 364 software (ISS). The distribution was obtained by converting the multiexponential fluorescence decays acquired in each pixel into the graphical representation of a phasor. In brief, the phasor transformation does not assume any fitting model for fluorescence lifetime decays. It simply expresses the overall decay in each pixel in terms of a vector of (s, g) polar coordinates in the so-called universal circle[123].

**Cells and reagents.** We stained cells with 25-NBD-cholesterol (810250C, Avanti Polar Lipids). To avoid overlapping of 25-NBD-cholesterol and GFP (from the MIGR1 retroviral vector) FLIM images, we subcloned Cav1 and PTRF into mCherry lentiviral vector and obtained stable expressing lines (Extended Data Fig. 4f-q). The lifetime of 25-NBD-cholesterol changes according to its incorporation within distinct subcellular pools as follows: it becomes longer when it is packed into cholesterol-dense domains and vice versa; it decreases when 25-NBD-cholesterol is accrued as part of cholesterol pools in less compacted areas.

**Structural analysis, in silico model and docking of mouse Cav1**
**In silico modelling of Cav1.** There are neither homology structures nor complete structural models for Cav1 or the putative membrane domain. Fasta sequences of mature Cav1 protein (Uniprot ID: P49817, residues 2-178) were submitted to a local implementation of Rosetta software suite v3.8 (www.rosettacommons.org)[124] for ab initio modelling using the mp_framework of the suite. After clustering and filtering, the best model with minimal energy and correct global topology (compatible with α-hairpin insertion in membrane, C-term parallel to membrane plane and exposed scaffold domain) was selected as final template for refinement using the mp_relax tool[125,126] of Rosetta suite v3.8 (www.rosettacommons.org). As before, the model with minimal energy and correct topology was selected as the final model (Extended Data Fig. 4r). To better show the protein position and orientation in the membrane, the model before was submitted to the PPM server (http://opm.phar.umich.edu/). This server is specialized in predicting and positioning membrane proteins from 3D structures using a large structural database (membranome) and computational methods (Extended Data Fig. 4s and Supplementary Video 9).

**Docking of homodimers for Cav1.** For docking the dimeric form of Cav1, two monomers models modelled before were positioned close to the putative dimeric interface for cav1 previously published in bibliography using the pymol v2.0 (The PyMOL Molecular Graphics System, Version 1.8.6.0 Open Source, Schrödinger, LLC. www.pymol.org) program and the new dimeric model was used as initial template. For symmetric docking, a new PDB file for the template before where the membrane protein structure is transformed into PDB coordinates (z axis is membrane normal) using the PPM server (http://opm.phar.umich.edu/server.php) was generated. To generate full symmetric spanfile from the PDB structure, the mp_spanfile_from_pdb application from the membrane framework of Rosetta suite v3.8 (www.rosettacommons.org) was used, and the obtained results were manually curated and modified as necessary. An initial cycle of relax for minimize E and clashes using the initial template and the full spanfile was made using the relax application from the membrane framework of Rosetta suite v3.8 (www.rosettacommons.org)[127]. The lowest-scoring refined model (lower E) was selected as input model, and an asymmetric input structure and symmetry C2 definition file was generated using the make_symmdef_file.pl script application from the Rosetta suite v3.8 (www.rosettacommons.org). Using the asymmetric monomer input and symmetry C2 definition generated before and fixing the known docking interface as ambiguous constraints, symmetry docking was made using the mp_symdock application[128] from the membrane framework of Rosetta suite v3.8 (www.rosettacommons.org). The best model with minimal E compatible with membrane topology and close to the theoretical dimer interface was selected. A final cycle of relax for minimize E and clashes using the model before as template and the full spanfile was made using a new cycle of docking with positional restrictions (dimer interface) using the relax application from the Rosetta suite v3.8 (www.rosettacommons.org)[125]. The lowest-scoring refined model (lower E) with correct topology and interface was selected as the final model. To better show the dimer position and orientation in the membrane (Extended Data Fig. 4t and Supplementary Video 10),

the models before were submitted to the PPM server (http://opm.phar. umich.edu/).

**In silico modelling of Cav1-PTM.** For modelling the post-translational modifications (PTM) of Y14, Cys 133, 143 and 156, Y14 was changed to Y-phosphorylated and Cys 133, 143 and 156 for *S*-palmitoyl-Cys in the Cav1 mouse model modelled before to obtain a new template with PTM. As before, the template was submitted to a local implementation of Rosetta software suite v3.9 (www.rosettacommons.org)[124] for modelling and refinement using the mp_relax tool[125,126] of Rosetta suite v3.9 (www.rosettacommons.org). The model with minimal energy and correct topology was selected as the final model.

**Docking of homodimers for Cav1-PTM.** Symmetric docking of the dimeric form of Cav1-PTM, was modelled as before using the Ca1-PTM monomer as initial template.

**Ligand docking of cholesterol and homodimers of Cav1-PTM.** For modelling the interaction of the cholesterol ligand and the symmetric homodimer of Cav1-mouse, the main conformer of cholesterol was positioned close to the putative binding site for Cav1 previously published in bibliography using the pymol v2.0 (The PyMOL Molecular Graphics System, Version 1.8.6.0 Open Source, Schrödinger, LLC. www.pymol.org) program and the new model was used as initial template for ligand docking using a ligand-docking script for the Rosetta suite v3.9 (www.rosettacommons.org). After clustering and filtering, the best model with minimal E compatible with membrane topology and the theoretical dimer interface and cholesterol binding site was selected. As before, to better show the dimer position and orientation in the membrane (Extended Data Fig. 4u), the model before was submitted to the PPM server.

### Statistics and reproducibility

Data are presented as mean ± s.e.m. unless otherwise indicated. Mean values were compared by unpaired two-tailed Student's *t*-test unless otherwise indicated. For box plot representations (unless otherwise indicated), a box was drawn from the first quartile (Q1) to the third quartile (Q3) and the central line corresponds to the median. The whiskers go from Q1/Q3 quartile to the lowest/greatest observed data point that falls at a distance of 1.5 times the interquartile range (IQR) below/above the corresponding quartile. Differences were considered statistically significant at $P < 0.05$ (*), $P < 0.01$ (**) and $P < 0.001$ (***). Figures 1d–f and 5a–d show representative electron micrographs from five independent EM sessions, with similar results. Figure 2a shows a representative OS image from ten independent sessions, with similar results. Figure 4a–c and Extended Data Fig. 3b show representative STORM images from ten independent super-resolution microscopy sessions, with similar results. Figure 7f shows a representative quantification of biochemical fractionation of Cav1 complexes from three independent assays, with similar results. Extended Data Fig. 1i–n shows representative confocal images from six independent microscopy sessions, with similar results. No statistical methods were used to pre-determine sample sizes, but our sample sizes are similar to those reported in previous publications[129,130]. No data were excluded from the analyses unless otherwise indicated; the experiments were not randomized; the Investigators were not blinded to allocation during experiments or outcome assessment.

### Reporting summary

Further information on research design is available in the Nature Portfolio Reporting Summary linked to this article.

### Data availability

Previously published data that were re-analysed here are available under accession code Uniprot ID: P49817, residues 2-178 (mature Cav1 protein). We have deposited the following files in Zenodo: dataset from YAP experiments (https://doi.org/10.5281/zenodo.7061911), script for YAP analysis (https://doi.org/10.5281/zenodo.7061924) and STORM images set (https://doi.org/10.5281/zenodo.7062213). Source data are provided with this paper. All other data supporting the findings of this study are available from the corresponding author on reasonable request.

### Code availability

A detailed report of the mathematical model can be found as Supplementary Note 1. For further information about code, please contact M.A. or N.W.

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

## Acknowledgements

We thank R. Parton (Institute for Molecular Biosciences, Queensland), P. Pilch (Boston University School of Medicine) and L. Liu (Boston University School of Medicine) for kindly providing PTRFKO cells and reagents, S. Casas Tintó for kindly providing SH-Sy5y cells, P. Bassereau (Curie Institute, Paris) for kindly providing OT setup, V. Labrador Cantarero from CNIC microscopy Unit for helping with ImageJ analysis, O. Otto and M. Herbig for providing help with RTDC experiments, S. Berr and K. Gluth for technical assistance in cell culture, F. Steiniger for support in electron tomography, and A. Norczyk Simón for providing pCMV-FLAG-PTRF construct. This project received funding from the European Union Horizon 2020 Research and Innovation Programme through Marie Sklodowska-Curie grant 641639; grants from the Spanish Ministry of Science and Innovation (MCIN/AEI/10.13039/501100011033): SAF2014-51876-R, SAF2017-83130-R co-funded by 'ERDF A way of making Europe', PID2020-118658RB-I00, PDC2021-121572-100 co-funded by 'European Union NextGenerationEU/PRTR', CSD2009-0016 and BFU2016-81912-REDC; and the Asociación Española Contra el Cáncer foundation (PROYE20089DELP) all to M.A.d.P. M.A.d.P. is member of the Tec4Bio consortium (ref. S2018/NMT¬4443; Comunidad Autónoma de Madrid/FEDER, Spain), co-recipient with P.R.-C. of grants from Fundació La Marató de TV3 (674/C/2013 and 201936-30-31), and coordinator of a Health Research consortium grant from Fundación Obra Social La Caixa (AtheroConvergence, HR20-00075). M.S.-A. is recipient of a Ramón y Cajal research contract from MCIN (RYC2020-029690-I). The CNIC Unit of Microscopy and Dynamic Imaging is supported by FEDER 'Una manera de hacer Europa' (ReDIB ICTS infrastructure TRIMA@CNIC, MCIN). We acknowledge the support from Deutsche Forschungsgemeinschaft through grants to M.M.K. (KE685/7-1) and B.Q. (QU116/6-2 and QU116/9-1). Work in D.N. laboratory was supported by grants from the European Union Horizon 2020 Research and Innovation Programme through Marie Sklodowska-Curie grant 812772 and MCIN (DPI2017-83721-P). Work in C.L. laboratory was supported by grants from Curie, INSERM, CNRS, Agence Nationale de la Recherche (ANR-17-CE13-0020-01) and Fondation ARC pour la Recherche (PGA1-RF20170205456). Work in P.R.-C. lab is funded by the MCIN (PID2019-110298GB-I00), the EC (H2O 20-FETPROACT-01-2016-731957). Work in X.T. lab is funded by the MICIN (PID2021-128635NB-I00), ERC (Adv-883739) and La Caixa Foundation (LCF/PR/HR20/52400004; co-recipient with P.R.-C.). IBEC is recipient of a Severo Ochoa Award of Excellence from the MINECO. The funders had no role in study design, data collection and analysis, decision to publish or preparation of the manuscript. The CNIC is supported by the Instituto de Salud Carlos III (ISCIII), the MCIN and the Pro CNIC Foundation, and is a Severo Ochoa Center of Excellence (grant CEX2020-001041-S funded by MICIN/AEI/10.13039/501100011033).

## Author contributions

F.-N.L. and M.A.d.P. conceived/supervised the project, designed experiments, analysed results and wrote the paper that was edited by M.S.-A. N.W. and M.A. developed the mathematical model with inputs from M.A.d.P. and F.-N.L., edited the text and contributed to the writing. E.S., M.M.K. and B.Q. designed, performed and analysed FRIL studies that led to the visualization of dolines, edited the text and contributed to the writing. D.Z. and S.A.S. designed and analysed Laurdan experiments together with F.-N.L. G.C. and F.-N.L designed and analysed OS experiments with supervision from J.G. F.-N.L. performed and analysed RT-DC experiments with supervision from J.G. and G.C. M.Z. and V.R.C. designed, performed and analysed FLIM studies together with F.-N.L., edited and contributed to the writing. C.V.d.L., C.L., F.-N.L. and M.A.d.P. designed OT experiments that were performed and analysed by C.V.d.L. F.M.d.B. developed the in silico model of Cav1 with inputs from F.-N.L. M.S.-A. designed, performed and analysed YAP experiments together with F.-N.L. J.J.U., A.E. and J.-C.E. performed and analysed AFM studies with supervision from D.N. and J.G. F.-N.L., V.R.C. and D.J.-C. designed, performed and analysed STORM experiments. F.-N.L. performed and analysed magnetic tweezers studies with supervision from P.R.-C. and X.T. D.M.P. performed experiments and contributed to data analysis.

## Competing interests

The authors declare no competing interests.

## Additional information

**Extended data** is available for this paper at https://doi.org/10.1038/s41556-022-01034-3.

**Correspondence and requests for materials** should be addressed to Fidel-Nicolás Lolo or Miguel A. del Pozo.

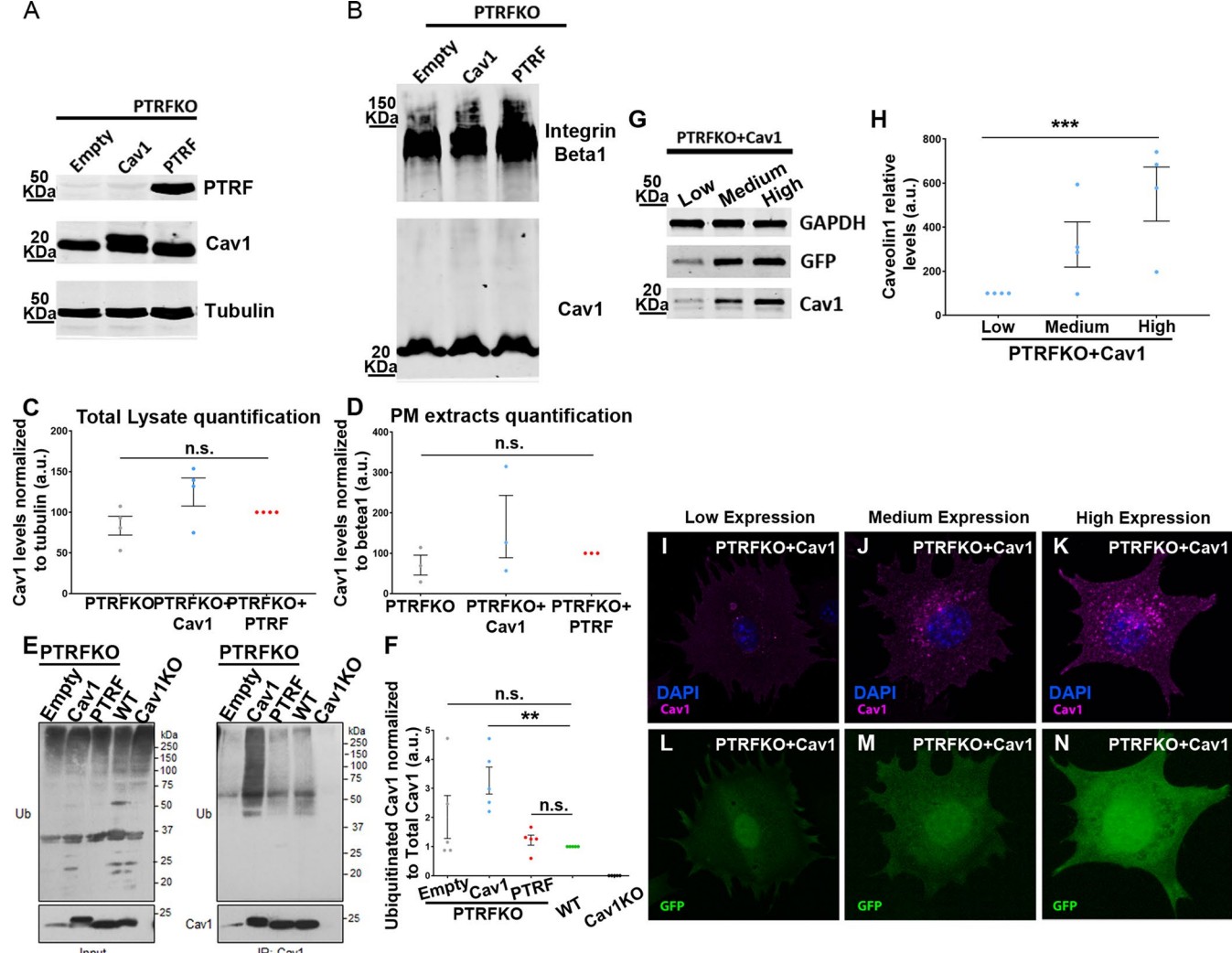

**Extended Data Fig. 1 | Extended Data Fig. 1, related to Fig. 1, Biochemical characterization. (A and C)** Western blot analysis of whole cell lysates from PTRFKO MEFs reconstituted with either empty vector, Cav1 or PTRF. Samples were immunoblotted for PTRF, Cav1, and tubulin (loading control; re-expressed Cav1 contains a Flag tag of 7 Aa, yielding a slight band shift), quantified in C (quantification of Cav1 levels normalized to the tubulin levels). Data are presented as mean ± s.e.m. *n* = 4 independent western blots were analyzed. Statistical comparisons were by two-tailed Student's t-test, not significant at p > 0.05. **(B and D)** Biochemical fractionation of PMs from PTRFKO MEFs reconstituted with either empty vector, Cav1 or PTRF. Samples were immunoblotted for Cav1 and total beta 1 integrin (loading control), quantified in D (quantification of Cav1 levels normalized to the beta 1 integrin levels). Data are presented as mean ± s.e.m. *n* = 3 independent western blots were analyzed. Statistical comparisons were by t-test, not significant at p > 0.05. **(E)** Indicated cell lines were lysed and Cav1 was immunoprecipitated. Immunopurified Cav1 fractions (IP: Cav1) and the whole cell lysates (input) were blotted against

mono- and poly-ubiquitin reactive antibody (Ub), and Cav1 antibody. Note that in negative control Cav1 KO MEFs neither ubiquitin nor Cav1 signal was detected, illustrating the specificity of both signals. **(F)** Quantification of the signal obtained in the Ub blot normalized to the Cav1 levels (signal of Cav1 in the IPs) is represented. Data are presented as mean ± s.e.m. *n* = 5 independent western blots were analyzed. Statistical comparisons were by two-tailed Student's t-test (p = 0.001; N.S.: not significant). **(G and H)** Western blot analysis of whole cell lysates from PTRFKO MEFs reconstituted with Cav1 and sorted into three populations (low, medium and high) according to EGFP marker (which is bicistronic with the Cav1 ORF) expression levels, EGFP expression correlates with Cav1 expression as quantified in H. Statistical comparisons were by two-tailed Student's t-test (p = 0.0003). **(I-N)** Immunofluorescence images of the three populations (low, medium and high) of PTRFKO MEFs reconstituted with Cav1. GFP is shown in green, Cav1 in magenta and DAPI in blue. **Source numerical data and unprocessed blots are available in source data.**

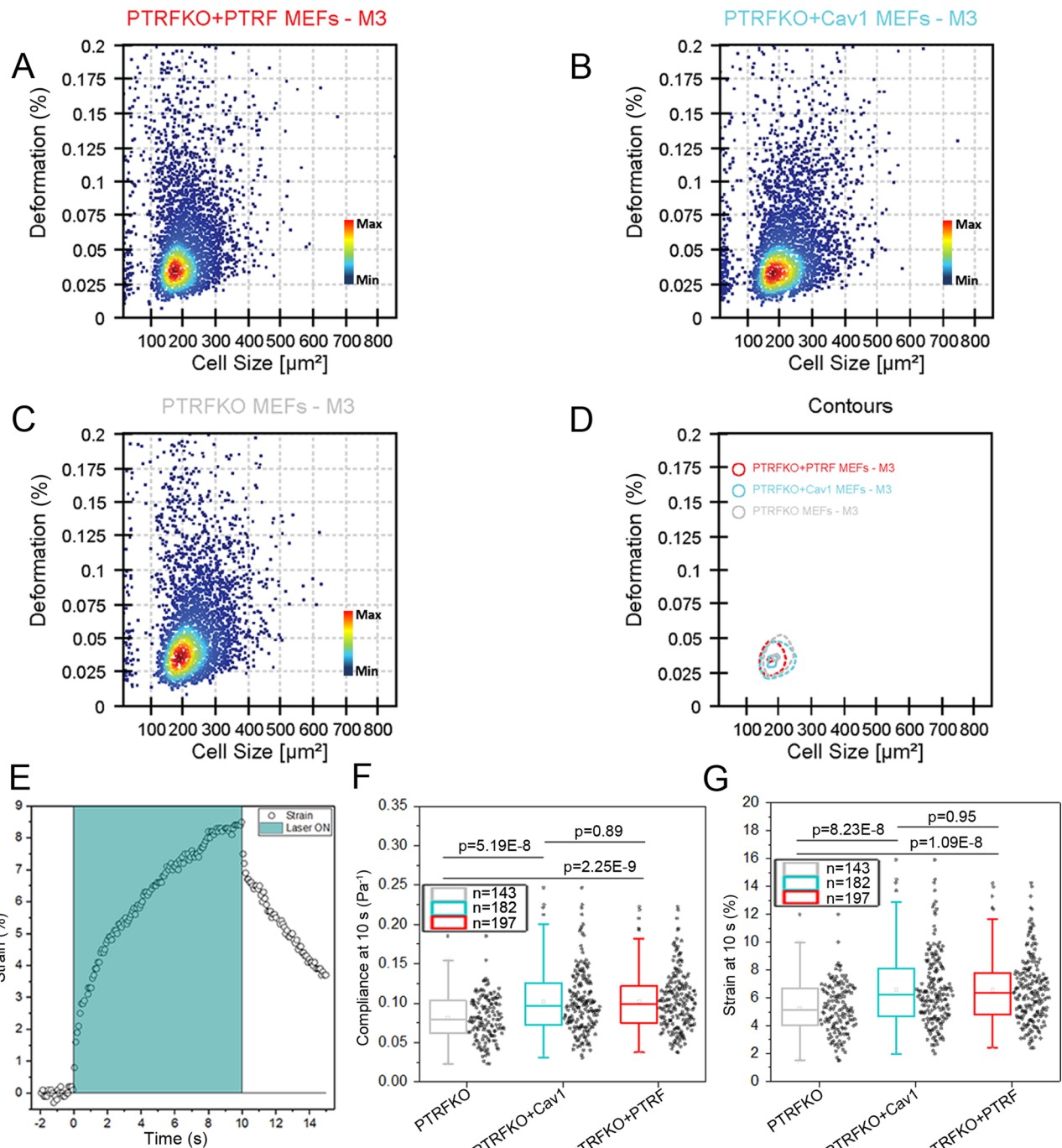

**Extended Data Fig. 2 | Extended Data Fig. 2, related to Fig. 2, RT-DC and OS. (A-D)** Real-time deformability scatter plots (deformation versus cell size in μm²) of PTRFKO MEFs reconstituted with PTRF [A], Cav1 [B] or PTRFKO MEFs without reconstitution [C]. Color indicates a linear density scale; a 50%-density contour plot of the three genotypes is shown [D]. Measurements were recorded at a flow rate of 0.32 μl/s (M3) in a 20 μm × 20 μm channel. **(E)** Optical Stretcher measurement scheme (strain in % vs time in seconds), showing the deformation (laser on) and relaxation regions. **(F and G)** Compliance and strain across different genotypes after 10 s stretching, right before relaxation. Statistical comparisons were by one-way ANOVA with Bonferroni post-test, with significance assigned at *p < 0.05 and not significant at p > 0.05. The number of total cells analyzed per genotype (N) is indicated, pooled from 4 independent experiments. **Source numerical data are available in source data**.

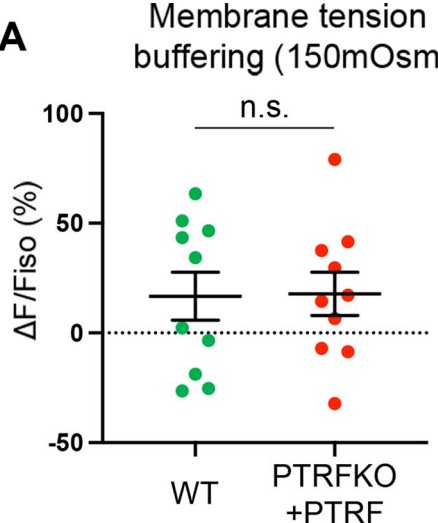

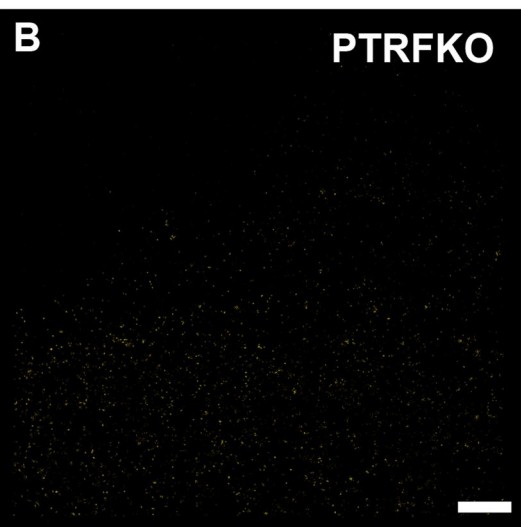

**Extended Data Fig. 3 | Extended Data Fig. 3, related to Fig. 3, OT and dSTORM. (A)** Relative change of the mean tether force after hypo-osmotic shock (150 mOsm) for wild type MEFs (WT, $n$ = 10) and PTRFKO MEFs reconstituted with PTRF ($n$ = 10). $n$ = cells pooled from 4 independent experiments. Individual values are plotted (data are presented as mean values + /- SEM), statistical analysis strategy used was ANOVA with Tukey's multiple comparisons test, N. S., non-significant. **(B)** dSTORM representative image of PTRFKO MEF. **Source numerical data are available in source data**.

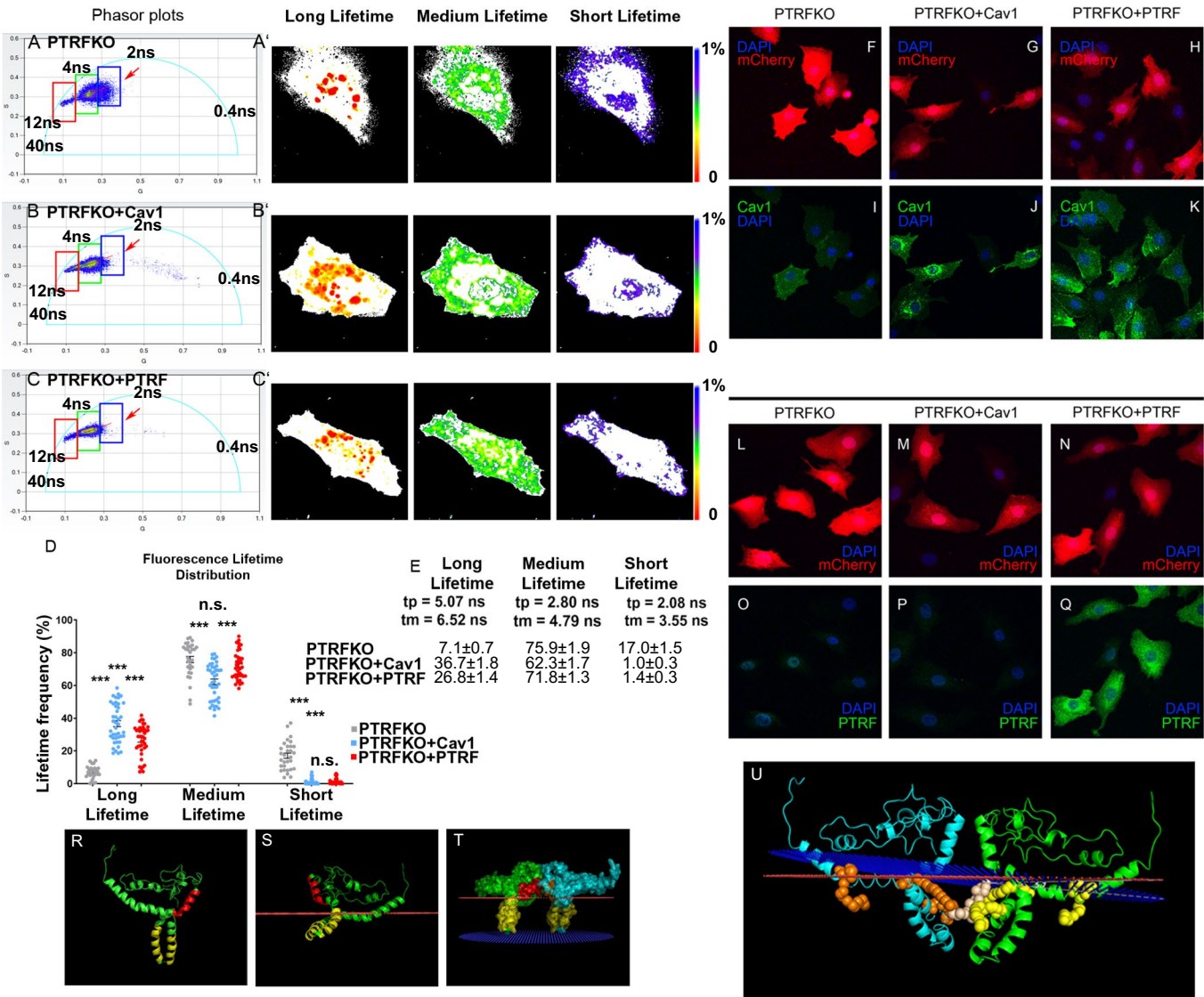

**Extended Data Fig. 4 | Extended Data Fig. 4, FLIM studies and in silico model.**
**(A-C)** Changes in the distribution of fluorescence lifetimes in either PTRFKO [A], PTRFKO + Cav1 [B] or PTRFKO + PTRF [C] MEFs were analyzed across 3 LUT scale intervals. Three different subpopulations of fluorescence lifetime were selected: 1) long fluorescence lifetime (center distribution coordinates: TP = 5.07 ns; TM = 6.52 ns); 2) medium (TP = 2.80 ns; TM = 4.79 ns) and 3) short (TP = 2.08 ns; TM = 3.55). Changes in the fractions of each distribution were compared across genotypes. [Phasor representation of fluorescence lifetime is not linear]. Red arrows: short lifetime values across genotypes highlighted. **(A'-C')** Representative images of PTRFKO [A'], PTRFKO + Cav1 [B'] and PTRFKO + PTRF [C'], pseudocolored according to their fluorescence lifetime values. **(D and E)** Average representation of the lifetime intervals for the three genotypes [D], also expressed in numbers [E] (*n* = 30 cells (PTRFKO), 42 (PTRFKO + Cav1) and 42 (PTRFKO + PTRF) pooled from 7 (PTRFKO) and 8 (PTRFKO + Cav1 and PTRFKO + PTRF) independent experiments. Plots: mean values ± s.e.m. Statistics: ANOVA, Sidak's multiple comparisons test, ***$p$ < 0.0001. Differences between PTRFKO Medium vs PTRFKO + PTRF Medium and PTRFKO + Cav1 Short

vs PTRFKO + PTRF Short were not significant (N.S., $p$ = 0.3227 and p > 0.999, respectively). **(F-H and L-N)** Confocal images of PTRFKO MEFs reconstituted with mCherry lentiviral vector, either empty [F and L], expressing Cav1 [G and M] or PTRF [H and N]. **(I-K and O-Q)** Immunofluorescence images of Cav1 [I-K] or PTRF staining [O-Q]. Red: mCherry; green: Cav1/PTRF; blue: DAPI. **(R, S)** 3D models of the Cav1 protein (cartoon representation) showing the putative alpha-hairpin membrane insertion in yellow and the scaffold domain in red [R]; membrane layers are depicted as red dots planes [S]. **(T)** Front view of the 3D model of the Cav1 symmetric dimer from ppm (https://opm.phar.umich.edu/ppm_server2) server (surface representation, green A chain, cyan B chain) showing the putative alpha-hairpin membrane insertion (yellow), the dimer interface (as red/orange) and the membrane layers as red-blue dots planes. **(U)** Cav1 dimer structural model, indicating the curvature angle imposed to the membrane (-5º), the three palmitoylated cysteines (orange and yellow), and one molecule of cholesterol between the monomers (wheat). **Source numerical data are available in source data. FLIM image galleries are available in** Supplementary Figs. 1, 2 and 3.

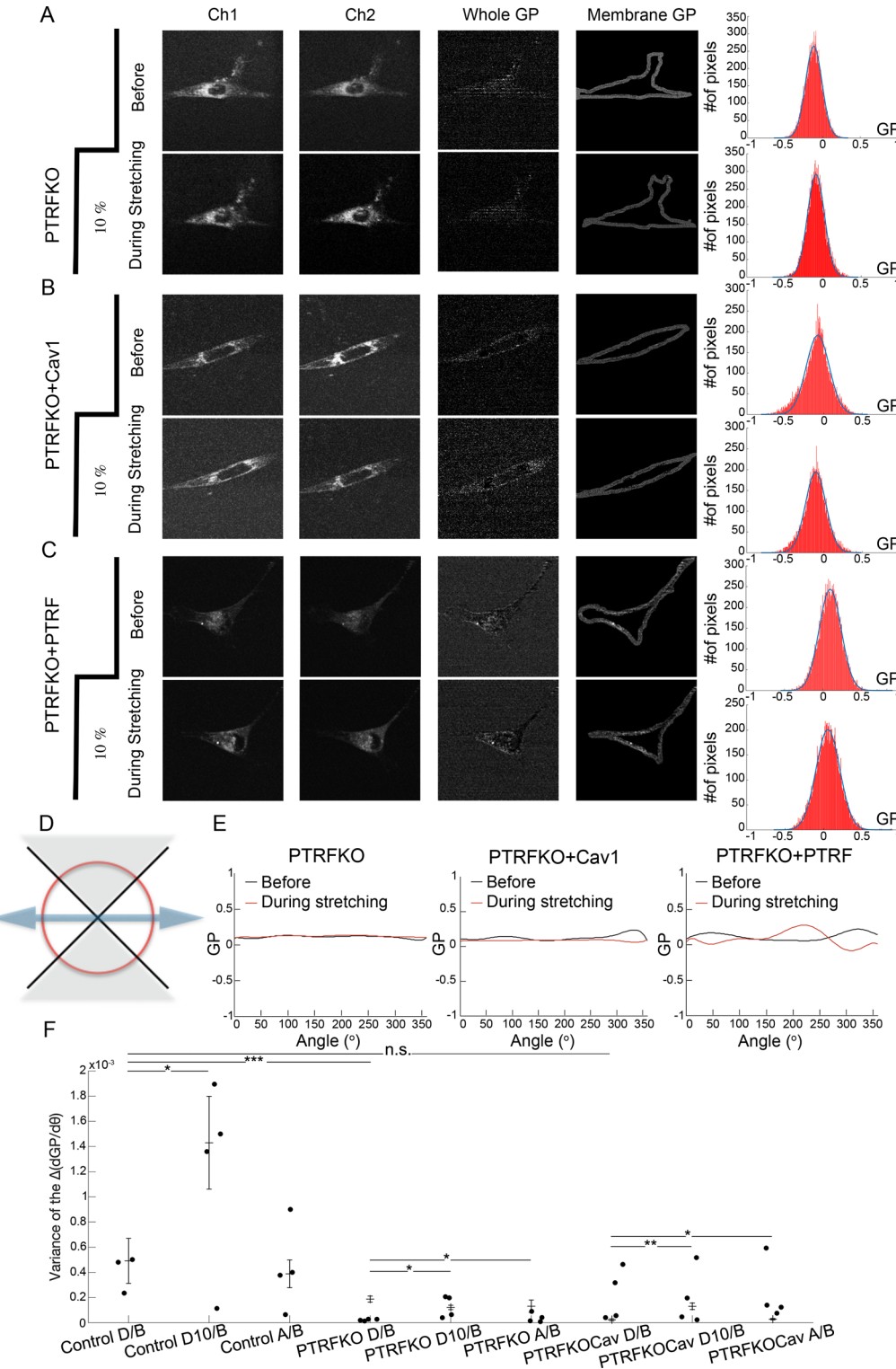

**Extended Data Fig. 5 | Extended Data Fig. 5, Laurdan studies. (A-C)**
Representative images acquired from Laurdan stained PTRFKO [A],
PTRFKO + Cav1 [B] and PTRFKO + PTRF [C] MEFs before and during stretch
(10%), for channels centered at 440 nm (Ch1) and 490 nm (Ch2), GP of the whole
cell, GP of the border of the cell at a certain z-plane, and the corresponding
histogram with Gaussian fitting. **(D)** Schematic of the parts of the cell stretched
by the unidirectional device. Cell borders where divided in sectors for angles
0-2π, with quadrants from π/4 to 3π/4 and from 5π/4 to 7π/4 containing the part
of the cell membrane that were under stretch. **(E)** GP of a representative cell for

the PTRFKO, PTRFKO + Cav1 and PTRFKO genotypes, as a function of the angle.
**(F)** Variance of the change in GP with the angle for: PTRFKO + PTRF (Control),
PTRFKO and PTRFKO + Cav1 cell lines (*n* = 4 cells for each cell line, pooled from 3
independent experiments). Data are presented as mean values ± s.e.m. (D: during
stretching, B: before stretching, D10: during stretching for 10 minutes, A: after
stretching). Statistical comparisons were by two-tailed Student's t-test, with
significance assigned at *p < 0.05, **p < 0.01, ***p < 0.001 and not significant at
*p* > 0.05. **Source numerical data are available in source data.**

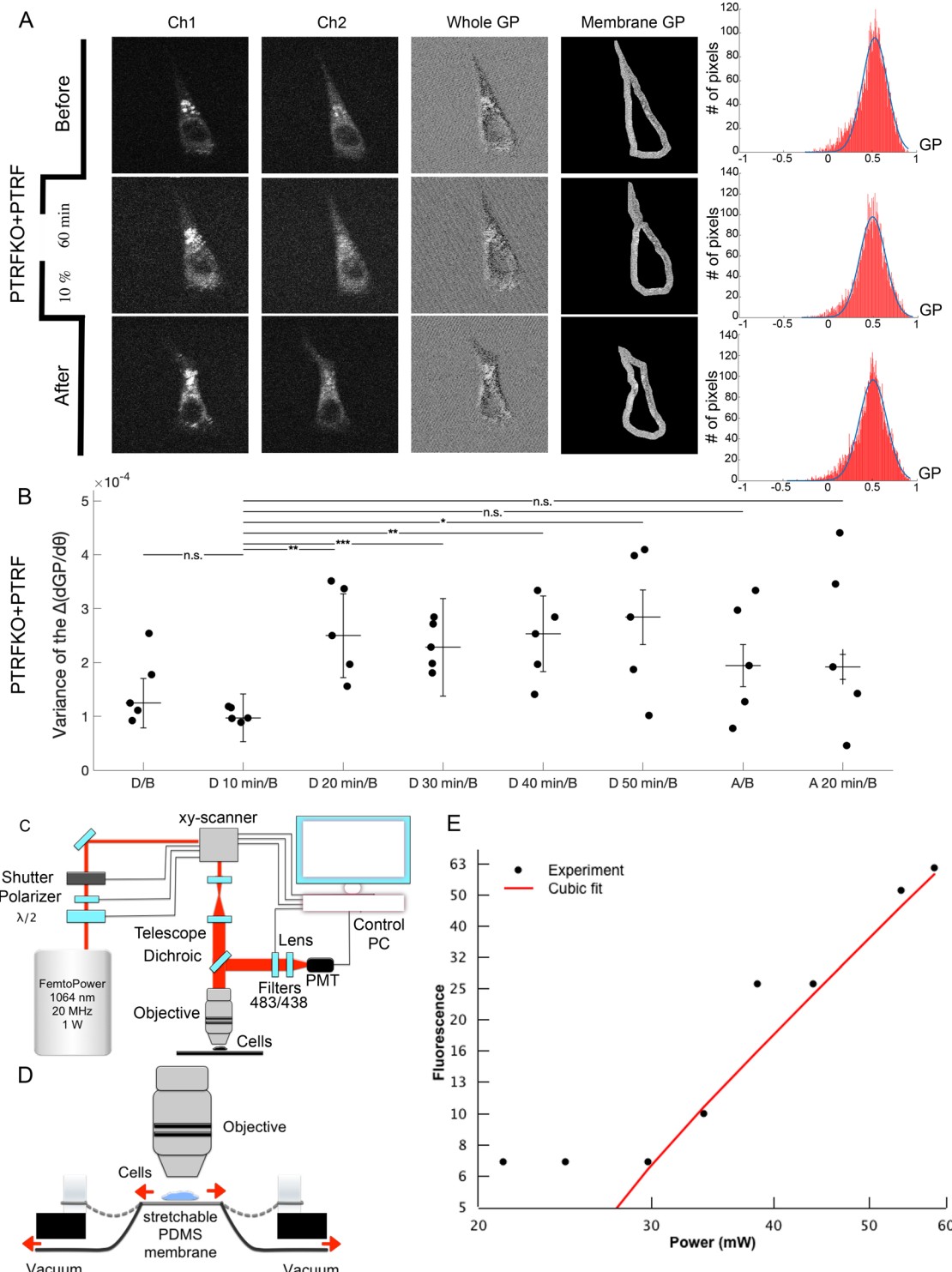

**Extended Data Fig. 6 | Extended Data Fig. 6, Laurdan studies. (A)** A representative of PTRFKO + PTRF MEFs before, during (10% for 60 min), and after stretch for channels centered at 440 nm (Ch1) and 490 nm (Ch2), GP of the whole cell, GP of the border of the cell at a certain z-plane, and the corresponding histogram with Gaussian fitting. **(B)** Variance of the change in GP with the angle for different time points during the experiment for PTRFKO + PTRF MEFs ($n = 5$ independent experiments, each of them with one cell analyzed over the indicated time points), D: during stretching, B: before stretching, D10, D20, D30, D40, D50:

during stretching for 10, 20, 30, 40 or 50 minutes, A: after stretching, A20: after 20 min post stretching. Data are presented as mean values ± s.e.m. Statistical comparisons were by paired two-tailed Student's t-test, with significance assigned at *$p < 0.05$, **$p < 0.01$, ***$p < 0.001$ and not significant at $p > 0.05$. **(C)** Schematic of the multi-photon set-up. **(D)** Schematic of the stretching device. **(E)** Plot of the intensity of the fluorescence as function of the input power showing three-photon behavior. **Source numerical data are available in source data.**

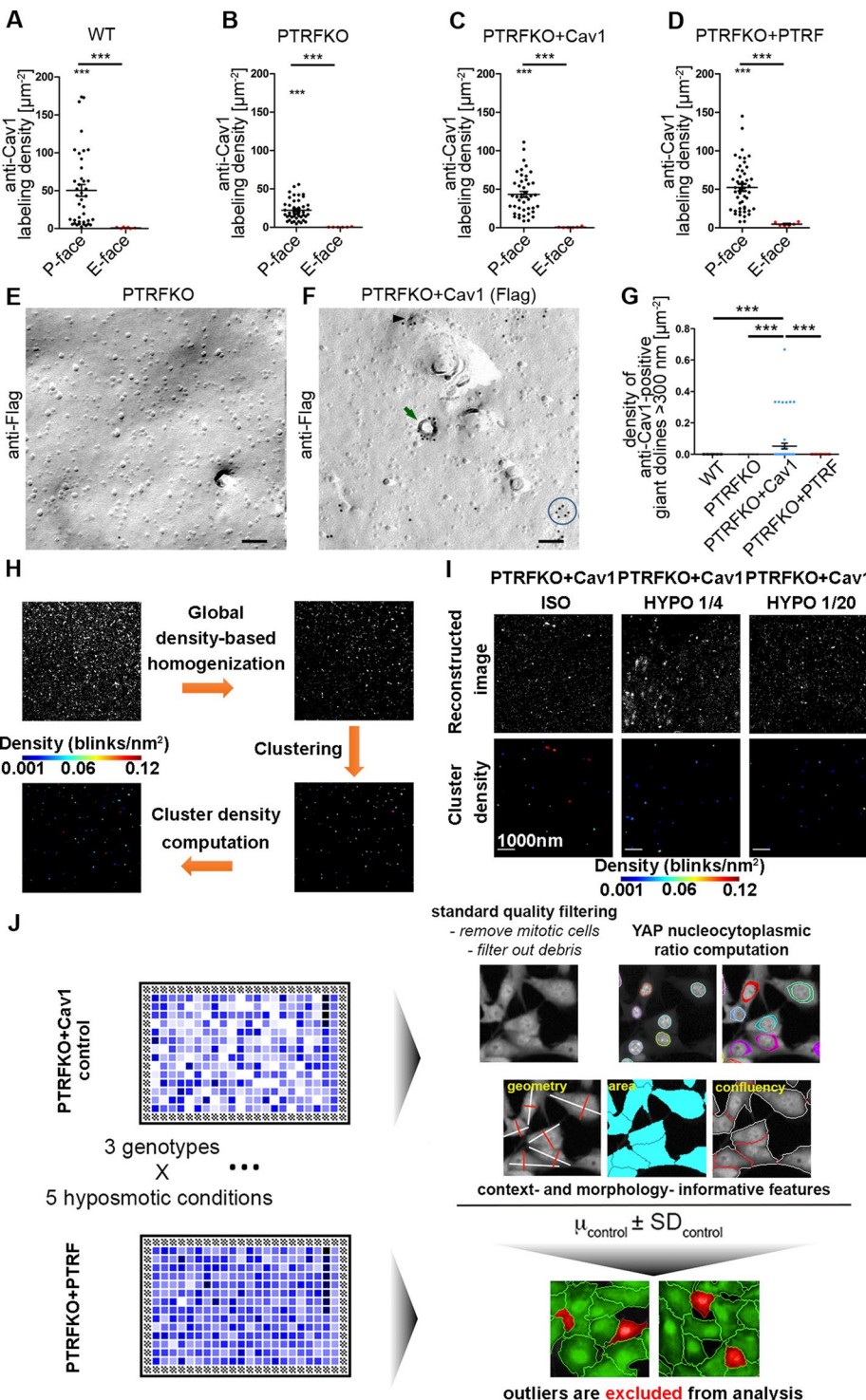

**Extended Data Fig. 7 | See next page for caption.**

**Extended Data Fig. 7 | Extended Data Fig. 7, related to Figs. 5 and 6, FRIL, dSTORM and YAP analysis. (A-D)** Anti-Cav1 immunogold labeling specificity in freeze-fracture replica samples from MEFs. Anti-Cav1 immunogold labeling density of P-faces of replica of WT [A], PTRFKO [B], PTRFKO + Cav1 [C] and PTRFKO + PTRF MEFs [D] as compared to labeling densities found at E-faces of the membrane (negative control). $n$ = number of images pooled from 3 independent experiments, 12 replicates per condition were examined. **(E, F).** Electron micrographs of freeze-fractured membranes from PTRFKO MEFs [E] and PTRFKO + Cav1 MEFs [F] immunostained with anti-Flag antibodies and 10 nm gold-conjugated secondary antibodies. Note that Flag-tagged Cav1 was detected at the very few deeply invaginated caveolae in PRTFKO + Cav1 cells [F, green arrow], at shallow caveolar structures [F, black arrowhead] as well as at flat membrane areas [F, blue circle shows cluster] but is virtually absent from replica of PTRFKO cells irrespective of membrane profile evaluated [E]. Bars, 100 nm. **(G)** Densities of Cav1-positive non-caveolar structures with diameters ≥ 300 nm (giant dolines). Plots: mean values ± s.e.m; WT, $n$ = 40 images/ROIs (P-face) and 7 images/ROI (E-face); PTRFKO, $n$ = 46 images/ROIs (P-face) and 6 images/ROI (E-face); PTRFKO + Cav1, $n$ = 44 images/ROIs (P-face) and 6 images/ROI (E-face); PTRFKO + PTRF (rescue), $n$ = 45 images/ROIs (P-face) and

6 images/ROI (E-face), pooled from 3 independent experiments, 12 replicates per condition were examined. Statistics: two-tailed Student's t-test [A-D, p values: all <0.0001] and Kruskal-Wallis with Dunn's post-test [G, p values: WT vs. PTRFKO + Cav1, 0.0005; WT vs. PTRFKO + PTRF, 0.0003; PTRFKO + Cav1 vs. PTRFKO + PTRF, 0.0003], respectively. **(H, I)** dSTORM pipeline and image/analysis examples. Blinks (top left) are filtered by global density-based homogenization (top right). Density of clusters obtained with DBSCAN (bottom right, colors represent different clusters) is computed (bottom left). Images reconstructed from coordinate-maps of blinks at 16 nm/pixel resolution. **(J)** YAP nucleocytoplasmic distribution analysis pipeline. ~2000 cells were imaged per genotype and hypoosmotic treatment condition. A first analysis step included standard filtering of non-cell objects, border-located and/or mitotic cells. YAP nucleocytoplasmic distribution was computed from intranuclear and perinuclear ('ring') region masks. Features were then collected (length-width ratio, area and local confluency ("neighbor fraction")). Cells exhibiting values for any of these features outside the interval ($\mu$ control ± SD control) were excluded from the analysis across conditions (in these example images, highlighted in red). **Source numerical data are available in source data**.

# Reporting Summary

## Statistics

For all statistical analyses, confirm that the following items are present in the figure legend, table legend, main text, or Methods section.

| n/a | Confirmed | |
|---|---|---|
| ☐ | ☒ | The exact sample size (*n*) for each experimental group/condition, given as a discrete number and unit of measurement |
| ☒ | ☐ | A statement on whether measurements were taken from distinct samples or whether the same sample was measured repeatedly |
| ☐ | ☒ | The statistical test(s) used AND whether they are one- or two-sided<br>*Only common tests should be described solely by name; describe more complex techniques in the Methods section.* |
| ☒ | ☐ | A description of all covariates tested |
| ☐ | ☒ | A description of any assumptions or corrections, such as tests of normality and adjustment for multiple comparisons |
| ☐ | ☒ | A full description of the statistical parameters including central tendency (e.g. means) or other basic estimates (e.g. regression coefficient) AND variation (e.g. standard deviation) or associated estimates of uncertainty (e.g. confidence intervals) |
| ☐ | ☒ | For null hypothesis testing, the test statistic (e.g. *F*, *t*, *r*) with confidence intervals, effect sizes, degrees of freedom and *P* value noted<br>*Give P values as exact values whenever suitable.* |
| ☒ | ☐ | For Bayesian analysis, information on the choice of priors and Markov chain Monte Carlo settings |
| ☒ | ☐ | For hierarchical and complex designs, identification of the appropriate level for tests and full reporting of outcomes |
| ☒ | ☐ | Estimates of effect sizes (e.g. Cohen's *d*, Pearson's *r*), indicating how they were calculated |

*Our web collection on statistics for biologists contains articles on many of the points above.*

## Software and code

Policy information about availability of computer code

| Data collection | LAS AF V 4.0.0. 11706 software; LAS-AF 2.6.0. build 7266 TIRF; Carl Zeiss ZEN, software 2011 64bits; EM902A, Zeiss; Nikon Corp. Software NIS Elements AR 4.30.02. Build 1053 LO, 64 bits; LABVIEW software 2015; MatLab 8.6; VistaVision_x64_V4.2_Build 364 software. |
|---|---|
| Data analysis | Super-resolution images were analyzed using LAS AF V 4.0.0. 11706 software, ThunderSTORM v1.3 and ImageJ v1.53q; Electron tomograms were computed and segmented using the software IMOD Package (Kremer JR, Mastronarde DN, McIntosh JR. Computer visualization of three-dimensional image data using IMOD. J Struct Biol. 1996, doi: 10.1006/jsbi.1996.0013).; Automated confocal microscopy images were acquired on a Opera HCSII station and analyzed using the Acapella studio image analysis suite (Perkin Elmer, v2.6); FLIM images were analyzed using SimFCS software, LFD, University of California, Irvine, CA and VistaVision_x64_V4.2_Build 364 software; In-silico data was analyzed by Rosetta software suite v3.8 and v3.9 (www.rosettacommons.org); TIRF videos were obtained using LAS-AF 2.6.0. build 7266 software; Optical Stretching data acquisition and analysis was done with a custom-built LABVIEW software(v 2015); Excel and GraphPad Prism (v 7.0) softwares were used to analyze statistical data and Image J was used for particle tracking in magnetic tweezers experiments, confocal and electron microscopy images analysis. A detailed report of the mathematical model can be found as supplementary Note 1. |

For manuscripts utilizing custom algorithms or software that are central to the research but not yet described in published literature, software must be made available to editors and reviewers. We strongly encourage code deposition in a community repository (e.g. GitHub). See the Nature Portfolio guidelines for submitting code & software for further information.

## Data

Policy information about availability of data

All manuscripts must include a data availability statement. This statement should provide the following information, where applicable:

- Accession codes, unique identifiers, or web links for publicly available datasets
- A description of any restrictions on data availability
- For clinical datasets or third party data, please ensure that the statement adheres to our policy

Previously published data that were re-analyzed here are available under accession code Uniprot ID: ID: P49817, residues2-178 (mature Cav1 protein). We have deposited the following files in Zenodo: dataset from YAP experiments (DOI: 10.5281/zenodo.7061911), script for YAP analysis (DOI: 10.5281/zenodo.7061924), and STORM images set (DOI: 10.5281/zenodo.7062213). Source data are provided with this study. All other data supporting the findings of this study are available from the corresponding author on reasonable request.

# Field-specific reporting

Please select the one below that is the best fit for your research. If you are not sure, read the appropriate sections before making your selection.

☒ Life sciences  ☐ Behavioural & social sciences  ☐ Ecological, evolutionary & environmental sciences

For a reference copy of the document with all sections, see nature.com/documents/nr-reporting-summary-flat.pdf

# Life sciences study design

All studies must disclose on these points even when the disclosure is negative.

| | |
|---|---|
| Sample size | The sample size was set by the clarity of the phenotype. The lower limit of the sample size was set so it was sufficient to observe clearly the differences (p value below 0.05), while the upper limit was set so it was easily reproducible at a reasonable time for an independent lab. In experiments were a tendency was observed with a small sample size, sample size was increased to determine whether the tendency was statistically significant or not. In some cases the tendency was confirmed and validated statistically while in other cases it was not, in which case the sample size was not further increased. |
| Data exclusions | No data were excluded from the analyses unless otherwise indicated. For STORM analysis: Clusters touching the ROI borders were excluded (as indicated in the manuscript). For FRIL analysis: no strongly scattering data points were excluded but all quantitative evaluation data points were taken into account and averaged to fully represent biological and technical variabilities. |
| Replication | All attempts at replication were successful for the experiments present in the paper. The number of replicates is indicated in each figure legend. |
| Randomization | Experiments were not randomized. |
| Blinding | Experimenter blinding was not possible as acquisition and analysis were performed by the same investigator. Important information (i.e. GFP label) was also required precluding classic randomization and blinding. |

# Reporting for specific materials, systems and methods

We require information from authors about some types of materials, experimental systems and methods used in many studies. Here, indicate whether each material, system or method listed is relevant to your study. If you are not sure if a list item applies to your research, read the appropriate section before selecting a response.

### Materials & experimental systems

| n/a | Involved in the study |
|---|---|
| ☐ | ☒ Antibodies |
| ☐ | ☒ Eukaryotic cell lines |
| ☒ | ☐ Palaeontology and archaeology |
| ☒ | ☐ Animals and other organisms |
| ☒ | ☐ Human research participants |
| ☒ | ☐ Clinical data |
| ☒ | ☐ Dual use research of concern |

### Methods

| n/a | Involved in the study |
|---|---|
| ☒ | ☐ ChIP-seq |
| ☒ | ☐ Flow cytometry |
| ☒ | ☐ MRI-based neuroimaging |

## Antibodies

| | |
|---|---|
| Antibodies used | The following primary antibodies were used: The following primary antibodies were used: rabbit monoclonal anti-mouse Caveolin-1, D46G3 Cell signalling XP®#3267; rabbit anti-caveolin 1, sc-894, SantaCruz; mouse monoclonal anti YAP, 63.7, sc-101199, Santa Cruz; |

monoclonal anti-Flag, M2, Sigma/F1804; rabbit polyclonal anti-mouse PTRF, ab48824, Abcam; mouse monoclonal anti-alpha tubulin, ab7291, Abcam; mouse anti-GFP, cat# 118114460001, Roche;  rat monoclonal anti-mouse total beta 1 integrin, clone MB1.2, MAB1997 Millipore; mouse monoclonal Cav1 SIGMA SAB4200216; Ubiquitin (Enzo, ENZ-ABS840-0100).

Validation

- Anti-caveolin-1, D46G3: this antibody is widely used and validated for western blot and immunofluorescence, applications used in this study, as indicated in the manufacturer´s website. This antibody has been previously validated for WB and immunostaining (DOI: 10.1074/jbc.M005448200).
- Anti-caveolin 1, sc-894: this antibody has been previously validated for EM (DOI: 10.7554/elife.29854)
- Anti mouse monoclonal Cav1 SIGMA SAB4200216: this antibody has been previously validated (DOI: 10.18632/oncotarget.15302)
- Anti Ubiquitin (Enzo, ENZ-ABS840-0100): this antibody has been previously validated (DOI: 10.1073/pnas.2114405119)
- Anti-YAP: this antibody has been extensively used in the YAP field (DOI: 10.1038/nature10137).
- Anti-Flag, M2: this antibody is widely used and validated for immunofluorescence as indicated in the manufacturer´s website; here was used for EM and validation is shown in Supplementary Figure 4D-4E.
- Anti-mouse PTRF, ab48824: this antibody is widely used and validated for western blot and immunofluorescence, applications used in this study, as indicated in the manufacturer´s website. This antibody has been previously validated for WB and immunostaining (DOI: 10.1083/jcb.202006178).
- Anti-alpha tubulin, ab7291: this antibody is widely used and validated for western blot, application used in this study, as indicated in the manufacturer´s website. This antibody has been previously validated for WB (DOI: 10.1016/j.redox.2022.102307).
- Anti-GFP, cat# 118114460001: this antibody is widely used and validated for western blot and immunofluorescence, applications used in this study, as indicated in the manufacturer´s website. This antibody has been previously validated for WB (DOI:10.1038/s41467-022-32364-3).
- Anti- total beta 1 integrin, MAB1997: this antibody is widely used and validated for western blot, application used in this study, as indicated in the manufacturer´s website. This antibody has been previously validated for WB (DOI:10.1091/mbc.E14-07-1203).

# Eukaryotic cell lines

Policy information about cell lines

| | |
|---|---|
| Cell line source(s) | PTRFKO mouse embryonic fibroblasts were a kind gift from Prof. Rob Parton and were originally developed in Paul Pilch's lab doi.org/10.1016/j.cmet.2008.07.008. SH-Sy5y neuroblastoma cell line were a kind gift from Dr. Sergio Casas Tintó (Cajal Institute, Madrid, Spain) and were originally obtained from ATCC. |
| Authentication | None of the cell lines were autheticated. |
| Mycoplasma contamination | All cell lines were screened for mycoplasma presence (with Mycoalert PLUS mycoplasma detection kit, Lonza) and the results were negative. |
| Commonly misidentified lines (See ICLAC register) | None. |

