## [Peer Review File · Nature Cell Biology]

Peer Review Information

Journal: Nature Cell Biology

Manuscript Title: Caveolin-1 dolines form a distinct and rapid caveolae-independent mechanoadaptation system

Corresponding author name(s): Dr Miguel del Pozo

Editorial Notes:

Reviewer Comments & Decisions:

Decision Letter, initial version:
--

*Please delete the link to your author homepage if you wish to forward this email to co-authors.

Dear Dr del Pozo,

I apologize once again for the delay.

Your manuscript, "Novel Caveolin1-dolines constitute a distinct and rapid mechanoadaptation system", has now been seen by 3 referees, who are experts in atomic force microscopy and the actomyosin cortex (referee 1); biophysics and computational simulations (referee 2); and caveolae (referee 3). As you will see from their comments (attached below) they find this work of potential interest, but have raised substantial concerns, which in our view would need to be addressed with considerable revisions before we can consider publication in Nature Cell Biology.

Nature Cell Biology editors discuss the referee reports in detail within the editorial team, including the chief editor, to identify key referee points that should be addressed with priority, and requests that are overruled as being beyond the scope of the current study. To guide the scope of the revisions, I have listed these points below. We are committed to providing a fair and constructive peer-review process, so please feel free to contact me if you would like to discuss any of the referee comments further.

In particular, it would be essential to:

A) Provide detailed descriptions of methodology and justifications for underlying conditions for computational models (all Reviewers)

B) Assess physiological relevance by testing whether Caveolin-1-containing dolines are present in physiological conditions (Reviewers #1 and #2)

C) Assess relevance of computational model predictions with more direct functional tests (all Reviewers)

D) Precisely characterize Caveolin-1 subcellular localization (all Reviewers) and assess Caveolin regulation/localization (Reviewer #3)

E) All other referee concerns pertaining to strengthening existing data, providing controls, methodological details, clarifications and textual changes, should also be addressed.

F) Finally please pay close attention to our guidelines on statistical and methodological reporting (listed below) as failure to do so may delay the reconsideration of the revised manuscript. In particular

please provide:

We would be happy to consider a revised manuscript that would satisfactorily address these points, unless a similar paper is published elsewhere, or is accepted for publication in Nature Cell Biology in the meantime.

- ensure that it conforms to our format instructions and publication policies (see below and <https://www.nature.com/nature/for-authors>).

- provide a point-by-point rebuttal to the full referee reports verbatim, as provided at the end of this letter.

- provide the completed Reporting Summary (found here <https://www.nature.com/documents/nr-reporting-summary.pdf>). This is essential for reconsideration of the manuscript will be available to editors and referees in the event of peer review. For more information see <http://www.nature.com/authors/policies/availability.html> or contact me.

When submitting the revised version of your manuscript, please pay close attention to our [href="https://www.nature.com/nature-research/editorial-policies/image-integrity">Digital Image Integrity Guidelines](https://www.nature.com/nature-research/editorial-policies/image-integrity). and to the following points below:

- that unprocessed scans are clearly labelled and match the gels and western blots presented in figures.

- that control panels for gels and western blots are appropriately described as loading on sample processing controls

- all images in the paper are checked for duplication of panels and for splicing of gel lanes.

Nature Cell Biology is committed to improving transparency in authorship. As part of our efforts in this direction, we are now requesting that all authors identified as 'corresponding author' on published papers create and link their Open Researcher and Contributor Identifier (ORCID) with their account on

the Manuscript Tracking System (MTS), prior to acceptance. ORCID helps the scientific community achieve unambiguous attribution of all scholarly contributions. You can create and link your ORCID from the home page of the MTS by clicking on 'Modify my Springer Nature account'. For more information please visit www.springernature.com/orcid.

This journal strongly supports public availability of data. Please place the data used in your paper into a public data repository, or alternatively, present the data as Supplementary Information. If data can only be shared on request, please explain why in your Data Availability Statement, and also in the correspondence with your editor. Please note that for some data types, deposition in a public repository is mandatory - more information on our data deposition policies and available repositories appears below.

[Redacted]

We would like to receive a revised submission within six months.

We hope that you will find our referees' comments, and editorial guidance helpful. Please do not hesitate to contact me if there is anything you would like to discuss.

Best wishes,

Daryl

Daryl J.V. David, PhD

Senior Editor, Nature Cell Biology
Consulting Editor, Nature Communications
Nature Portfolio

Heidelberger Platz 3, 14197 Berlin, Germany
Email: daryl.david@nature.com
ORCID: <https://orcid.org/0000-0002-9253-4805>

Reviewers' Comments:

Reviewer #1:

Remarks to the Author:

Reviewer report on "Novel caveolin1-dolines constitute a distinct and rapid mechanoadaptation

system" by F. Lolo et al. (2021)

In this manuscript the authors report that caveolin-1-mediated invaginations "dolines" provides plasma membrane buffering independently from caveolae when membrane tension is increased, therefore providing additional membrane flexibility for mechanoadaptation. The manuscript explores the role of caveolin1 (Cav1) in modulating plasma membrane curvature when moderate to high mechanical tensile cues are applied to the membrane to provide cell mechanical protection and survival. The authors use multidisciplinary approaches including confocal and superresolution fluorescence microscopy, atomic force microscopy, optical tweezers, magnetic tweezers, Freeze fracture electron microscopy, and computational modeling. The authors show that Cav1 provides plasma membrane protection by increased deformability due to Cav1 invaginations unfolding/flattening. This Cav1-mediated plasma membrane curvature modulation and protection mechanism is independent of canonical caveolae. In addition, the authors show that this Cav1 invaginations are sensitive to moderate membrane tension, while standard caveolae structures are sensitive to higher membrane tensions. Therefore, the authors show a complementary plasma membrane buffering system that allow cells to adapt to a wide range of membrane forces. Most of the experiments seem to be performed carefully, and in my view the manuscript shows interesting relevant results. However, there are some experimental technical aspects, data analysis, and some additional clarifications that the authors need to address.

Broadly speaking, I see a clear contribution of this paper towards understanding the role of caveolae and specific caveolae membrane scaffolding proteins for plasma membrane changes in response to excessive mechanical stimuli to provide protection. The presented research work is interesting, and I felt it could be of broad interest to the Nature Cell Biology readership and I may recommend its publication after the authors address the following major and minor comments:

Major comments:

1) Figure 1 panels G and H, Could the authors clarify why the graphs Y axis are so different in magnitude? The graphs represent the percentage of dead cells after exposed to hypoosmotic shock for 1 minute. I would have expected to observe similar cell dead levels for PTRFKO cells in both graphs (~6% or less than 10%) and PTRFKO+Cav1 cells levels being much lower than PTRFKO cells that will indicate increased mechanoprotection directly correlated with Cav1 expression levels. However, the percentage of dead cells in graph Fig. 1H are more than 3-5 times higher than Fig. 1G and even the observed cell death of PTRFKO+Cav1 cells in Fig.1H are much higher than PTRFKO cells in Fig. 1G. Is there any difference in the hypoosmotic shock assay used between the two graphs (duration, level of hypoosmotic shock)? Or maybe there could be undesired influence on cell death from the centrifugation step? The authors must clarify these apparent conflicting results.

2) Figures 1, 2, 3, and 7 diagrams, I suggest that the manuscript diagrams should be recreated using a graphics design software (i.e. Adobe Photoshop) for improved description of the working models and experimental approaches. The current schematics are low quality and sloppy. In addition, Figures 1 and 7 illustrations would greatly be benefitted if a clear legend is included to specify the distinct molecular components.

3) For Atomic Force Microscopy using quasi-static force curves to determine the cellular Young's modulus of 2D adherent cells, the location where the measurements were performed is critical to be clearly mentioned in the manuscript main text. All AFM measurements for all cells and conditions were

performed on the same cellular region, nuclear region?

4) In addition, the materials and methods description for the AFM measurements state that cells were probed at room temperature. There is evidence published in peer-reviewed journals describing the changes in cell mechanical properties when probed at room temperature (21°C) and physiological temperature (37°C) (Sunyer, R. et al. Phys. Biol. 2009, <https://iopscience.iop.org/article/10.1088/1478-3975/6/2/025009>). The authors must discuss the biological implications of measuring the Young's modulus at room temperature rather than at physiological temperature.

5) Along these lines, the authors must clarify at what temperature the cells were probed when performing; (i) optical stretching, magnetic tweezers, optical tweezers, and real-time deformability cytometry. Again, cell mechanics are temperature dependent, therefore performing these live cells biomechanical approaches using another temperature rather than physiological 37°C will provide results that are not entirely physiologically relevant and could significantly impact their interpretation with caveolae and caveolin 1 plasma membrane buffering and mechanoadaptation.

6) The authors must provide in the materials and methods or supplementary information a complete detailed description of the real-time deformability cytometry approach used in this manuscript and not say the description can be found elsewhere. This will increase clarity and reproducibility.

7) Supplementary Figure 2, panels F and G; to me it seems that there are no significant statistical differences between all conditions since the data for all conditions are highly spread (large deviations). Could the authors better describe the statistical analysis method used to evaluate this plot and get those apparently solid p-values ($p=10^{-8}$ to 10^{-9}).

8) Figure 3, panel C; the graph represents the change in mean tether force after hypoosmotic shock of PTRFKO+Cav1 cells with different Cav1 expression levels. The plotted data possesses a high level of variability that is hard to argue the use of linear regression to fit the data, describe the data behavior, and draw any robust conclusion. The linear fit shown in this plot is highly biased and does not represent the data. Furthermore, could the authors comment and clarify what is the biological and physiological meaning of having negative values for the change in mean tether force ($\Delta F/F_{iso}$)? Is the PM nanotube instead of pulling the bead now pushing? If it is plasma membrane buffering or release of excess PM for a given force there will be a large extension without an increase in force, but that won't reflect in a negative (opposite direction) mean tether force.

9) Figure 6 A-C, for the mathematical modeling used to investigate the effect of membrane tension for invaginations and caveolae-like structures formation and maintenance, are the membrane tension levels (from 0-0.5mN/m) used in simulations physiologically relevant? The authors must provide references that support the use of the membrane tension values or provide a reasonable explanation of why these membrane tension values are relevant.

10) Figure 6E, for YAP nuclear translocation experiments there are some apparent conflicting results between some of the data in the graph and fluorescence images. To me, the YAP fluorescence image for PTRFKO+PTRF at 1/7 hypoosmotic shock is not clear and it may suggest a reduced nuclear translocation in comparison to the slight increase shown in the plot (this is not a good image, and the authors should replace it for a better image). In addition, the YAP fluorescence image for PTRFKO+PTRF at 1/20 hypoosmotic shock, to me, seems to have a reduced nuclear translocation in

comparison to the PTRFKO cells and PTRKO+Cav1 cells (the image does not show the dramatic increase depicted in the plot).

11) The manuscript will significantly benefit if a true control is measured using all the experimental approaches and contrasted/compared with all the conditions. Throughout the manuscript the authors claim that the use of PTRFKO+PTRF cells as a control is sufficient since is capable in developing "normal" caveolae-like structures with a low expression of Cav1. However, a true control is required to observe the experimental results of unperturbed MEF cells, capable of producing physiological caveolae and other plasma membrane structures. I suggest the authors performed experiments using unperturbed MEF WT cells and provide the results in the revised manuscript.

Minor comments:

1) In the Introduction, page 4; the sentence "Furthermore, we do not know whether the full caveolar structure..." is too long. Consider splitting the long sentence into two shorter sentences.

2) Supplementary Figure 1; statistical analysis and significance labels seems to be missing in presented data for western blot quantification of Cav 1 expression levels in Supplementary Fig. 1D.

3) Page 9, sentence "As expected, due to the lack of caveolae-dependent..." I suggest at the end of the sentence to add reference "Figure 1G".

4) Page 17; I suggest the authors include in the supplementary information the data claiming that "PTRFKO MEFs could be hardly imaged, showing very few particles of labelled Cav1 (not shown) under our experimental conditions.

5) Supplementary Figure 3, Apparently only 8 cells per condition were performed for Fluorescence Lifetime experiments. More cells need to be measured for a more robust analysis and data interpretation. The data variability for these measurements are also too high and its hard to see significant differences between all conditions measured.

6) Page 27, again, another statement without data supporting it. Please, provide the data for the following statement "Tilt series can be applied for analyses of freeze-fracture ..., Cav1-decorated membrane structures in PTRFKO-Cav1 cells (our unpublished data).".

7) Supplementary Figure 5, statistical analysis and significance labels are missing in presented data for variance of dGP/angle in Supplementary Fig. 5F. In addition, I suggest the authors fit/add gaussian distribution to the distribution plots presented in Fig 5 A, B, and C). It's hard to discern any difference between the added distributions before and after stretching. This is most important for the PTRFKO+PTRF cells conditions since is the one that apparently should show differences.

8) Supplementary Figure 6, I also suggest fit/adding gaussian distributions to the distribution/histograms graphs included in Fig 6A.

Reviewer #2:

Remarks to the Author:

Lolo&al discuss the contribution of two proteins, Cav1 and PTRF which are constituents of caveolins on the cell membrane, to cell membrane area regulation. The authors study the properties of cell lines expressing Cav1, Cav1 and PTRF, or none of the two proteins, through different assays. They find that cells expressing Cav1 or both Cav1 and PTRF tend to be softer and more resistant to osmotic shocks than cells which do not express these two proteins. The authors then proceed to analyse superresolution and electron microscopy images of the cell membrane under different conditions. Super resolution images do not show a clear difference in clusters of Cav1 with or without PTRF. Electron microscopy images reveal large, shallow membrane invaginations in Cav1 expressing lines, which contrast with standard caveolae. The authors then develop a minimal physical model representing Cav1 and PTRF as molecules inducing low or high spontaneous curvature, and show that under general conditions these molecules give rise to physically distinct response to membrane tension, with Cav1-like molecules responding more strongly at low tension. These ideas are tested by subjecting cells to a range of osmotic shocks and quantifying the effect of the shock on cell death or YAP nuclear translocation.

In this work the authors bring together an impressive number of experimental techniques to investigate the role of Cav1, without PTRF, in regulating cell membrane area. The data supporting the existence of the ability of Cav1 to form membrane reservoirs in the absence of PTRF is generally convincing.

Major comments:

- Figures 4 and 5 and the data on membrane shape and Cav1 distribution seem especially interesting and in my view raise many questions which are not discussed in the manuscript. If it is correct that black dots in Figs. 5A-C indicate the position of single Cav1-molecules through immunogold labelling, can one expect that all Cav1 molecules are labelled with this technique? If so, can their position be correlated to the membrane shape (in line with the mathematical model discussed in Fig. 6); for instance is the Cav1 local density related to membrane curvature or the depth of membrane invaginations? Isn't it that they are rather sparse and doesn't this preclude how much membrane spontaneous curvature they can introduce? How can the distribution of molecules in Figs. 5A-C be understood in relation to clusters seen by superresolution in Fig. 4A? It looks like these distributions are at different scales, but are they compatible with each other; so for instance should one see Figs. 5A-C as close-up views of a Cav1 dense region in Fig. 6A? In general is it possible to use the data of Fig. 5, notably the shapes seen in Figs 5N,P, R to further constrain the mathematical model?

- I could not find a detailed discussion of the continuum model (results in Fig. 6) in the supplementary information. Is this part missing?

- Despite this missing information, judging from Fig. 6 and Ref. 48, the continuum model appears to be sound, quite generic, which is well adapted to the study here, and gives interesting results. Two of the most interesting and general results are the appearance and disappearance of hysteretic behaviour, depending on the spontaneous curvature introduced by Cav1 or PTRF; and the contrast seen in the membrane response to tension by domain splitting or snapping. Model testing is relatively indirect, based on percentage of cell death or YAP translocation following an osmotic shock. Would there be a way to test these model predictions with a more direct quantitative assay; for instance with membrane tether pulling as described in Fig. 3?

- I understand that Laurdan generalised polarisation is related to the degree of packing in the

membrane. The relationship between this quantity and the continuum theory, which sounds more coarse-grained, is not clear to me?

- While the use of a « pure » Cav1 expressing cell without PTRF allows the authors to study its effect separately, it is not clear whether Cav1-containing « dolines » are actually present in physiological conditions. Is it possible to verify this experimentally?

- I find that Fig. 7 is difficult to follow and seems to go largely beyond what the authors are studying in the manuscript.

Other comments:

- I find the use of the word « particle » to refer to clusters in Fig.4 confusing. Particle evokes a single object, but I assume that the authors are referring to clusters containing multiple molecules?

- Page 17, I didn't see which data supports the sentence « sizes heterogeneity and density variability were more evident in PTRFKO+Cav1 MEFs cells than the control cells expressing endogenous Cav1»?

Reviewer #3:

Remarks to the Author:

Membrane domains known as caveolae play important roles in mechanoprotection and mechanotransduction. The manuscript by Lolo et al addresses the interesting question of whether the major structural protein of caveolae- caveolin-1- can fulfill similar roles independently of caveolae. To test this idea, the authors use a broad range of sophisticated experimental and computational approaches to compare the properties of caveolin in the presence and absence of PTRF/cavin-1, using reconstituted PTRF-/- MEFs as a model. Through an elegant and comprehensive series of studies, they show that shallowly curved patches of caveolin-1 form on the plasma membrane in the absence of PTRF that endow cells with the ability to respond to low and medium mechanical forces. Based on their observations, the authors propose an interesting model that suggests that caveolae serve as mechanical switches, whereas dolines serve as springs. Thus, dolines serve as a fundamentally different, yet complementary mechanoadaptation system to caveolae.

Overall, these findings represent a substantial advance in our understanding of the biological functions of caveolin as well as physical mechanisms that contribute to mechanoadaptation. The overall significance of the work is thus very high. In addition, the work is clearly presented, utilizes state-of-the-art techniques, appears to be high quality, and is based on convincing and well controlled data. Given these many strengths, I am very enthusiastic about this manuscript. There are however several points that require further clarification.

Major questions

1. To help place the current work in better context, it would be helpful if the authors could briefly discuss current models for how other components of the plasma membrane contribute to the ability of cells to sense and respond to mechanical forces.
2. Previous studies have shown that under conditions where cavin-1 levels are knocked down, caveolin-1 undergoes accelerated degradation as the result of ubiquitination (Hayer et al. 2010). Because the properties of caveolin-1 in the absence and presence of PTRF are a main focus of this

study, the authors should confirm whether caveolin-1 is ubiquitinated under the conditions of their experiments.

3. The finding that caveolin-1 forms patches of considerable size at the plasma membrane in the absence of PTRF raises the question of how they are held together. As a precedent for this behavior, the manuscript points to previous super resolution studies that reported caveolae-dependent scaffolds of caveolin-1. What is less clear is how these relate to biochemically defined caveolin-1 complexes- specifically, 70S complexes- and whether they form in a cholesterol-dependent manner (Hayer et al. 2010). This question is also relevant to the modeling, since how the domains form appears to be one of the model input parameters (i.e. whether protein-protein interactions or curvature dominate, p.30).

4. Supplementary Figure 3 examines the effect of caveolin-1 versus PTRF expression on cholesterol "stabilization". It is not clear that these experiments contribute much to the story and their interpretation seems both complicated and speculative. If they are left in, at minimum larger version of panels A-E should be provided since they are difficult to read in their current form.

5. Several new sets of experiments and data are introduced in the Discussion, and many of these datasets are presented as being "preliminary". In general, the reader expects all of the new results to be presented in the Results section and the Discussion to focus on its implications. The results currently presented in the Discussion should thus be moved to the Results, or removed for later publication.

6. As part of their study, the authors generate a computational model to investigate the response of caveolae versus dolines to membrane tension. The main text indicates the model is described in the supplementary material and extended supplementary material. However, I was unable to find this information in the provided documents. Without this information, it is not possible to assess the utility or validity of the model.

7. Figure 7 attempts to summarize the main findings of the study, but in its current form it seems overly complicated and confusing. One potential way to simplify it would be to separate the doline model and caveolae model into separate panels or rows so that it is easier to appreciate how they respond to changes in membrane tension, rather than intertwining them. Because actin and filamin are not the focus of the current study, they could potentially be removed for clarity.

Other issues

1. The term "dolines" is not widely used in English (at least, not to my knowledge; I had to look up what it means). It would thus be helpful to include a brief nod to the actual definition in the text so that the reader understands what this means and why it was chosen to describe these structures. Because dolines is a new term, the authors may also want to consider not using it in the title.

2. Supplementary Figure 1 describes several important control experiments including levels of caveolin-1 assessed by Western blotting. It would be helpful if some quantitation of the blots in panels A and B could be included. The authors should also confirm that these images are not saturated.

3. The text labels on many of the figures is extremely small and hard to read. The font size should be increased wherever possible.

4. In Figure 6, the immunofluorescence images in panel E should be made larger. They are difficult to view at their current size.

5. When the in silico model of caveolin-1 is first introduced in the main text, the authors should acknowledge that the structure of caveolin-1 is in fact unknown in order to put these findings into appropriate context.

References

Hayer, A., M. Stoeber, C. Bissig and A. Helenius (2010). "Biogenesis of caveolae: stepwise assembly

of large caveolin and cavin complexes." *Traffic* 11(3): 361-382.

Hayer, A., M. Stoeber, D. Ritz, S. Engel, H. H. Meyer and A. Helenius (2010). "Caveolin-1 is ubiquitinated and targeted to intraluminal vesicles in endolysosomes for degradation." *J Cell Biol* 191: 615-629.

AUTHOR CONTRIBUTIONS – must be included after the Acknowledgements, detailing the contributions of each author to the paper (e.g. experimental work, project planning, data analysis etc.). Each author

should be listed by his/her initials.

Methods should be written concisely, but should contain all elements necessary to allow interpretation and replication of the results. As a guideline, Methods sections typically do not exceed 3,000 words. The Methods should be divided into subsections listing reagents and techniques. When citing previous methods, accurate references should be provided and any alterations should be noted. Information must be provided about: antibody dilutions, company names, catalogue numbers and clone numbers for monoclonal antibodies; sequences of RNAi and cDNA probes/primers or company names and catalogue numbers if reagents are commercial; cell line names, sources and information on cell line identity and authentication. Animal studies and experiments involving human subjects must be reported in detail, identifying the committees approving the protocols. For studies involving human subjects/samples, a statement must be included confirming that informed consent was obtained. Statistical analyses and information on the reproducibility of experimental results should be provided in a section titled "Statistics and Reproducibility".

All Nature Cell Biology manuscripts submitted on or after March 21 2016 must include a Data availability statement as a separate section after Methods but before references, under the heading "Data Availability". For Springer Nature policies on data availability see <http://www.nature.com/authors/policies/availability.html>; for more information on this particular policy see <http://www.nature.com/authors/policies/data/data-availability-statements-data-citations.pdf>. The Data availability statement should include:

- Accession codes for primary datasets (generated during the study under consideration and designated as "primary accessions") and secondary datasets (published datasets reanalysed during the study under consideration, designated as "referenced accessions"). For primary accessions data should be made public to coincide with publication of the manuscript. A list of data types for which

submission to community-endorsed public repositories is mandated (including sequence, structure, microarray, deep sequencing data) can be found here <http://www.nature.com/authors/policies/availability.html#data>.

- Unique identifiers (accession codes, DOIs or other unique persistent identifier) and hyperlinks for datasets deposited in an approved repository, but for which data deposition is not mandated (see here for details <http://www.nature.com/sdata/data-policies/repositories>).
- At a minimum, please include a statement confirming that all relevant data are available from the authors, and/or are included with the manuscript (e.g. as source data or supplementary information), listing which data are included (e.g. by figure panels and data types) and mentioning any restrictions on availability.
- If a dataset has a Digital Object Identifier (DOI) as its unique identifier, we strongly encourage including this in the Reference list and citing the dataset in the Methods.

We recommend that you upload the step-by-step protocols used in this manuscript to the Protocol Exchange. More details can found at www.nature.com/protocolexchange/about.

All imaging data should be accompanied by scale bars, which should be defined in the legend. Cropped images of gels/blots are acceptable, but need to be accompanied by size markers, and to retain visible background signal within the linear range (i.e. should not be saturated). The boundaries of panels with low background have to be demarked with black lines. Splicing of panels should only be considered if unavoidable, and must be clearly marked on the figure, and noted in the legend with a statement on whether the samples were obtained and processed simultaneously. Quantitative comparisons between samples on different gels/blots are discouraged; if this is unavoidable, it should only be performed for samples derived from the same experiment with gels/blots were processed in parallel, which needs to be stated in the legend.

- For line art, graphs, charts and schematics we prefer Adobe Illustrator (.AI), Encapsulated PostScript (.EPS) or Portable Document Format (.PDF). Files should be saved or exported as such directly from the application in which they were made, to allow us to restyle them according to our journal house style.
- We accept PowerPoint (.PPT) files if they are fully editable. However, please refrain from adding PowerPoint graphical effects to objects, as this results in them outputting poor quality raster art. Text used for PowerPoint figures should be Helvetica (preferred) or Arial.
- We do not recommend using Adobe Photoshop for designing figures, but we can accept Photoshop generated (.PSD or .TIFF) files only if each element included in the figure (text, labels, pictures, graphs, arrows and scale bars) are on separate layers. All text should be editable in 'type layers' and line-art such as graphs and other simple schematics should be preserved and embedded within 'vector smart objects' - not flattened raster/bitmap graphics.
- Some programs can generate Postscript by 'printing to file' (found in the Print dialogue). If using an application not listed above, save the file in PostScript format or email our Art Editor, Allen Beattie for advice (a.beattie@nature.com).

SUPPLEMENTARY INFORMATION – Supplementary information is material directly relevant to the conclusion of a paper, but which cannot be included in the printed version in order to keep the manuscript concise and accessible to the general reader. Supplementary information is an integral part of a Nature Cell Biology publication and should be prepared and presented with as much care as the main display item, but it must not include non-essential data or text, which may be removed at the editor's discretion. All supplementary material is fully peer-reviewed and published online as part

of the HTML version of the manuscript. Supplementary Figures and Supplementary Notes are appended at the end of the main PDF of the published manuscript.

The total number of Supplementary Figures (not including the “unprocessed scans” Supplementary Figure) should not exceed the number of main display items (figures and/or tables (see our Guide to Authors and March 2012 editorial <http://www.nature.com/ncb/authors/submit/index.html#suppinfo>; <http://www.nature.com/ncb/journal/v14/n3/index.html#ed>). No restrictions apply to Supplementary Tables or Videos, but we advise authors to be selective in including supplemental data.

GUIDELINES FOR EXPERIMENTAL AND STATISTICAL REPORTING

REPORTING REQUIREMENTS – We are trying to improve the quality of methods and statistics reporting in our papers. To that end, we are now asking authors to complete a reporting summary that collects information on experimental design and reagents. The Reporting Summary can be found here <https://www.nature.com/documents/nr-reporting-summary.pdf> If you would like to reference the guidance text as you complete the template, please access these flattened versions at <http://www.nature.com/authors/policies/availability.html>.

STATISTICS – Wherever statistics have been derived the legend needs to provide the n number (i.e. the sample size used to derive statistics) as a precise value (not a range), and define what this value represents. Error bars need to be defined in the legends (e.g. SD, SEM) together with a measure of centre (e.g. mean, median). Box plots need to be defined in terms of minima, maxima, centre, and percentiles. Ranges are more appropriate than standard errors for small data sets. Wherever statistical significance has been derived, precise p values need to be provided and the statistical test used needs to be stated in the legend. Statistics such as error bars must not be derived from $n < 3$. For sample sizes of $n < 5$ please plot the individual data points rather than providing bar graphs. Deriving statistics from technical replicate samples, rather than biological replicates is strongly discouraged. Wherever statistical significance has been derived, precise p values need to be provided and the

statistical test stated in the legend.

Author Rebuttal to Initial comments

Reviewers' Comments:

Reviewer #1:

Remarks to the Author:

Reviewer report on "Novel caveolin1-dolines constitute a distinct and rapid mechanoadaptation system" by F. Lolo et al. (2021).

In this manuscript the authors report that caveolin-1-mediated invaginations "dolines" provides plasma membrane buffering independently from caveolae when membrane tension is increased, therefore providing additional membrane flexibility for mechanoadaptation. The manuscript explores the role of caveolin1 (Cav1) in modulating plasma membrane curvature when moderate to high mechanical tensile cues are applied to the membrane to provide cell mechanical protection and survival. The authors use multidisciplinary approaches including confocal and superresolution fluorescence microscopy, atomic force microscopy, optical tweezers, magnetic tweezers, Freeze fracture electron microscopy, and computational modeling. The authors show that Cav1 provides plasma membrane protection by increased deformability due to Cav1 invaginations unfolding/flattening. This Cav1-mediated plasma membrane

curvature modulation and protection mechanism is independent of canonical caveolae. In addition, the authors show that this Cav1 invaginations are sensitive to moderate membrane tension, while standard caveolae structures are sensitive to higher membrane tensions. Therefore, the authors show a complementary plasma membrane buffering system that allow cells to adapt to a wide range of membrane forces. Most of the experiments seem to be performed carefully, and in my view the manuscript shows interesting relevant results. However, there are some experimental technical aspects, data analysis, and some additional clarifications that the authors need to address.

Broadly speaking, I see a clear contribution of this paper towards understanding the role of caveolae and specific caveolae membrane scaffolding proteins for plasma membrane changes in response to excessive mechanical stimuli to provide protection. The presented research work is interesting, and I feel it could be of broad interest to the Nature Cell Biology readership and I may recommend its publication after the authors address the following major and minor comments:

We thank the reviewer for her/his positive appreciation of our study.

Major comments:

1) Figure 1 panels G and H, Could the authors clarify why the graphs Y axis are so different in magnitude? The graphs represent the percentage of dead cells after exposed to hypoosmotic shock for 1 minute. I would have expected to observe similar cell dead levels for PTRFKO cells in both graphs (~6% or less than 10%) and PTRFKO+Cav1 cells levels being much lower than PTRFKO cells that will indicate increased mechanoprotection directly correlated with Cav1 expression levels. However, the percentage of dead cells in graph Fig. 1H are more than 3-5 times higher than Fig. 1G and even the observed cell death of PTRFKO+Cav1 cells in Fig.1H are much higher than PTRFKO cells in Fig. 1G. Is there any difference in the hypoosmotic shock assay used between the two graphs (duration, level of hypoosmotic shock)? Or maybe there could be undesired influence on cell death from the centrifugation step? The authors must clarify these apparent conflicting results.

We thank the reviewer for these comments. Although we have always performed the fragility assay under the same experimental conditions, it is true that there is a certain level of variability among experimental replicates, which also prompted us to further characterize the different genotypes with other biophysical approaches as shown in the manuscript. Apart from the duration and the level of hypoosmotic shock, we have observed differences depending on the water batch, although genotypes behaved generally in the same direction. On top of that, if, instead of absolute values (in Figures 1G and 1H), we compare relative differences, we can conclude that dead cell rates in PTRFKO MEFs are ~50% higher as compared to PTRFKO+Cav1 (in Figure 1G), and as much as compared to PTRFKO+Cav1 high (in Figure 1H). To further clarify this point, we have repeated the analysis across all genotypes combining data from different replicates (also including MEFs wild type, as requested by all three referees), and obtained results analogous to those previously observed. We have substituted previous G and H graphs from Figure 1 with a new graph, now shown as Figure 1G.

2) Figures 1, 2, 3, and 7 diagrams, I suggest that the manuscript diagrams should be recreated using a graphics design software (i.e. Adobe Photoshop) for improved description of the working models and experimental approaches. The current schematics are low quality and sloppy. In addition, Figures 1 and 7 illustrations would greatly be benefitted if a clear legend is included to specify the distinct molecular components.

We apologize for the lack of clarity in the diagrams and models. We have now redesigned diagrams of the indicated figures and added legend keys to specify molecular components in Figures 1A-1C, 2E and 7G. Figure 7G has been completely remade using Adobe Photoshop and the working model explained in more detail.

3) For Atomic Force Microscopy using quasi-static force curves to determine the cellular Young's modulus of 2D adherent cells, the location where the measurements were performed is critical to be clearly mentioned in the manuscript main text. All AFM measurements for all cells and conditions were performed on the same cellular region, nuclear region?

We apologize if this was not clearly transmitted, as we agree the location of the AFM measures is critical. Indeed we stated this information in the originally submitted version, in the corresponding material and method section: "Force-displacement curves were recorded at mid-distance between nucleus and cell edge" (previous page 54, now 32). We have now explicitly stated it also in both figure legend (see new schematic drawing in Figure 2C) and main text (page 8).

4) In addition, the materials and methods description for the AFM measurements state that cells were probed at room temperature. There is evidence published in peer-reviewed journals describing the changes in cell mechanical properties when probe at room temperature (21°C) and physiological temperature (37°C) (Sunyer, R. et al. Phys. Biol. 2009, <https://iopscience.iop.org/article/10.1088/1478-3975/6/2/025009>). The authors must discuss the biological implications of measuring the Young's modulus at room temperature rather than at physiological temperature. *We appreciate Reviewer 1 raises an important point, which was thoroughly discussed with our biophysicist collaborators (in fact, two of them are authors also of the reference kindly provided by the referee). Available data suggest that there is no clear general dependence of cell mechanical features on temperature, and differences across experimental systems likely include contextual effects. While Sunyer et al. reported weak cell stiffening when temperature increased from room temperature to 37°C¹, other authors reported a weak negative temperature dependence of cell stiffness, showing that temperature effect on cell mechanics probably depends on specific differences between cell types and experimental conditions². In this regard, it should be noted that our mechanical measurements aiming not at assessing absolute stiffness values but relative mechanical differences across genotypes in a single cell type. Moreover, all orthogonal techniques used to assess cell mechanics in our study provide consistent results. We nonetheless specifically reference this contextuality and its interest regarding future studies in our discussion.*

5) Along these lines, the authors must clarify at what temperature the cells were probed when performing; (i) optical stretching, magnetic tweezers, optical tweezers, and real-time deformability cytometry. Again, cell mechanics are temperature dependent, therefore performing these live cells biomechanical approaches using another temperature rather than physiological 37°C will provide results that are not

entirely physiologically relevant and could significantly impact their interpretation with caveolae and caveolin 1 plasma membrane buffering and mechanoadaptation.

Along the lines of the previous point, we fully agree with reviewer 1 on the importance of specifying these parameters across all techniques used, bearing in mind that comparisons across genotypes always used the same temperature in each experimental approach. Optical stretching and RT-DC require being performed at room temperature, however, magnetic tweezers and plasma membrane pulling with optical tweezers were performed at physiological conditions (37°C). We have now clarified these details in the corresponding material and methods section and thank the reviewer's comment pointing this. As aforementioned, our biophysical comparative characterization across genotypes is consistent regardless of temperature. Accordingly, a report from Josef Käs's lab ³, where the effect of temperature on cell deformation was extensively studied, concluded that temperature affects the rate at which cells deform, but does not have an impact on the relative deformation extent. Furthermore, a recent report from our coauthor Jochen Guck ⁴, where the effect of temperature during optical stretching was specifically studied, indicated that although temperature changes affect cell mechanics, it always influences in the same direction (either increasing or decreasing stiffness) within the same cell type. These bodies of evidence and our own observations across orthogonal assays strongly support a marginal effect, if any, on relative measurements and their interpretation.

6) The authors must provide in the materials and methods or supplementary information a complete detailed description of the real-time deformability cytometry approach used in this manuscript and not say the description can be found elsewhere. This will increase clarity and reproducibility.

We apologize for omitting this methodological information. We have now included a specific section in supplementary information to describe in detail how RT-DC experiments were conducted.

7) Supplementary Figure 2, panels F and G; to me it seems that there are no significant statistical differences between all conditions since the data for all conditions are highly spread (large deviations). Could the authors better describe the statistical analysis method use to evaluate this plot and get those apparently solid p-values ($p=10^{-8}$ to 10^{-9}).

*This is again a pertinent query from reviewer 1, and we thank her/him for it. We have recomputed the analysis: statistical comparisons were by one-way ANOVA with Bonferroni post-test, with significance assigned at $*p < 0.05$ and not significant at $p > 0.05$. A total of 143 PTRFKO MEFs, 182 PTRFKO+Cav1 MEFs and 197 PTRFKO+PTRF MEFs were analyzed. We have now described these details in the corresponding figure legend and the supplementary information and provided the raw data from which the statistical analysis was done.*

8) Figure 3, panel C; the graph represents the change in mean tether force after hypoosmotic shock of PTRFKO+Cav1 cells with different Cav1 expression levels. The plotted data possesses a high level of variability that is hard to argue the use of linear regression to fit the data, describe the data behavior, and draw any robust conclusion. The linear fit shown in this plot is highly biased and does not represent the data. Furthermore, could the authors comment and clarify what is the biological and physiological meaning of having negative values for the change in mean tether force ($\Delta F/F_{iso}$)? Is the PM nanotube instead of pulling the bead now pushing? If it is plasma membrane buffering or release of excess PM for a given force there will be a large extension without an increase in force, but that won't reflect in a negative (opposite direction) mean tether force.

We appreciate reviewer 1 bringing up this question for clarification. As with other experimental systems, we aimed at increasing our statistical power with additional measurements as designed in the previous figure. However, this was not possible with the Optical Tweezers device we used before (from Dr. Lamaze's lab), despite our efforts, as electronics supply shortages after the COVID pandemic have precluded the repair of this apparatus. Nonetheless, we were able to use a similar device from Prof. Patricia Bassereau (also at Curie Institute, now mentioned in acknowledgements) for several OT measurements. As a general comment to this set of experiments, the device set up in Patricia's lab is slightly different (specifically the working mode) which makes it difficult to compare the fluorescence-depending sorting we used in previous measurements. Moreover, as the reviewer probably knows, these are tough experiments. However, to better show the statistical differences among PTRFKO+Cav1 MEFs with different levels of Cav1 expression, we have substituted the linear regression of Figure 3C by three groups of cells sorted according to GFP expression levels (which correlates with Cav1 levels, suppl. Figure 1G-1N). Statistical analysis was then

done by ANOVA with Tukey's multiple comparisons post-test, with significance assigned at * $p < 0.05$ and not-significant at $p > 0.05$. As results now show more clearly, the highest Cav1-expressing cells show statistically significant differences with the lowest ones.

The $\Delta F/F_{iso}$ is calculated by subtracting the tether force measured at iso-osmotic conditions from the one measured after the hypo osmotic shock and divide the result by the iso-osmotic value to obtain a percentage (please have a look at the scheme below): $(F_{hypo} - F_{iso})/F_{iso}$. When the membrane tension is enhanced by the osmotic shock, we obtain a positive value and when the membrane tension is diminished, we obtain negative value. The plasma membrane nanotube in this case is not "pushing on the bead" but just pulling a bit less on the bead than in the isotonic condition. Obtaining negative values of $\Delta F/F$ happens regularly, especially on cells that have good capacities to buffer mechanical cues; published examples can be found elsewhere (Figure 4 A-D in ⁵; Figure 3F in ⁶; Figure 2B in ⁷ and Figure 4H-I in ⁸). The negative values are generally quite close to zero and would rather reflect a light over-compensation to go back to a value in a range close to the isotonic membrane tension.

9) Figure 6 A-C, for the mathematical modeling used to investigate the effect of membrane tension for invaginations and caveolae-like structures formation and maintenance, are the membrane tension levels (from 0-0.5mN/m) used in simulations physiologically relevant? The authors must provide references that support the use of the membrane tension values or provide a reasonable explanation of why these membrane tension values are relevant.

We would like to explicitly offer our sincerest apologies for having omitted the mathematical model; this is quite strange as it was submitted originally, so we think there could have been a mistake during manuscript transfer. This is again an important point for improvement of our manuscript and we thank reviewer 1 for it. 0.5 mN/m is indeed a physiological upper-range magnitude of membrane tension⁹, about one order of magnitude smaller than the lysis tension of the plasma membrane¹⁰⁻¹³ (now stated in page 18). Having said this, the intrinsic tension controlling the assembly-disassembly of curved domains depends essentially on the membrane bending rigidity, on the protein spontaneous curvature, and on the saturation protein coverage (see expression for σ^{eff} after Eq. 13 in the theory supplement- again, we apologize for the omission of this information). These parameters can be reasonably estimated, but these estimations are not fully quantitative. Consequently, the tension range and the thresholds for assembly/disassembly are not quantitative either. However, the model does predict fundamentally different curved domain geometries and mechanical behaviors, at the single domain (Figure 6A) and at the ensemble (Figure 6C1-C3) levels, depending on the magnitude of the spontaneous curvature, and thus provides a plausible mechanistic interpretation of many of the experimental results. The quantitative gap between experiments in living cells and our theoretical model could be bridged using controlled in vitro systems, but this is beyond the scope of the present work. However, our new TIRF and dSTORM analysis do support the mechanical behavior predicted by the mathematical model (see new figures 7A-7E).

10) Figure 6E, for YAP nuclear translocation experiments there are some apparent conflicting results between some of the data in the graph and fluorescence images. To me, the YAP fluorescence image for PTRFKO+PTRF at 1/7 hypoosmotic shock is not clear and it may suggest a reduced nuclear translocation

in comparison to the slight increase shown in the plot (this is not a good image, and the authors should replace it for a better image). In addition, the YAP fluorescence image for PTRFKO+PTRF at 1/20 hypoosmotic shock, to me, seems to have a reduced nuclear translocation in comparison to the PTRFKO cells and PTRKO+Cav1 cells (the image does not show the dramatic increase depicted in the plot).

We apologize as raw field images, without any processing or indexing, were used in the previous manuscript version. It may also be noted that the high content system yields a limited resolution as compared to standard, high resolution confocal microscopes, favoring fast, automated acquisition of extensive datasets. As explained in the Methods section, data were derived from cells filtered according to three main morphological/context parameters: size, relative confluency (as inferred from parameter “Neighbor fraction”, or percentage of perimeter contacting other cells) and length-width ratio. These morphological parameters influence YAP activation state, explain a significant share of the heterogeneity that can be observed in standard cell cultures, and may obscure the contribution of other parameters, such as RNAi perturbations or genotype (see for example Sero and Bakal, Cell Systems 2017¹⁴). We thus filtered cells from each genotype in order to compare cells with similar scores for those pertinent morphological features (taking as reference $\mu \pm SD$ from the PTRFKO+Cav1 population for each parameter), to minimize their contribution to changes observed in YAP nucleocytoplasmic distribution across conditions, as opposed to direct responses to tension. Serendipitously, in the two alluded panels a major share of cells in those fields are indeed filtered out (by low relative area). We now provide an updated figure with two panels showing fields more representative of the computed values for each condition, and a supplementary figure 7 summarizing the analysis and filtering procedure (new panel J).

11) The manuscript will significantly benefit if a true control is measured using all the experimental approaches and contrasted/compared with all the conditions. Throughout the manuscript the authors claim that the use of PTRFKO+PTRF cells as a control is sufficient since is capable in developing “normal” caveolae-like structures with a low expression of Cav1. However, a true control is required to observe the experimental results of unperturbed MEF cells, capable of producing physiological caveolae and other plasma membrane structures. I suggest the authors performed experiments using unperturbed MEF WT cells and provide the results in the revised manuscript.

We thank the reviewer for pointing out this important room for improvement in our study. Recapitulating our study in its whole entirety including this additional cell background is not feasible as different mechanical phenotyping techniques were performed across many different labs and required already many years to be completed. However, as a reasonable alternative, we have performed key additional experiments that demonstrate that PTRFKO+PTRF MEFs behave like a wild type MEF. The following new experiments are now included in the revised version of the manuscript: i) a new fragility assay (as aforementioned) included in Figure 1G, together with new optical tweezers experiments (Suppl. Figure 3A), where PTRFKO+PTRF and wild type MEFs are shown to behave alike in their response to hypoosmotic treatment; ii) a new super-resolution analysis of Cav1 clusters under different osmotic conditions by dSTORM included in a revised version of Figure 7Ei, showing again that PTRFKO+PTRF and wild type MEFs behave similarly, also experimentally showing the splitting behavior of dolines, a process that was only predicted by the mathematical model in the previous version of the manuscript (please see below, third comment from Reviewer 2); iii) and finally, a detailed electron microscopy by FRIL included in a revised version of Figure 5, where we now show that dolines are also present in wild type MEFs, together with two supplementary tilt series movies reconstructing the 3D tomographies of both caveolae and dolines (suppl. Video 3 and 4).

Minor comments:

1) In the Introduction, page 4; the sentence “Furthermore, we do not know whether the full caveolar structure...” is too long. Consider splitting the long sentence into two shorter sentences.

This is an appropriate suggestion and we have now split the sentence into two shorter ones.

2) Supplementary Figure 1; statistical analysis and significance labels seems to be missing in presented data for western blot quantification of Cav 1 expression levels in Supplementary Fig. 1D.

We apologize for this omission. We have now included the statistical analysis of relative Cav1 levels in a revised version of Suppl. Figure 1H.

3) Page 9, sentence “As expected, due to the lack of caveolae-dependent...” I suggest at the end of the sentence to add reference “Figure 1G”.

We now comply with this appropriate suggestion and include the figure reference in the main text.

4) Page 17; I suggest the authors include in the supplementary information the data claiming that “PTRFKO MEFs could be hardly imaged, showing very few particles of labelled Cav1 (not shown) under our experimental conditions.

We now explicitly state this technical limitation as part of a re-designed Suppl. Figure 3B.

5) Supplementary Figure 3, apparently only 8 cells per condition were performed for Fluorescence Lifetime experiments. More cells need to be measured for a more robust analysis and data interpretation. The data variability for these measurements are also too high and its hard to see significant differences between all conditions measured.

We appreciate this important suggestion from reviewer 1, aiming at acquiring more statistical power. We have performed again FLIM experiments analyzing a total of 30 PTRFKO MEFs, 42 PTRFKO+Cav1 MEFs and 42 PTRFKO+PTRF MEFs (thus increasing ‘n’ for each condition by 4-5 fold). We have re-designed the corresponding Suppl. Figure 3 (now Suppl. Figure 4) to show better the significance differences across conditions.

6) Page 27, again, another statement without data supporting it. Please, provide the data for the following statement “Tilt series can be applied for analyses of freeze-fracture ..., Cav1-decorated membrane structures in PTRFKO-Cav1 cells (our unpublished data).”

We apologize for this additional omission and thank the reviewer for spotting it. We have now included two new supplementary videos (Suppl. Video 3 and 4) showing tilt series reconstructing the 3D tomographies of both caveolae and dolines, backing our statement.

7) Supplementary Figure 5, statistical analysis and significance labels are missing in presented data for variance of dGP/angle in Supplementary Fig. 5F. In addition, I suggest the authors fit/add gaussian distribution to the distribution plots presented in Fig 5 A, B, and C). It’s hard to discern any difference

between the added distributions before and after stretching. This is most important for the PTRFKO+PTRF cells conditions since is the one that apparently should show differences.

We apologize for not having included this information; we have now added the requested analysis, labels and gaussian distributions to the corresponding plots and figure legends (Suppl. Figure 5A-5C).

8) Supplementary Figure 6, I also suggest fit/adding gaussian distributions to the distribution/histograms graphs included in Fig 6A.

We have now included the requested gaussian distributions to the corresponding plots (Suppl. Figure 6A).

Reviewer #2:

Remarks to the Author:

Lolo&al discuss the contribution of two proteins, Cav1 and PTRF which are constituents of caveolins on the cell membrane, to cell membrane area regulation. The authors study the properties of cell lines expressing Cav1, Cav1 and PTRF, or none of the two proteins, through different assays. They find that cells expressing Cav1 or both Cav1 and PTRF tend to be softer and more resistant to osmotic shocks than cells which do not express these two proteins. The authors then proceed to analyse superresolution and electron microscopy images of the cell membrane under different conditions. Super resolution images do not show a clear difference in clusters of Cav1 with or without PTRF. Electron microscopy images reveal large, shallow membrane invaginations in Cav1 expressing lines, which contrast with standard caveolae. The authors then develop a minimal physical model representing Cav1 and PTRF as molecules inducing low or high spontaneous curvature, and show that under general conditions these molecules give rise to physically distinct response to membrane tension, with Cav1-like molecules responding more strongly at low tension. These ideas are tested by subjecting cells to a range of osmotic shocks and quantifying the effect of the shock on cell death or YAP nuclear translocation.

In this work the authors bring together an impressive number of experimental techniques to investigate the role of Cav1, without PTRF, in regulating cell membrane area. The data supporting the existence of

the ability of Cav1 to form membrane reservoirs in the absence of PTRF is generally convincing.

We sincerely thank reviewer 2 for her/his very positive appreciation of our work.

Major comments:

- Figures 4 and 5 and the data on membrane shape and Cav1 distribution seem especially interesting and in my view raise many questions which are not discussed in the manuscript. If it is correct that black dots in Figs. 5A-C indicate the position of single Cav1-molecules through immunogold labelling, can one expect that all Cav1 molecules are labelled with this technique? If so, can their position be correlated to the membrane shape (in line with the mathematical model discussed in Fig. 6); for instance is the Cav1 local density related to membrane curvature or the depth of membrane invaginations? Isn't it that they are rather sparse and doesn't this preclude how much membrane spontaneous curvature they can introduce? How can the distribution of molecules in Figs. 5A-C be understood in relation to clusters seen by superresolution in Fig. 4A? It looks like these distributions are at different scales, but are they compatible with each other; so for instance should one see Figs.5A-C as close-up views of a Cav1 dense region in Fig. 6A? In general is it possible to use the data of Fig. 5, notably the shapes seen in Figs 5N,P, R to further constrain the mathematical model?

We thank the reviewer for these comments. Our analysis of super-resolution data from dSTORM is based on 2D coordinates-maps of blinking events. Changes in cluster organization is therefore determined by variations in local blink density, which allows for analysis of large cell areas but without any 3D information about clusters geometry. Freeze-fractured immunogold labelling (FRIL), on the other hand, allows for preservation of membrane topologies from where very detailed 3D information can be extracted, however only smaller areas can be analyzed at a time. Combining both techniques then offers the possibility of overcoming their respective limitations, providing both overall changes in Cav1 domains (dSTORM) and detailed close-ups views with ultrastructural information (FRIL). Despite these advantages, any immunolabeling of thin sections is intrinsically inefficient (due to limitations such as steric hindrances, epitope accessibility -specially for Cav1, which forms a densely packed coat at the plasma membrane-, or

size of the labelling probes), and only a sub-fraction of the target molecules is finally detected¹⁵ (around 2-10% according to our estimations). Therefore, it is certainly true that Cav1 immunolabelling does not seem to correlate with any particular membrane curvature, but since only a fraction of the total pool of Cav1 molecules are detected, we cannot fully determine whether deep structures (such as those in Fig 5M) are decorated with a few Cav1-positive gold dots because they actually contain a few Cav1 molecules or because the labelling was not as efficient in this particular area as it is around proper caveolae for instance (Figure 5O). However, several lines of evidence from both our own *in silico* prediction (now part of the a new version of Suppl. Figure 4R-U) and previous and recent reports pertaining the ability of Cav1 to modify plasma membrane organization (examples include papers where Cav1 discs structures are resolved by Cryo-EM^{16,17}; Cav1 is shown to form heterologous caveolae-like structures in both bacteria and invertebrates^{18,19}; small Cav1 hemi-spheres contain less Cav1 molecules than larger structures²⁰), clearly indicate that: i) Cav1 alone is able to induce membrane curvature and ii) that there seems to be a correlation between membrane curvature and Cav1 density. This is especially important for our mathematical model, as Cav1 local density is one of the key parameters being computed. However, and despite the labelling constrains, briefly sketched above, all the mathematical predictions of our model have been experimentally proved (now showing the splitting behavior that was previously missing, included in the revised version of Figure 7A-7E and Suppl. Videos 11-14). Further refinements of the model would require additional controlled *in vitro* studies, all of which is beyond the scope of our study and will be part of a future work.

- I could not find a detailed discussion of the continuum model (results in Fig. 6) in the supplementary information. Is this part missing?

We sincerely apologize for this gross omission and thank reviewer 2 for having spotted it. Our original manuscript was transferred to Nat Cell Biol from another journal, and we suspect processing and formatting during this transfer might be a cause for this piece of information to be missing. We have now included the model as a separate file in the revised version of the manuscript.

- Despite this missing information, judging from Fig. 6 and Ref. 48, the continuum model appears to be

sound, quite generic, which is well adapted to the study here, and gives interesting results. Two of the most interesting and general results are the appearance and disappearance of hysteretic behaviour, depending on the spontaneous curvature introduced by Cav1 or PTRF; and the contrast seen in the membrane response to tension by domain splitting or snapping. Model testing is relatively indirect, based on percentage of cell death or YAP translocation following an osmotic shock. Would there be a way to test these model predictions with a more direct quantitative assay; for instance with membrane tether pulling as described in Fig. 3?

We thank the reviewer for the positive comments about the continuum model. As indicated by the referee, we started by performing some preliminary experiments with optical tweezers under different osmotic conditions (200mOsm, mild forces and 50mOsm, high forces, see Figure R1). Despite having many technical problems (mainly due to COVID19-derived shortage of chips required for fixing the optical tweezers setup in Dr. Lamaze's lab, which forced us to use a different device, thanks to Prof. P. Bassereau – see response to point 8 from Referee 1), we observed a clear tendency, although not

significant, of PTRFKO+Cav1 MEFs towards lower plasma membrane tension at mild forces (200mOsm) as compared to control and wild type cells. This buffering ability was lost at high forces (50mOsm), which is in agreement with our previous results (Figure 6D). This follows, in a more quantitative way, the mathematical predictions, as Cav1 in the absence of PTRF, is able to respond at very low forces.

Figure R1 for referee inspection

However, in order to better characterize the mathematical predictions indicated by the referee we have performed two complementary experiments now shown in new Figure 7A-7E and Suppl. Videos 11-14. Firstly, we have analyzed PTRFKO+Cav1, PTRFKO+PTRF and wild type MEFs by dSTORM before and after the two hypoosmotic conditions that gave a significant difference in the fragility assay shown in Figure 6D: 1/4 dilution (mild forces, where PTRFKO+Cav1 MEFs showed significant protection as compare to control cells) and 1/20 dilution (high forces, where PTRFKO+PTRF MEFs were significantly more protected than the other genotypes). After imaging and filtering Cav1 blinking events, we applied density-based clustering algorithms (as now explained in the supplementary material and Suppl. Figure 7H and 7I) to determine spatial groupings of blinks in a fair and unbiased manner. The density of each cluster was measured as the number of blinks divided by the area enclosed by its boundary blinks. Since densities may vary depending on the size of the clusters found, we stratified the structures in three

groups depending on their enclosed area: small ($\text{area} < 25^2 \cdot \pi \text{ nm}^2$), medium ($25^2 \cdot \pi \text{ nm}^2 \leq \text{area} < 50^2 \cdot \pi \text{ nm}^2$), and large ($\text{area} \geq 50^2 \cdot \pi \text{ nm}^2$), and analyzed the possible changes across conditions. Interestingly, whereas PTRFKO+Cav1 MEFs already showed a reduction in cluster density after 1/4 dilution, both control and wild type MEFs only showed the same reduction after 1/20 dilution (new Figure 7E). This suggests that after mild forces, Cav1 clusters formed in PTRFKO+Cav1 MEFs break into smaller ones (where Cav1 is less condensed), as compared to Cav1 clusters in control or wild type cells, where they only change after high forces, in line with model predictions.

Secondly, we analyzed *de novo* formation of Cav1 clusters by TIRF microscopy by co-electroporating PTRFKO MEFs with either Cav1GFP and empty vector (Cav1 alone cells), or Cav1GFP and PTRF vectors (control cells), following a previously published protocol²¹. Interestingly, while control cells increased Cav1GFP intensity until a certain plateau was reached, due to PTRF domain size restriction, as previously reported²¹; MEFs expressing Cav1 alone kept growing, showing higher GFP intensities within the same time period (Figure 7A and 7B). This might indicate that in the absence of PTRF, the size of Cav1 domains is not restricted to a specific length, and therefore they can grow larger, forming the giant structures (dolines) we have found by FRIL (Figure 5M). We then treated cells with methyl-beta-cyclodextrin (CD) to study the role of cholesterol in Cav1GFP cluster stabilization. It has been shown that CD treatment induces caveolae flattening increasing PM tension by extracting PM cholesterol²², and therefore represents an additional test of our mathematical model. Strikingly, after cholesterol removal, Cav1GFP clusters started breaking into pieces (new Figure 7C), in accordance with the splitting behavior predicted by the mathematical model (Figure 6C3). This was more frequently observed on cells expressing Cav1 alone as compared to control cells (as visualized in both new Figure 7C, 7D and the new Suppl. Videos 11-14).

- I understand that Laurdan generalised polarisation is related to the degree of packing in the membrane. The relationship between this quantity and the continuum theory, which sounds more coarse-grained, is not clear to me?

We thank the reviewer for this important comment. We agree that the degree of packing as obtained by Laurdan experiments is not captured by the theoretical model. The model by itself only predicts that clustered domains of Cav1 would tend to split with increasing PM tension. Since Cav1 clusters are enriched in cholesterol we inferred that they would have higher order. So, as a primary method to test the

theoretical model, we did the Laurdan experiments as an indirect way to identify whether clustered domains were transforming into homogeneous domains upon stretch, which indeed is the case (Suppl. Figure 5 and 6). We realize that our previous phrasing might have been misleading where we wrote on Page 38 of previous submission that “According to our model, Cav1 clusters are ordered domains that disassemble upon acute increase of PM tension” which has been replaced by “According to our model, Cav1 clusters into domains that disassemble upon acute increase of PM tension” (page 26).

- While the use of a « pure » Cav1 expressing cell without PTRF allows the authors to study its effect separately, it is not clear whether Cav1-containing « dolines » are actually present in physiological conditions. Is it possible to verify this experimentally?

We thank the reviewer for this important comment. We have performed new experiments to show that: i) PTRFKO+PTRF MEFs behave like a wild type MEF and ii) that dolines are also present in physiological conditions. The following experiments are now included in the revised version of the manuscript: i) a new fragility assay included in Figure 1G together with new optical tweezers experiments (Suppl. Figure 3A), where PTRFKO+PTRF and wild type MEFs are shown to behave alike in their response to hypoosmotic treatment; ii) a new super-resolution analysis of Cav1 clusters under different osmotic conditions by dSTORM included in a revised version of Figure 7E, showing again that PTRFKO+PTRF and wild type MEFs behave similarly (also gathering experimental data to support the splitting behavior, as aforementioned); iii) and finally, a detailed electron microscopy by FRIL included in a revised version of Figure 5, where we now show that dolines are also present in wild type MEFs, together with two supplementary tilt series movies reconstructing the 3D tomographies of both caveolae and dolines (suppl. Video 3 and 4).

- I find that Fig. 7 is difficult to follow and seems to go largely beyond what the authors are studying in the manuscript.

We agree with the reviewer and have now done a completely new Fig. 7G which we think conveys better the main message of the paper, and leave the preliminary results from actin cytoskeleton and Filamin staining for a future study.

Other comments:

- I find the use of the word « particle » to refer to clusters in Fig.4 confusing. Particle evokes a single object, but I assume that the authors are referring to clusters containing multiple molecules?

Yes, we are referring to clusters formed by multiple Cav1 molecules. To avoid any further confusion, we have now substituted the term particle for cluster when appropriate.

- Page 17, I didn't see which data supports the sentence « sizes heterogeneity and density variability were more evident in PTRFKO+Cav1 MEFs cells than the control cells expressing endogenous Cav1»?

We apologize for any less –than- optimal or unclear reference to our data. The claim refers to Figure 4E where we analyze frequency across different cluster sizes. We based our observation on the fact that overall statistical deviations are larger for PTRFKO+Cav1 than for the other genotypes, suggesting that heterogeneity is favored when Cav1 forms clusters in the absence of PTRF, as afterwards is fully determined by FRIL (Figure 5). We have now explicitly stated this in the text.

Reviewer #3:

Remarks to the Author:

Membrane domains known as caveolae play important roles in mechanoprotection and mechanotransduction. The manuscript by Lolo et al addresses the interesting question of whether the major structural protein of caveolae- caveolin-1- can fulfill similar roles independently of caveolae. To test this idea, the authors use a broad range of sophisticated experimental and computational approaches to compare the properties of caveolin in the presence and absence of PTRF/cavin-1, using reconstituted PTRF-/- MEFs as a model. Through an elegant and comprehensive series of studies, they show that shallowly curved patches of caveolin-1 form on the plasma membrane in the absence of PTRF that endow cells with the ability to respond to low and medium mechanical forces. Based on their observations, the authors propose an interesting model that suggests that caveolae serve as mechanical switches, whereas

dolines serve as springs. Thus, dolines serve as a fundamentally different, yet complementary mechanoadaptation system to caveolae.

Overall, these findings represent a substantial advance in our understanding of the biological functions of caveolin as well as physical mechanisms that contribute to mechanoadaptation. The overall significance of the work is thus very high. In addition, the work is clearly presented, utilizes state-of-the-art techniques, appears to be high quality, and is based on convincing and well controlled data. Given these many strengths, I am very enthusiastic about this manuscript. There are however several points that require further clarification.

We are truly thankful by these very positive comments from reviewer 3. Her/his subsequent remarks have contributed to strengthen mechanistic concepts as to how dolines assemble and behave in response to different conditions, as opposed to classical caveolae.

Major questions:

1. To help place the current work in better context, it would be helpful if the authors could briefly discuss current models for how other components of the plasma membrane contribute to the ability of cells to sense and respond to mechanical forces.

We thank the reviewer for these comments. We have now briefly mentioned other mechanisms (eisosomes²³ and CLIC-dependent endocytosis²⁴) pertaining plasma membrane tension regulation in the introduction.

2. Previous studies have shown that under conditions where cavin-1 levels are knocked down, caveolin-1 undergoes accelerated degradation as the result of ubiquitination (Hayer et al. 2010). Because the properties of caveolin-1 in the absence and presence of PTRF are a main focus of this study, the authors should confirm whether caveolin-1 is ubiquitinated under the conditions of their experiments.

We are thankful to reviewer 3 for her/his point, important to better understand the behavior of the different genotypes at system level. We have now quantified the level of ubiquitinated Cav1 across

genotypes (including Cav1 wild type and Cav1 KO MEFs –as the negative control-). For this, we used the same assay shown in Hayer et al., 2010²⁵. These results, now included in Suppl. Figure 1E and 1F, show that there is more ubiquitinated Cav1 in PTRFKO+Cav1 MEFs, despite having the same Cav1 protein levels as in the PTRFKO+PTRF cell line and wild type control MEFs (Suppl. Figure 1A and 1C); however, this does not prevent that a certain pool of Cav1 reaches the PM to form the structures we have found by FRIL (Figure 5). Similarly, we detected a slight increase in the ubiquitinated Cav1 forms in PTRFKO MEFs, despite having relatively low levels of Cav1. In summary, in MEFs, similar to HEK 293 cells²⁵, the presence of PTRF prevents the formation of ubiquitinated Cav1 forms.

3. The finding that caveolin-1 forms patches of considerable size at the plasma membrane in the absence of PTRF raises the question of how they are held together. As a precedent for this behavior, the manuscript points to previous super resolution studies that reported caveolae-dependent scaffolds of caveolin-1. What is less clear is how these relate to biochemically defined caveolin-1 complexes- specifically, 70S complexes- and whether they form in a cholesterol-dependent manner (Hayer et al. 2010). This question is also relevant to the modeling, since how the domains form appears to be one of the model input parameters (i.e. whether protein-protein interactions or curvature dominate, p.30).

This is again a pertinent suggestion from reviewer 3 to better understand the biology of our experimental system and which we are thankful for. To better analyze both the dynamics and biochemical organization of Cav1 domains in the absence of PTRF, we have followed two experimental approaches of the many included in the paper referred to by the referee²¹. Firstly, we have followed de novo Cav1GFP clusters formation by TIRF microscopy, comparing PTRFKO MEFs co-electroporated with either Cav1GFP and empty vector (Cav1 alone cells), or Cav1GFP and PTRF vectors (control cells). Interestingly, while control cells increased Cav1GFP intensity until a certain plateau was reached, due to PTRF domain size restriction, as previously reported²¹; MEFs expressing Cav1 alone kept growing, showing higher GFP intensities within the same time period (Figure 7A and 7B). This might indicate that in the absence of PTRF, the size of Cav1 domains is not restricted to a specific length, and therefore they can grow larger, forming the giant structures (dolines) we have found by FRIL (Figure 5). We then treated cells with methyl-beta-cyclodextrin to study the role of cholesterol in Cav1GFP cluster stabilization. Strikingly, after cholesterol removal,

Cav1GFP clusters started breaking into pieces. Remarkably, this process is very reminiscent of the splitting behavior predicted by the mathematical model (Figure 6C3), which we further characterized by new dSTORM experiments (now part of Figure 7E and Suppl. Figure 7H and 7I). Secondly, we obtained by biochemical fractionation Cav1 complexes from PTRFKO+PTRF (control cells) and PTRFKO+Cav1 MEFs running 10-40% continuous sucrose gradients as indicated in ²¹. While in control cells we detected the two previously reported peaks (corresponding to 8S and 70S Cav1 complexes); only one peak corresponding to 8S complexes was found in cells expressing Cav1 alone (Figure 7F). One could possibly argue that either 8S complexes are the building blocks of Cav1 domains in PTRFKO+Cav1 MEFs, or that 70S complexes are less stable in the total absence of PTRF, and therefore lost under the experimental conditions of the sucrose gradients. Methyl-beta-cyclodextrin (CD) treatment disrupted 70S complexes in control cells as previously shown ²¹, while no further effect was observed in the case of PTRFKO+Cav1-treated MEFs (Figure 7F). Altogether, these results might indicate that the larger Cav1 structures (dolines) we have found in PTRFKO+Cav1 MEFs (also in some regions of PTRFKO+PTRF MEFs and now as well in wild type MEFs, although to a lesser extent, FIGURE 5K), are formed by locally clustering 8S complexes, where cholesterol condensation plays an important role for domain stabilization. Interestingly, it has been recently shown that CD treatment leads to membrane tension increase ²². This might indicate that dolines – considered as 8S complexes clusters – undergo splitting after cholesterol removal (as a result of both biochemical and physical modifications), but remain as biochemically resistant smaller 8S domains. These small clusters could well represent the newly discovered Cav1 discs ¹⁷.

4. Supplementary Figure 3 examines the effect of caveolin-1 versus PTRF expression on cholesterol “stabilization”. It is not clear that these experiments contribute much to the story and their interpretation seems both complicated and speculative. If they are left in, at minimum larger version of panels A-E should be provided since they are difficult to read in their current form.

This is a most pertinent query from reviewer 3 which we also received from reviewer 1. To address it, we performed additional FLIM measurements again, increasing the total number of cells per condition by 4-5 fold (N=30 PTRFKO MEFs, N=42 PTRFKO+Cav1 MEFs and N=42 PTRFKO+PTRF MEFs), and re-designed Suppl. Figure 3 (now Suppl. Figure 4A-4Q) to show better the significance differences, also enlarging panels

A-E. PTRFKO+Cav1 MEFs show longer cholesterol lifetimes, indicating that Cav1 alone favors cholesterol stabilization. This is potentially connected with our aforementioned TIRF and biochemical experiments (Figure 7A-7D and 7F), where methyl-beta-cyclodextrin treatment leads to rapid Cav1 clusters disruption (domain splitting, Figure 7C and Suppl. Videos 11-14). All these results might indicate that in the absence of PTRF, Cav1 would constitute specific domains extremely sensitive to any biophysical perturbation (either mechanical or chemical), allowing for a fast mechanoadaptation system.

5. Several new sets of experiments and data are introduced in the Discussion, and many of these datasets are presented as being “preliminary”. In general, the reader expects all of the new results to be presented in the Results section and the Discussion to focus on its implications. The results currently presented in the Discussion should thus be moved to the Results, or removed for later publication.

We have now discussed experiments pertaining membrane order by Laurdan staining (shown in Suppl. Figure 5 and 6) in the results section and removed actin and filamin studies for a later publication.

6. As part of their study, the authors generate a computational model to investigate the response of caveolae versus dolines to membrane tension. The main text indicates the model is described in the supplementary material and extended supplementary material. However, I was unable to find this information in the provided documents. Without this information, it is not possible to assess the utility or validity of the model.

We would like to hereby express our most sincere apologies for having omitted the mathematical model. Because the manuscript was transferred directly from other journal, the possibility exists that there could have been a mistake during the process. We have now included the model as a separate file in the revised version of the manuscript.

7. Figure 7 attempts to summarize the main findings of the study, but in its current form it seems overly complicated and confusing. One potential way to simplify it would be to separate the doline model and caveolae model into separate panels or rows so that it is easier to appreciate how they respond to changes in membrane tension, rather than intertwining them. Because actin and filamin are not the focus of the current study, they could potentially be removed for clarity.

We thank reviewer 3 for this pertinent remark, coincident with that from reviewers 1 and 2. We have now completely redone the final model (now included in Figure 7G) to simplify the main differences between the two mechanoresponsive elements, leaving the preliminary results from actin cytoskeleton and Filamin staining for a future study.

Other issues

1. The term “dolines” is not widely used in English (at least, not to my knowledge; I had to look up what it means). It would thus be helpful to include a brief nod to the actual definition in the text so that the reader understands what this means and why it was chosen to describe these structures. Because dolines is a new term, the authors may also want to consider not using it in the title.

The word dolines is of slovenian origin, referring to the fact that karst processes frequently form big depressions in the ground by collapse of the surface layer. Apart from that we chose this term after the “Gran Dolina”, a huge cavern found in the archaeological site of Atapuerca (Burgos, northern Spain, link to the website: The great Doline). We have now shortly included this explanation in the results section (where we have also cited a Science paper referring to this archeological site ²⁶). Considering the novelty of its discovery we would like to keep it in the title.

2. Supplementary Figure 1 describes several important control experiments including levels of caveolin-1 assessed by Western blotting. It would be helpful if some quantitation of the blots in panels A and B could be included. The authors should also confirm that these images are not saturated.

We have now included quantifications of the indicated blots (now shown in Suppl. Figure 1C and 1D).

3. The text labels on many of the figures is extremely small and hard to read. The font size should be increased wherever possible.

We have enlarged some labels when possible due to size restrictions.

4. In Figure 6, the immunofluorescence images in panel E should be made larger. They are difficult to view at their current size.

We apologize for this mistake. The updated figure in the revised manuscript has been relatively enlarged, also taking on account the clarifications exposed to reviewer 1.

5. When the in silico model of caveolin-1 is first introduced in the main text, the authors should acknowledge that the structure of caveolin-1 is in fact unknown in order to put these findings into appropriate context.

We thank the reviewer for this comment. We have now specifically acknowledged the in silico model when appropriate in the text. During the course of our in-depth revision, the group of Anne K. Kenworthy reported the structure of human Cav1 discs¹⁷, which we now have appropriately cited in the text.

References

- Hayer, A., M. Stoeber, C. Bissig and A. Helenius (2010). "Biogenesis of caveolae: stepwise assembly of large caveolin and cavin complexes." *Traffic* **11**(3): 361-382.
- Hayer, A., M. Stoeber, D. Ritz, S. Engel, H. H. Meyer and A. Helenius (2010). "Caveolin-1 is ubiquitinated and targeted to intraluminal vesicles in endolysosomes for degradation." *J Cell Biol* **191**: 615-629.

References

- 1 Sunyer, R., Trepats, X., Fredberg, J. J., Farre, R. & Navajas, D. The temperature dependence of cell mechanics measured by atomic force microscopy. *Phys Biol* **6**, 025009, doi:10.1088/1478-3975/6/2/025009 (2009).
- 2 Aermes, C., Hayn, A., Fischer, T. & Mierke, C. T. Cell mechanical properties of human breast carcinoma cells depend on temperature. *Sci Rep* **11**, 10771, doi:10.1038/s41598-021-90173-y (2021).
- 3 Tobias R Kießling, Roland Stange, Josef A Kas" & Fritsch, A. W. Thermorheology of living cells—impact of temperature variations on cell mechanics. *New Journal of Physics*, doi:doi:10.1088/1367-2630/15/4/045026 (2013).
- 4 Huster, C. et al. Stretching and heating cells with light—nonlinear photothermal cell rheology. *New Journal of Physics*, doi:https://doi.org/10.1088/1367-2630/aba14b (2020).
- 5 Sinha, B. et al. Cells respond to mechanical stress by rapid disassembly of caveolae. *Cell* **144**, 402-413, doi:10.1016/j.cell.2010.12.031 (2011).
- 6 Torrino, S. et al. EHD2 is a mechanotransducer connecting caveolae dynamics with gene transcription. *J Cell Biol* **217**, 4092-4105, doi:10.1083/jcb.201801122 (2018).

- 7 Dewulf, M. et al. Dystrophy-associated caveolin-3 mutations reveal that caveolae couple IL6/STAT3 signaling with mechanosensing in human muscle cells. *Nat Commun* **10**, 1974, doi:10.1038/s41467-019-09405-5 (2019).
- 8 Echarrri, A. et al. An Abl-FBP17 mechanosensing system couples local plasma membrane curvature and stress fiber remodeling during mechanoadaptation. *Nat Commun* **10**, 5828, doi:10.1038/s41467-019-13782-2 (2019).
- 9 Sens, P. & Plastino, J. Membrane tension and cytoskeleton organization in cell motility. *J Phys Condens Matter* **27**, 273103, doi:10.1088/0953-8984/27/27/273103 (2015).
- 10 Lieber, A. D., Yehudai-Resheff, S., Barnhart, E. L., Theriot, J. A. & Keren, K. Membrane tension in rapidly moving cells is determined by cytoskeletal forces. *Curr Biol* **23**, 1409-1417, doi:10.1016/j.cub.2013.05.063 (2013).
- 11 Gauthier, N. C., Masters, T. A. & Sheetz, M. P. Mechanical feedback between membrane tension and dynamics. *Trends Cell Biol* **22**, 527-535, doi:10.1016/j.tcb.2012.07.005 (2012).
- 12 Wang, G. & Galli, T. Reciprocal link between cell biomechanics and exocytosis. *Traffic* **19**, 741-749, doi:10.1111/tra.12584 (2018).
- 13 Kozlov, M. M. & Chernomordik, L. V. Membrane tension and membrane fusion. *Curr Opin Struct Biol* **33**, 61-67, doi:10.1016/j.sbi.2015.07.010 (2015).
- 14 Sero, J. E. & Bakal, C. Multiparametric Analysis of Cell Shape Demonstrates that beta-PIX Directly Couples YAP Activation to Extracellular Matrix Adhesion. *Cell Syst* **4**, 84-96 e86, doi:10.1016/j.cels.2016.11.015 (2017).
- 15 Seemann, E. et al. Deciphering caveolar functions by syndapin III KO-mediated impairment of caveolar invagination. *Elife* **6**, doi:10.7554/eLife.29854 (2017).
- 16 Han, B. et al. Structure and assembly of CAV1 8S complexes revealed by single particle electron microscopy. *Sci Adv* **6**, doi:10.1126/sciadv.abc6185 (2020).
- 17 Porta, J. C. et al. Molecular architecture of the human caveolin-1 complex. *Sci Adv* **8**, eabn7232, doi:10.1126/sciadv.abn7232 (2022).
- 18 Walser, P. J. et al. Constitutive formation of caveolae in a bacterium. *Cell* **150**, 752-763, doi:10.1016/j.cell.2012.06.042 (2012).
- 19 Bhattachan, P. et al. Ascidian caveolin induces membrane curvature and protects tissue integrity and morphology during embryogenesis. *Faseb J* **34**, 1345-1361, doi:10.1096/fj.201901281R (2020).
- 20 Khater, I. M., Liu, Q., Chou, K. C., Hamarneh, G. & Nabi, I. R. Super-resolution modularity analysis shows polyhedral caveolin-1 oligomers combine to form scaffolds and caveolae. *Sci Rep* **9**, 9888, doi:10.1038/s41598-019-46174-z (2019).
- 21 Hayer, A., Stoeber, M., Bissig, C. & Helenius, A. Biogenesis of caveolae: stepwise assembly of large caveolin and cavin complexes. *Traffic* **11**, 361-382, doi:10.1111/j.1600-0854.2009.01023.x (2010).
- 22 Cox, C. D., Zhang, Y., Zhou, Z., Walz, T. & Martinac, B. Cyclodextrins increase membrane tension and are universal activators of mechanosensitive channels. *Proc Natl Acad Sci U S A* **118**, doi:10.1073/pnas.2104820118 (2021).
- 23 Kabeche, R., Howard, L. & Moseley, J. B. Eisosomes provide membrane reservoirs for rapid expansion of the yeast plasma membrane. *J Cell Sci* **128**, 4057-4062, doi:10.1242/jcs.176867 (2015).

- 24 *Thottacherry, J. J. et al. Mechanochemical feedback control of dynamin independent endocytosis modulates membrane tension in adherent cells. Nat Commun* **9**, 4217, doi:10.1038/s41467-018-06738-5 (2018).
- 25 *Hayer, A. et al. Caveolin-1 is ubiquitinated and targeted to intraluminal vesicles in endolysosomes for degradation. J Cell Biol* **191**, 615-629, doi:10.1083/jcb.201003086 (2010).
- 26 *Bermudez de Castro, J. M. et al. A hominid from the lower Pleistocene of Atapuerca, Spain: possible ancestor to Neandertals and modern humans. Science* **276**, 1392-1395, doi:10.1126/science.276.5317.1392 (1997).

Decision Letter, first revision:

Our ref: NCB-D46470A

27th July 2022

Dear Dr. del Pozo,

Thank you for submitting your revised manuscript "Novel Caveolin1-dolines constitute a distinct and rapid mechanoadaptation system" (NCB-D46470A). It has now been seen by the original referees and their comments are below. The reviewers find that the paper has improved in revision, and therefore we'll be happy in principle to publish it in Nature Cell Biology, pending minor revisions to satisfy the referees' final requests and to comply with our editorial and formatting guidelines.

Please note that we would not require the removal of the structural analysis nor modelling (as otherwise suggested by Reviewer #3) but would encourage you to keep the data in the paper, and would require a discussion to address their utility/limitations in this study. We would also not recommend removal of the data on caveolin on cholesterol stabilization (also as suggested by Reviewer #3), but would require discussion and appropriate citation of relevant literature to discuss potential mechanisms (Reviewer #3).

As the current version of your manuscript is in a PDF format, please email us a copy of the file in an editable format (Microsoft Word or LaTeX)-- we can not proceed with PDFs at this stage.

Thank you again for your interest in Nature Cell Biology Please do not hesitate to contact me if you have any questions.

Sincerely,
Daryl

Daryl J.V. David, PhD

Senior Editor, Nature Cell Biology
Consulting Editor, Nature Communications
Nature Portfolio

Heidelberger Platz 3, 14197 Berlin, Germany
Email: daryl.david@nature.com
ORCID: <https://orcid.org/0000-0002-9253-4805>

Reviewer #1 (Remarks to the Author):

The authors have satisfactorily addressed my raised comments.

Reviewer #2 (Remarks to the Author):

The authors have answered my comments and I am happy to recommend publication of the manuscript.

Reviewer #3 (Remarks to the Author):

The authors have gone to great lengths to address the concerns of the reviewers in the revised manuscript, strengthening it even further. I have two final comments for their consideration.

1. The authors now provide biochemical evidence that caveolin in dolines is likely in the form of 8S complexes. Furthermore, as the authors indicate, a cryoEM structure of 8S complexes of caveolin was recently reported. The section on the structural analysis, in silico modeling, and docking studies of mouse caveolin-1 monomers and dimers is thus no longer relevant and should ideally be removed. I realize that significant work was put into this effort, but given these new experimental developments, leaving these computational models in the manuscript could lend confusion and therefor dilutes the rest of the story.
2. I still do not find the section on the impact of caveolin on cholesterol stabilization to be especially illuminating. If the authors feel this should be included, they should at provide references to support their assertion that the changes in cholesterol lifetimes are somehow linked to changes/differences in membrane order.

Decision Letter, final checks:

Our ref: NCB-D46470A

10th August 2022

Dear Dr. del Pozo,

Thank you for your patience as we've prepared the guidelines for final submission of your Nature Cell Biology manuscript, "Novel Caveolin1-dolines constitute a distinct and rapid mechanoadaptation system" (NCB-D46470A). Please carefully follow the step-by-step instructions provided in the attached file, and add a response in each row of the table to indicate the changes that you have made. Please also check and comment on any additional marked-up edits we have proposed within the text. Ensuring that each point is addressed will help to ensure that your revised manuscript can be swiftly handed over to our production team.

We would like to start working on your revised paper, with all of the requested files and forms, as soon as possible (preferably within one week). Please get in contact with us if you anticipate delays.

In recognition of the time and expertise our reviewers provide to Nature Cell Biology's editorial process, we would like to formally acknowledge their contribution to the external peer review of your manuscript entitled "Novel Caveolin1-dolines constitute a distinct and rapid mechanoadaptation system". For those reviewers who give their assent, we will be publishing their names alongside the published article.

Nature Cell Biology offers a Transparent Peer Review option for new original research manuscripts submitted after December 1st, 2019. As part of this initiative, we encourage our authors to support increased transparency into the peer review process by agreeing to have the reviewer comments, author rebuttal letters, and editorial decision letters published as a Supplementary item. When you submit your final files please clearly state in your cover letter whether or not you would like to participate in this initiative. Please note that failure to state your preference will result in delays in accepting your manuscript for publication.

Cover suggestions

As you prepare your final files we encourage you to consider whether you have any images or illustrations that may be appropriate for use on the cover of Nature Cell Biology.

Nature Cell Biology has now transitioned to a unified Rights Collection system which will allow our Author Services team to quickly and easily collect the rights and permissions required to publish your work. Approximately 10 days after your paper is formally accepted, you will receive an email in providing you with a link to complete the grant of rights. If your paper is eligible for Open Access, our Author Services team will also be in touch regarding any additional information that may be required to arrange payment for your article.

Please note that *Nature Cell Biology* is a Transformative Journal (TJ). Authors may publish their research with us through the traditional subscription access route or make their paper immediately open access through payment of an article-processing charge (APC). Authors will not be required to make a final decision about access to their article until it has been accepted. Find out more about Transformative Journals

Please use the following link for uploading these materials:
[Redacted]

Best regards,

Adam Lipkin
Staff
Nature Cell Biology

On behalf of

Daryl J.V. David, PhD

Senior Editor, Nature Cell Biology
Consulting Editor, Nature Communications
Nature Portfolio

Heidelberger Platz 3, 14197 Berlin, Germany
Email: daryl.david@nature.com
ORCID: <https://orcid.org/0000-0002-9253-4805>

Reviewer #1:

Remarks to the Author:

The authors have satisfactorily addressed my raised comments.

Reviewer #2:

Remarks to the Author:

The authors have answered my comments and I am happy to recommend publication of the manuscript.

Reviewer #3:

Remarks to the Author:

The authors have gone to great lengths to address the concerns of the reviewers in the revised manuscript, strengthening it even further. I have two final comments for their consideration.

1. The authors now provide biochemical evidence that caveolin in dolines is likely in the form of 8S complexes. Furthermore, as the authors indicate, a cryoEM structure of 8S complexes of caveolin was recently reported. The section on the structural analysis, in silico modeling, and docking studies of mouse caveolin-1 monomers and dimers is thus no longer relevant and should ideally be removed. I realize that significant work was put into this effort, but given these new experimental developments, leaving these computational models in the manuscript could lend confusion and therefore dilutes the rest of the story.

2. I still do not find the section on the impact of caveolin on cholesterol stabilization to be especially illuminating. If the authors feel this should be included, they should at provide references to support

their assertion that the changes in cholesterol lifetimes are somehow linked to changes/differences in membrane order.

Author Rebuttal, first revision:

Responses to Reviewer 3's Comments

Reviewer #3: Remarks to the Author:

The authors have gone to great lengths to address the concerns of the reviewers in the revised manuscript, strengthening it even further. I have two final comments for their consideration.

We are again thankful by comments from reviewer 3. Her/his remarks have contributed to strengthen very relevant mechanistic concepts of our study.

1. The authors now provide biochemical evidence that caveolin in dolines is likely in the form of 8S complexes. Furthermore, as the authors indicate, a cryoEM structure of 8S complexes of caveolin was recently reported. The section on the structural analysis, in silico modeling, and docking studies of mouse caveolin-1 monomers and dimers is thus no longer relevant and should ideally be removed. I realize that significant work was put into this effort, but given these new experimental developments, leaving these computational models in the manuscript could lend confusion and therefor dilutes the rest of the story.

We thank the reviewer for this comment. Although it is true that during the course of our in-depth revision, the group of Anne K. Kenworthy reported the structure of human Cav1 discs¹ (as we indicated in the revised version of the manuscript), our model still provides new insights that were not captured in past and current models^{1,2}. Specifically, it includes the whole N-terminal domain of Cav1, currently missing in the recent model¹. Additionally, according to dimer organization, spacing between monomers allows for increased cholesterol condensation, which might be playing a role in dolines stabilization, as both our TIRF studies under cyclodextrin treatment and FLIM analysis indicate. We have now included these ideas in the discussion section (page 19), as requested by the Editor.

2. I still do not find the section on the impact of caveolin on cholesterol stabilization to be especially illuminating. If the authors feel this should be included, they should at provide references to support their assertion that the changes in cholesterol lifetimes are somehow linked to changes/differences in membrane order.

We thank the reviewer for this comment. Although, according to our estimations (included in our mathematical model), cholesterol de-condensation does not seem to efficiently contribute to plasma

membrane (PM) tension buffering, we still consider that FLIM studies provide relevant information to better understand the role of dolines mechanoadaptation. Specifically, enhanced cholesterol condensation in doline-containing cells, might be key for domain stability and sensitivity, as suggested by our TIRF studies under cyclodextrin treatment (Figure 7A-7D), where we observed the splitting behavior. This might indicate that in the absence of PTRF, Cav1 would constitute specific domains extremely sensitive to changes in cholesterol organization, which suggests that their mechanoadaptation capacity is not only restricted to mechanical perturbations but also to any stress impacting on PM cholesterol homeostasis. For these studies we used the fluorescent derivative 25-NBD cholesterol, which has been previously shown to reflect changes in PM membrane order^{3,4,5} as we indicate in the manuscript (References 47, 48 and new reference 49). We have now explicitly stated this in the text in both results (page 9) and discussion (pages 19-20) sections.

References

- 1 Porta, J. C. et al. Molecular architecture of the human caveolin-1 complex. *Sci Adv* **8**, eabn7232, doi:10.1126/sciadv.abn7232 (2022).
- 2 Stoeber, M. Et al. Model for the architecture of caveolae based on a flexible, net-like assembly of Cavin1 and Caveolin discs. *PNAS* **113** (50) E8069-E8078 (2016).
- 3 Ostasov, P. et al. FLIM studies of 22- and 25-NBD-cholesterol in living HEK293 cells: plasma membrane change induced by cholesterol depletion. *Chem Phys Lipids* **167-168**, 62-69, doi:10.1016/j.chemphyslip.2013.02.006 (2013).
- 4 Chattopadhyay, S. H. a. A. in *Fluorescent Methods to Study Biological Membranes Vol. 13* (ed Y. Mély and G. Duportail) 37-50 (Springer-Verlag Berlin Heidelberg 2012, 2013).
- 5 Kaiser, H.J., Lingwood, D., Levental, I., Sampaio, J.L., Kalvodova, L., Rajendran, L., Simons, K., 2009. Order of lipid phases in model and plasma membranes. *Proceedings of the National Academy of Sciences of the United States of America* **106**, 16645–16650.

Final Decision Letter:

Dear Dr del Pozo,

I am writing on behalf of my colleague, Dr. Daryl David, who is out of the office.

I am pleased to inform you that your manuscript, "Caveolin-1 dolines form a distinct and rapid caveolae-independent mechanoadaptation system", has now been accepted for publication in *Nature Cell Biology*.

Over the next few weeks, your paper will be copyedited to ensure that it conforms to *Nature Cell Biology* style. Once your paper is typeset, you will receive an email with a link to choose the appropriate publishing options for your paper and our Author Services team will be in touch regarding any additional information that may be required.

Publication is conditional on the manuscript not being published elsewhere and on there being no announcement of this work to any media outlet until the online publication date in *Nature Cell Biology*.

Please note that *Nature Cell Biology* is a Transformative Journal (TJ). Authors may publish their research with us through the traditional subscription access route or make their paper immediately open access through payment of an article-processing charge (APC). Authors will not be required to make a final decision about access to their article until it has been accepted. Find out more about

Transformative Journals

If you have not already done so, we strongly recommend that you upload the step-by-step protocols used in this manuscript to the Protocol Exchange (www.nature.com/protocolexchange), an open online resource established by Nature Protocols that allows researchers to share their detailed experimental know-how. All uploaded protocols are made freely available, assigned DOIs for ease of citation and are fully searchable through nature.com. Protocols and Nature Portfolio journal papers in which they are used can be linked to one another, and this link is clearly and prominently visible in the online versions of both papers. Authors who performed the specific experiments can act as primary authors for the Protocol as they will be best placed to share the methodology details, but the Corresponding Author of the present research paper should be included as one of the authors. By uploading your Protocols to Protocol Exchange, you are enabling researchers to more readily reproduce or adapt the methodology you use, as well as increasing the visibility of your protocols and papers. You can also establish a dedicated page to collect your lab Protocols. Further information can be found at www.nature.com/protocolexchange/about

All the best,

Christina

==

Christina Kary, PhD
Chief Editor
Nature Cell Biology
1 New York Plaza
Tel: +44 (0) 207 843 4924
